# CaSpER identifies and visualizes CNV events by integrative analysis of single-cell or bulk RNA-sequencing data

Akdes Serin Harmanci[1,5], Arif O. Harmanci[2,5] & Xiaobo Zhou[1,3,4]*

RNA sequencing experiments generate large amounts of information about expression levels of genes. Although they are mainly used for quantifying expression levels, they contain much more biologically important information such as copy number variants (CNVs). Here, we present CaSpER, a signal processing approach for identification, visualization, and integrative analysis of focal and large-scale CNV events in multiscale resolution using either bulk or single-cell RNA sequencing data. CaSpER integrates the multiscale smoothing of expression signal and allelic shift signals for CNV calling. The allelic shift signal measures the loss-of-heterozygosity (LOH) which is valuable for CNV identification. CaSpER employs an efficient methodology for the generation of a genome-wide B-allele frequency (BAF) signal profile from the reads and utilizes it for correction of CNVs calls. CaSpER increases the utility of RNA-sequencing datasets and complements other tools for complete characterization and visualization of the genomic and transcriptomic landscape of single cell and bulk RNA sequencing data.

[1] Center for Computational Systems Medicine, School of Biomedical Informatics, University of Texas Health Science Center at Houston, Houston, TX 77030, USA. [2] Center for Precision Health, School of Biomedical Informatics, University of Texas Health Science Center at Houston, Houston, TX 77030, USA. [3] Department of Integrative Biology and Pharmacology, McGovern Medical School at The University of Texas Health Science Center at Houston, Houston, TX 77030, USA. [4] School of Dentistry, University of Texas Health Science Center at Houston, Houston, TX 77054, USA. [5] These authors contributed equally: Akdes Serin Harmancı, Arif O. Harmanci. *email: Xiaobo.Zhou@uth.tmc.edu

Over the past few years, the development and application of single-cell DNA- and RNA-sequencing methods have revolutionized cancer research[1]. Single-cell RNA-sequencing (scRNA-seq) is a powerful deep molecular profiling method for detecting different cell types, states, and functions in cancer[2–10]. Several previous studies characterized the heterogeneity and crosstalk within the tumor microenvironment in various cancer types using scRNA-seq data[2,4,6]. Single-cell DNA sequencing is another powerful approach for understanding genomic diversity of tumor clonal architecture[1,9]. Even though a number of approaches for simultaneously assaying DNA and RNA information from the same single cell has been developed, it still remains technically challenging to assay both the genome and transcriptome from the same cell[10], particularly in terms of cost and labor.

RNA-seq experiments are predominantly performed for the purpose of estimating gene activity through quantification of gene and transcript expression. These datasets, however, contain a substantial amount of information about the genomic variants in the samples and are severely underutilized. For example, RNA-seq data have been used to identify single nucleotide polymorphisms (SNPs) and short indels[11–13]. Identification of these variants from RNA-seq data increases the utility of RNA-seq experiments significantly compared to using RNA-seq only for gene expression quantification because researchers can integrate a portion of the genomic landscape of the tumor cells (as much as it is revealed by RNA-seq) with the transcriptomic landscape rather than studying the transcriptomic landscape of the cells alone. Identification of other variants can enable an even more complete characterization and present higher utility for RNA-seq data. Among these variants, copy number variants (CNVs) are very important for cancer research because they are a major class of genetic drivers of cancer. Identification of CNVs from RNA-seq data, however, is very challenging because the dynamic and highly non-uniform coverage of the genome by RNA-seq signal makes it very hard to distinguish between deletion and amplification events and the dynamic variation of gene expression levels. Considering the growing number of RNA-seq studies, especially with the release of TCGA[14], ENCODE[15], GTEx[16], Human Cell Atlas (HCA)[17], Human Tumor Atlas Network (HTAN), and Human Biomolecular Atlas Program (HuBMAP) consortium datasets, there is an increasing need for developing CNV inference algorithm from RNA-seq data.

Although there are many tools that identify CNVs from exome- and whole-genome sequencing data[18–20], there is much scarcity of methods for detecting CNVs solely from RNA-seq data[2,21]. One relevant method is inferCNV[2], which enables only visual inspection of expression profiles from scRNA-seq data. Another method is HoneyBADGER[21], which detects CNVs from scRNA-seq data. Another recently published method, clonealign, performs statistical integration of independent single-cell RNA-seq and DNA-seq from human cancers[22].

In this study, we present CaSpER, a statistical framework for the detection and visualization of the CNVs using genome-wide RNA-seq signal profiles. To study CNV events at multiple scales, CaSpER utilizes a multiscale signal-processing framework. CaSpER integrates two types of signals in the multiscale processing. First is the genome-wide gene expression profile. This profile is a vector where the entries are the gene expression levels. The entries are sorted with respect to the location of genes they correspond to. The second signal is the allelic shift signal profile, which is a vector whose elements represent the allelic shift measured at numerous single nucleotide variants and the elements are sorted with respect to the genomic position of the variants. This profile quantifies the genome-wide loss-of-heterozygosity (LOH) events, which have been previously shown to be extremely useful for identifying CNVs[20]. Unlike most other tools, CaSpER does not require a high-quality heterozygous variant call set to generate the allelic shift profile[18,23]. Thus, CaSpER does not require an SNV variant call set as an input. CaSpER utilizes the multiscale decomposition to smooth the expression and allelic shift signals in multiple length scales. This processing removes much of the noise and enhances the copy number information within the expression and allelic shift signals. CaSpER also performs a number of downstream analyses for a comprehensive characterization and visualization of the CNV information. CaSpER identifies and visualizes mutually exclusive and co-occurring CNV alterations. For scRNA-seq experiments, it infers the CNV-based clonal evolution of the cells using the detected CNVs. CaSpER also identifies the gene expression signatures of mutually exclusive CNV sub-clones and performs gene ontology (GO) enrichment analysis. CaSpER performs well in terms of sensitivity compared to other tools using numerous datasets and comparison metrics. In summary, CaSpER broadens the number of potential use cases of RNA-seq datasets since CaSpER can use RNA-seq data to probe the CNV landscape of the cells in addition to their transcriptomic landscapes.

## Results

**Identification of CNV events from RNA-seq data.** The overview of the CaSpER algorithm is shown in Fig. 1 (see Methods). The input to CaSpER consists of aligned RNA-seq reads and the window lengths to be used in multiscale analysis[24].

CaSpER uses expression values and B-allele frequencies (BAFs) from RNA-seq reads to estimate CNV events. The BAF is a relative normalized measure of the allelic intensity ratio of two alleles (A and B). The allele A is the reference allele whereas the allele B is the non-reference allele. The BAF value of 1 and 0 corresponds to the absence of one allele, i.e., BB and AA consecutively, and the BAF value of 0.5 corresponds to the presence of both alleles, AB. CaSpER first generates an expression signal by quantifying the expression values of all the genes from aligned RNA-seq reads. The expression values for the genes are treated as a genome-wide signal profile. In order to eliminate the noise in the initial expression signal profile, CaSpER performs sliding window-based median filtering and computes the $N$-level multiscale decomposition of the expression signal in multiple window length scales, where $N$ denotes the number of smoothing scales. The window length is increased between consecutive scales so that the higher scales correspond to a more extensively smoothed signal compared to smaller scales. Next, for the smoothed expression signal at each scale, a 5-state Hidden Markov Model (HMM) is used to assign copy number states to regions and segment the signal into regions of similar copy number states. The states correspond to the CNV states; 1: homozygous deletion, 2: heterozygous deletion, 3: neutral, 4: one-copy-amplification, 5: high-copy-amplification. The emission probabilities of the HMM states are initialized by estimating them from the data. The HMM is used to segment and assign the CNV states to the $N$ smoothed expression signal profiles. The basic motivation for using a 5-state HMM stems from how expression signal is interpreted and how the B-allele shift signal is integrated with the HMM states: states 0 and 5 represent deletion and amplification events that show very high evidence in terms of expression signal, i.e., very low or very high expression levels. We do not require an accompanying BAF shift signal for these states and we assign them deletion and amplification calls, respectively. The states 2 and 4 represent heterozygous deletion and one-copy-amplification states that require an accompanying BAF shift to be assigned final deletion and amplification calls. This is convenient

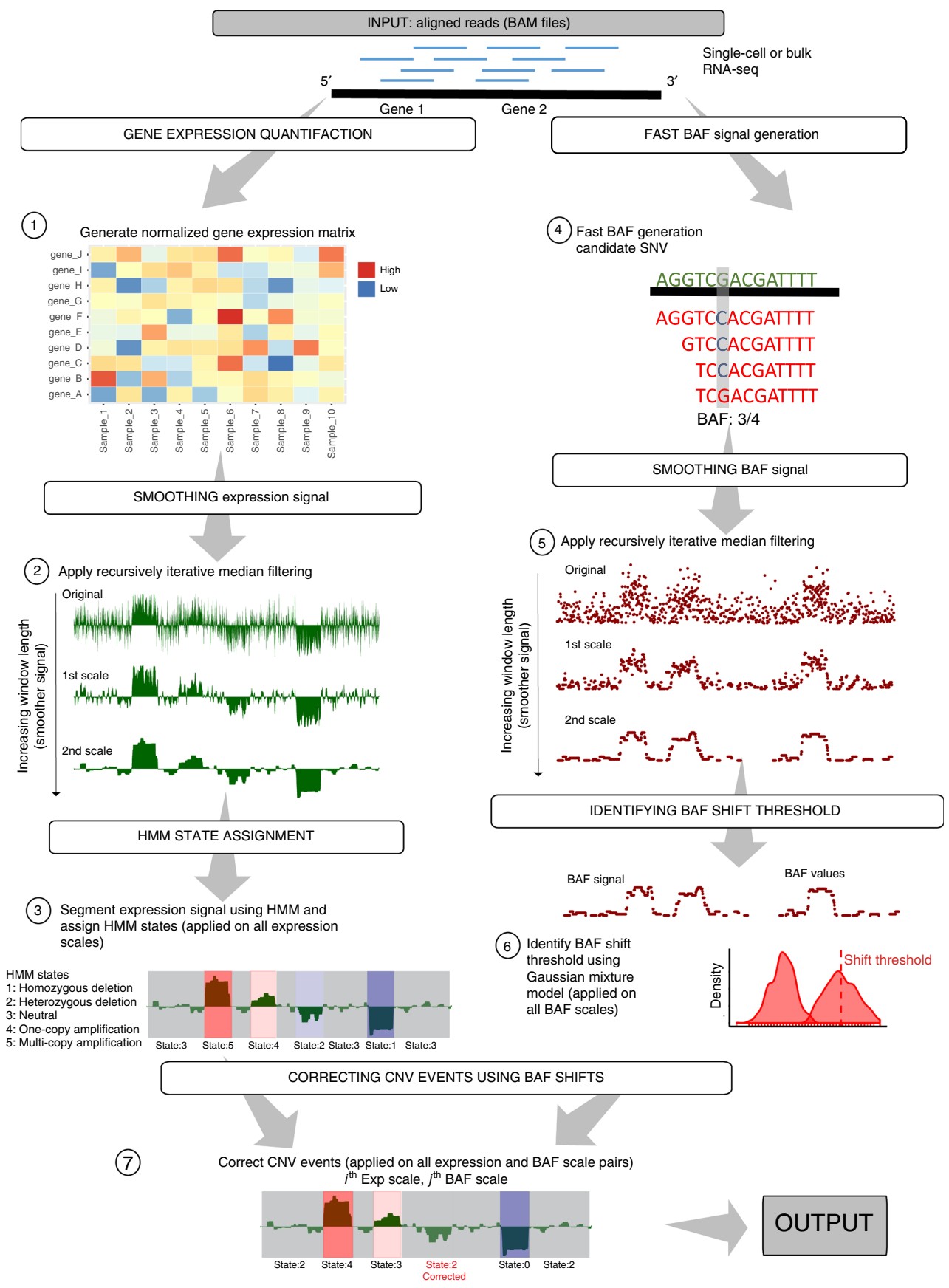

**Fig. 1 Flowchart of CaSpER algorithm.** The CaSpER algorithm uses expression values and B-allele frequencies (BAF) from RNA-seq reads to estimate CNV events. A normalized gene expression matrix is generated (Step 1). Expression signal is smoothed by applying recursive iterative median filtering. Three-scale resolution of the expression signal is computed. (Step 2). For the smoothed signal at each scale, HMM is used to assign CNV states to regions and segment the signal into regions of similar copy number states (Step 3). Five CNV states are used in HMM model; 1: homozygous deletion, 2: heterozygous deletion, 3: neutral, 4: one-copy amplification, 5: multi-copy amplification. BAF information incorporated into the segmented CNV events. BAF information is extracted from mapped RNA-seq reads using an optimized BAF generation algorithm (Step 4). BAF signal is smoothed by applying recursive iterative median filtering. Three-scale resolution of the allele-based frequency signal is computed (Step 5). BAF shift threshold is estimated using a Gaussian mixture (Step 6). CNV events are corrected using BAF shifts and final CNV correction is applied to all the CNV and BAF scale pair combinations (Step 7).

because states 2 and 4 generally are associated with not-so-strong expression signal evidence to be labeled with deletion and amplification calls. Therefore, they must be corroborated with the BAF shift signals since the heterozygous deletion and one-copy-amplifications must be accompanied by allelic shifts. The HMM segmentation yields $N$ segmentations corresponding to the $N$ smoothings of the expression signal.

After the assignment of HMM states, CaSpER integrates the BAF shift signal with the assigned states to generate the final CNV calls. For this, CaSpER first generates the BAF shift signal. BAF shift signal is extracted directly from the mapped RNA-seq reads using an optimized BAF generation algorithm (see Methods). Unlike other methods, BAF generation does not rely on an existing set of variant calls and this considerably speeds up the process of estimating the BAF signal. After the BAF shift signal is generated, it is smoothed using $M$-level multiscale decomposition. CaSpER smooths BAF shift signal at $M$ increasing window lengths where the window length is increased between consecutive scales. The next step is the assignment of the CNV calls by integrating the $M$ smoothed BAF shift signal with the CNV states assigned to the $N$ sets of HMM segments of the expression signals. The segments detected in each smoothed expression signal is compared with all the $M$ smoothed BAF shift signal profiles.

Given the CNV segments from expression scale $n$ and the smoothed BAF shift signal $m$, any segment with HMM state 1 and HMM state 5 is assigned deletion and amplification call, respectively. Any segment with HMM state 2 (heterozygous deletion) or HMM state 4 (one-copy amplification) is assigned deletion or amplification calls, respectively, if there is an accompanying BAF shift on the segment. BAF shifts are detected by thresholding the smoothed BAF signal profile. The BAF shift threshold is estimated by pooling BAF information across all samples and fitting a Gaussian mixture model (GMM) on the distribution of the smoothed BAF values within segmented regions. For each segment in expression scale $n$, CaSpER assigns $M$ CNV calls using the BAF shift signal. The final CNV event calls for all the pairwise combinations of expression and BAF shift signals are stored as the output from CaSpER's CNV calling steps (Fig. 1). Finally, CaSpER harmonizes the CNV calls on all scales and identifies the most commonly observed CNV call among all pairwise comparisons of expression and BAF shift. These harmonized calls are used to assign the final CNV calls at the large-scale (chromosome arms level), at the gene level, and at the segment level.

CaSpER outputs the CNV assignments to all the segments for all expression and BAF scale pairs. Moreover, it outputs the large-scale, gene-level, and segment-level CNV calls. For single-cell RNA-seq datasets, CaSpER infers CNV-based clonal evolution depicted as a phylogenetic tree and summarizes mutually exclusive and co-occurring CNV events using graph-based visualization (Fig. 2).

**Accuracy of CNV events detected from bulk RNA-seq.** We validated CaSpER algorithm on two datasets from the TCGA

Project[25], namely TCGA-GBM (160 samples), TCGA-BRCA (150 samples), and another publicly available meningioma cancer study (17 samples), where both bulk RNA-seq and genotyping data are available[26,27]. We explain below the outputs and accuracy of CaSpER on these datasets.

We first tested CaSpER on the meningioma dataset[26]. Meningioma tumors have relatively stable genomes and represent simpler cases for testing CNV calling algorithms compared to many other cancer types. We first quantified the expression values of all the genes across all meningioma samples. After quantification, the expression signal is smoothed. Heatmap of smoothed data clearly shows chromosome arm size deletion events (Fig. 3a). HMM is applied to the smoothed signal at each scale to assign CNV states to segmented regions. Concurrently, the BAF signal is calculated from aligned RNA-seq reads using the optimized BAF generation method. Similar to expression signal, the BAF signal is also smoothed using recursive median filtering (Fig. 3b). Smoothed BAF signal shows accompanying shifts in chromosomes with deletion events (Fig. 3b). For each scale, the GMM is fitted to the BAF values and this identified two sets of events. The first set of copy number events contains no BAF shifts whereas the second set contains BAF shifts. (Fig. 3c). We observed negative correlation between BAF and expression values at chromosomes that are recurrently deleted (chr1p, $r^2 = -0.43$, $P = 0.0008$; chr6q $r^2 = -0.54$, $P = 0.0003$; chr22q $r^2 = -0.14$, $P = 0.43$) (Fig. 3d). We validated the accuracy of CaSpER using an existing genotyping array for the same dataset as the gold standard. After harmonizing CNV calls for the segments from all scales, we summarized the CNV calls into large-scale CNV calls, where the large-scale event is defined as an event that impacted more than 1/3 of the chromosome arm (see Methods). We finally computed the true-positive rate (TPR) and false-positive rate (FPR) of the large-scale CNV calls using the genotyping array as an independent gold standard (see Methods). CaSpER achieves 95% TPR and 0.3% FPR for detecting large-scale deletion events (Fig. 3e, Supplementary Fig. 1) and 69% TPR and 0.3% FPR for gene-based deletion events. This test case indicates that CaSpER is potentially effective for detecting large-scale events in tumor genomes with low complexity.

We next focused on more complex tumors starting with TCGA-GBM dataset. For TCGA-GBM dataset, expression values of all the genes are quantified across all samples. The recursive median filtering effectively removes the fluctuations in the genome-wide expression introduced by noise and fluctuations in expression (Fig. 4a). For the smoothed signal at each scale, we applied HMM to assign CNV states to segmented regions. Simultaneously, the BAF signal is extracted from RNA-seq bam files using the BAF generation method and is smoothed using recursive median filtering. BAF shift threshold is estimated by fitting the GMM. GMM identified three classes of BAF shift groups where the first group corresponds to no shift regions whereas the second and the third groups correspond to BAF shift regions with loss or amplification events (Supplementary Fig. 2). We also investigated the correlation of expression values and BAF values in recurrently amplified chromosome 7q arm and deleted

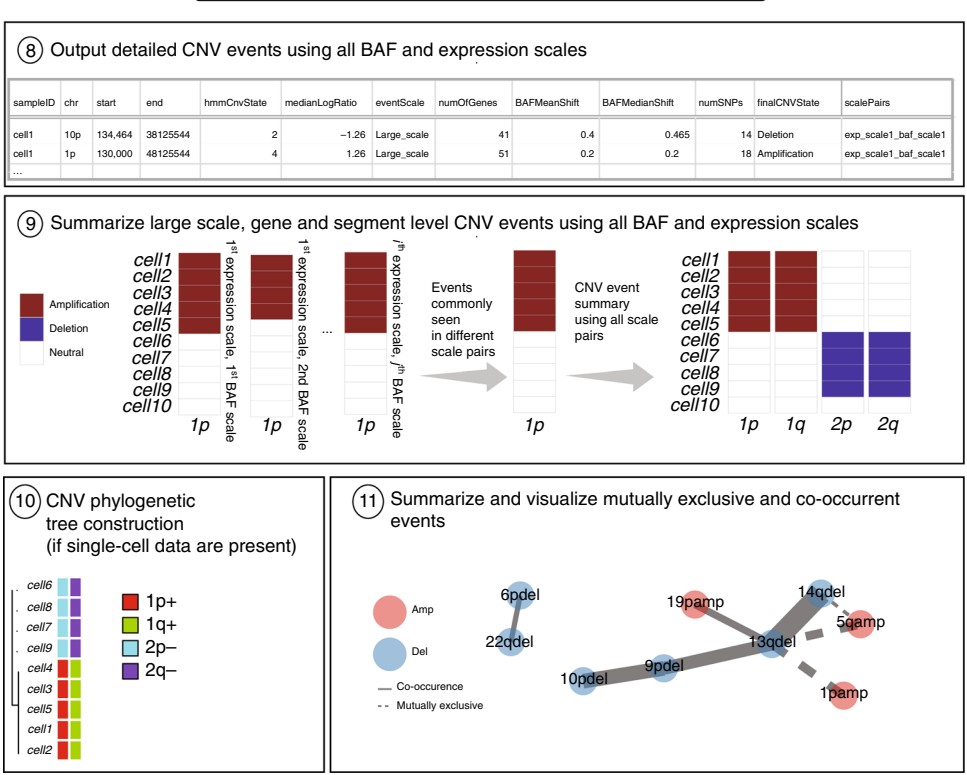

**Fig. 2 The output of CaSpER algorithm.** CaSpER algorithm outputs detailed CNV event calls for all CNV and BAF scale pairs (Step 8). Large-scale, gene- and segment-based CNV events that are commonly seen in all scale pairs are reported (Step 9). For single-cell RNA sequencing studies, CaSpER infers CNV-based clonal evolution depicted as a phylogenetic tree (Step 10). Mutually exclusive and co-occurring CNV events are summarized using graph-based visualization (Step 11).

chromosome 10q arm. We observed a significant correlation between BAF and expression values (chr7p, $r^2 = 0.17$, $P = 0.0004$; chr10q, $r^2 = −0.60$, $P < 2.2E^{−16}$) (Fig. 4b). Regions with CNV states 2 (heterozygous deletion) and 4 (one-copy gain) that are below the BAF shift threshold are corrected to be neutral (Fig. 4b).

We next used the CNV calls identified from genotyping arrays to measure the accuracy of CaSpER in large-scale, gene-, and segment-level CNV calls (see Methods). For each sample, we identified the large-scale deletions and amplifications from a genotyping array. We next calculated the accuracy of the CNV calls. We used the genotyping array ($n = 160$) as the gold standard and computed the appropriate accuracy metrics for the CNV calls that are harmonized at the large scale, at the gene level, and at the segment level. CaSpER achieves 79.5% TPR and 2% FPR for detecting large-scale amplification events, and 88% TPR and 3.5% FPR for detecting large-scale deletion events (Fig. 4c, d). At the gene level, CaSpER achieves 77% TPR and 3.6% FPR for detecting gene-level deletion events, and 61.6% TPR and 3.4% FPR for detecting gene-level amplification events (Supplementary Fig. 3). For the segment-level events, CaSpER achieves 70% positive predictive value (PPV) and 76% TPR for deletion events, and 68.5% PPV and 66% TPR for amplification events Supplementary Figs. 4 and 5). We also evaluated the accuracy of focal segments at different length thresholds (Supplementary Note 1, Supplementary Fig. 6).

While interpreting the accuracy results, it is important to keep in mind that there is a certain portion of the genome that is not accessible for CNV detection by RNA-seq datasets (Supplementary Figs. 7, 8, Supplementary Note 2). This is because RNA-seq

signal is mainly concentrated on the exons of genes. In fact, the smallest resolution that we can detect CNVs is at the gene boundaries. An important implication of this is that CaSpER and other RNA-seq-based CNV calling methods cannot identify CNVs that are primarily in the intergenic space. Thus, these factors may impact the TPR estimates adversely.

We also focused on breast cancer as a representative of a more complex tumor type. We randomly selected 150 samples from the TGCA breast cancer cohort, which contains around 1000 breast cancer RNA-seq samples. After the processing steps similar to the ones that were detailed in previous datasets, CaSpER achieves 60.3% TPR and 3.8% FPR for detecting large-scale amplification events, and 79% TPR and 7.7% FPR for detecting large-scale deletion events (Supplementary Fig. 9). CaSpER achieves 62.8% TPR and 6% FPR for detecting gene-level deletion events, and 56.6% TPR and 7% FPR for detecting gene-level amplification events (Supplementary Fig. 10).

**Inference of CNV architecture in scRNA-seq.** We next used CaSpER to infer subclonal CNV architecture from single-cell Glioblastoma Multiforme (GBM) RNA-seq data[6]. Single-cell GBM RNA-seq data contain 430 single cells extracted from five patient samples. The smoothed expression signal in Fig. 5a shows that chromosome 7 amplification and chromosome 10 deletion is recurrent across GBM samples. We observed that the BAF shift signal is much more stable and accurate when the data from all the cells are pooled. Therefore, for the BAF shift signal generation step, we pooled the single-cell reads from each patient separately. We then extracted the BAF shift signal from the pooled patient-

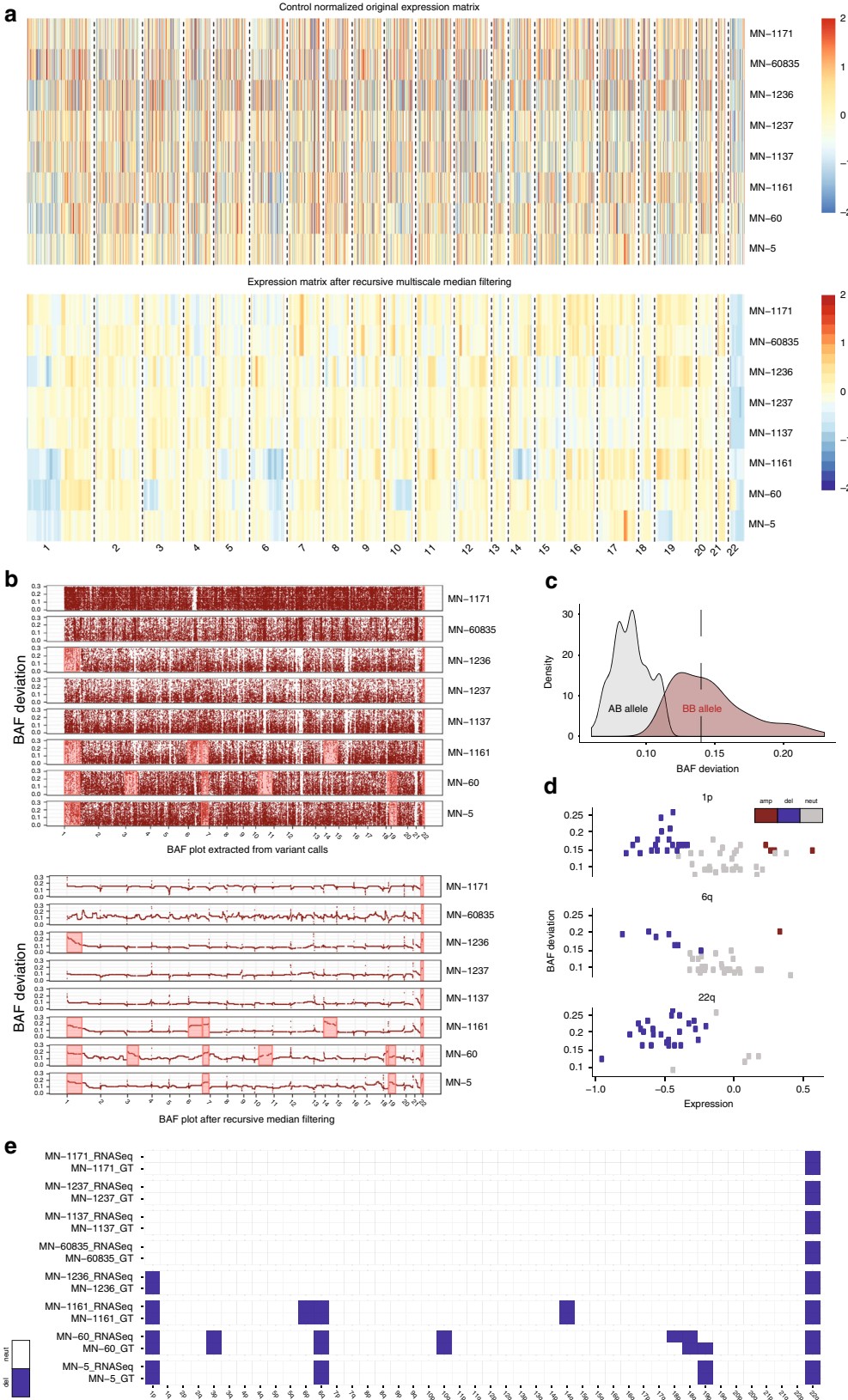

specific reads instead of single-cell reads since the BAF shift signal extracted from one single cell is very sparse and is not informative. The smoothed patient-specific BAF shift signal shows clear shifts in chromosomes 7 and 10 (Fig. 5b). We next used the large-scale CNV calls to identify the co-occurring and mutually exclusive events in all scale pairs (Fig. 5c). Interestingly, MGH31 consists of two mutually exclusive subclones where one subclone contains chromosome 5q amplification whereas the other sub-clone contains chromosome 14q deletion (Fig. 5d, e). Additionally, one subclone contains 1p amplification and the other

**Fig. 3 CaSpER algorithm applied to bulk meningioma RNA-seq dataset. a** Heatmap of normalized expression values of all the genes across all samples ($n = 8$ samples) is shown in the top panel. The smoothed expression signal by recursive iterative median filtering is shown on the bottom panel. **b** Original and smoothed BAF signal is plotted and shows shifts in chromosomes with deletion events. **c** BAF shift threshold is estimated by fitting Gaussian mixture model (GMM). GMM identified two classes of BAF shift groups where the first group corresponds to no shift regions whereas the second group corresponds to BAF shift regions with loss or amplification events. BAF shift threshold is the median of the BAF values in the second group, which is calculated to be 0.14. **d** The correlation of expression values and BAF values in recurrently deleted chromosomes are plotted (chr1p, $r^2 = -0.43$, $P = 0.0008$; chr6q $r^2 = -0.54$, $P = 0.0003$; chr22q $r^2 = -0.14$, $P = 0.43$). Each dot represents one CNV segment. **e** Heatmap of large-scale CNV events identified from RNA-seq and genotyping is shown in the plot. The color codes are explained on the right.

subclone contains 13q deletion, which has not been reported previously. For MGH31, clustering of large-scale CNV calls stratified the cells harboring 1p and 5q amplification from cells harboring 13q and 14q deletion (Fig. 5e). A small number of cells from patient MGH31 was shown in the original publication to be normal cells (oligodendrocytes) (Supplementary Fig. 11). These normal cells are clustered separately with a low CNV profile using large-scale event summary generated by CaSpER (Fig. 5e).

CaSpER also reports the mutually exclusive and co-occurring CNV events and plots the significant events as a graph. This is useful for visually inspecting the co-occurring and mutually exclusive events that may be otherwise hard to visualize. Similarly, from the graph, we can clearly see the mutually exclusive 1p:13q and 5q:13q, 5q:14q event pairs for patient MGH31 (Fig. 5f). Moreover, we detected mutually exclusive 8q:20p, 5q:19p event pairs for patient MGH28, 6p:7p event pair for patient MGH30 that was not reported in the previous publications (Supplementary Figs. 12–14). We next identified the gene expression signatures of each of the mutually exclusive clones and performed GO enrichment analysis (Supplementary Note 3).

**Performance of CaSpER on CNVs of varying size and clonality.** We observed a slight difference in deletion TPR rates, between meningioma, TCGA-GBM, and TCGA-BRCA datasets. This potentially stems from the fact that meningioma tumors exhibit less intratumor heterogeneity and have lower clonality rates compared to GBM and breast cancer tumors. Thus, lower clonality rates in meningiomas lead to better deletion CNV event identification. Similarly, high clonality rates in GBM and breast cancer tumors lower the detection accuracy of low-level amplification events, which then lead to a low amplification TPR rate. We assessed the performance of CaSpER on CNVs of varying size and clonality using simulated gene expression data (see Methods, Supplementary Fig. 15, Supplementary Note 4).

**Accuracy of CNV events detected from scRNA-seq.** An important benchmark is the accuracy evaluation of CNV detection from scRNA-seq data. However, this requires known CNVs for the cells with matching scRNA-seq data. We have been challenged for finding ground truth data for CNVs in single cells matching to the scRNA-seq experiments. This is because the methods for simultaneous measurement of scRNA-seq and single-cell DNA sequencing (or copy number profiling) are still very much under development and they do not provide high number of cells where CNVs and RNA-seq being probed simultaneously[28,29]. We used the data from the original DR-Seq[29] study, where the simultaneous RNA and copy number measurements are publicly available for seven cells from Breast cancer cells with many amplification and deletion events.

In the DR-Seq study, the authors used a computational approach and assigned absolute copy numbers to segments (Supplementary Note 5). We first evaluated how the copy number states from CaSpER correlate with the copy number assigned to each gene (Supplementary Fig. 16). We show below, for each cell,

the distribution of the DR-Seq assigned copy numbers corresponding to the HMM copy number states. We observed a high consistency between the HMM copy number state and the DR-Seq assigned copy number within each cell. Higher copy number states in CaSpER correspond to the amplification and lower copy number states correspond well to the deletions (Supplementary Fig. 16).

We first processed the absolute copy numbers from the DR-Seq study to build the relative copy numbers (i.e., amplification, deletion, neutral) in each cell (see Methods). We evaluated the TPR and FPR for the detected large-scale deletion and amplification events. CaSpER achieves 32% TPR and 1.6% FPR in deletion events and 49% TPR and 3.8% FPR in amplification events. If we relax the parameters (with $\gamma = 1$), CaSpER achieves 45% TPR and 3% FPR in deletion events and 62.6% TPR and 8.6% FPR in amplification events.

**Comparing the performance of CaSpER with existing tools.** We also compared the performance of CaSpER with HoneyBADGER on bulk and scRNA-seq datasets. We first used the data published in the DR-Seq[29] study, where the simultaneous RNA and copy number measurements are publicly available for seven cells from Breast cancer cells. HoneyBADGER could not detect any CNV events even though the ground truth data have many amplification and deletion events. We next used bulk TCGA-GBM and TCGA-BRCA RNA-seq datasets using genotyping arrays as the gold standard (Supplementary Figs. 17–20). We show that HoneyBADGER is very specific and has lower sensitivity in bulk RNA-seq datasets since it is designed primarily for scRNA-seq datasets (Supplementary Tables 1–2). InferCNV is primarily for visualization of the copy number events and does not provide a set of CNV events. From these comparisons, CaSpER stands out as a tool that is broadly applicable to detection of CNVs from single cell and bulk RNA-seq datasets and can be used for visualization and downstream analysis of the detected CNVs.

**Identification of scale-specific CNV regions.** CaSpER identifies CNV regions at different expression scales, which yields scale-specific CNV (SSCNV) regions. In general, SSCNVs at lower scales correspond to more focal CNV events compared to SSCNVs at higher scale, which represent broader CNV events. Figure 6 shows the scale length characteristics of different CNV events of TCGA bulk RNA-seq data. Focal amplification of the PDGFRA gene is identified using small-scale lengths whereas broad chromosome arm-level deletion in chromosome 22 is identified using a higher scale length (Fig. 6a–b).

**Discussion**
We presented an algorithm, CaSpER, for identification, visualization, and integrative analysis of focal and large-scale CNV events in multiscale resolution using either bulk or scRNA-seq data. We demonstrated that CaSpER performs well in identifying CNV events using both single-cell and bulk RNA-seq data. We presented several examples where CaSpER can effectively complement the existing set of RNA-seq analysis tools and can give

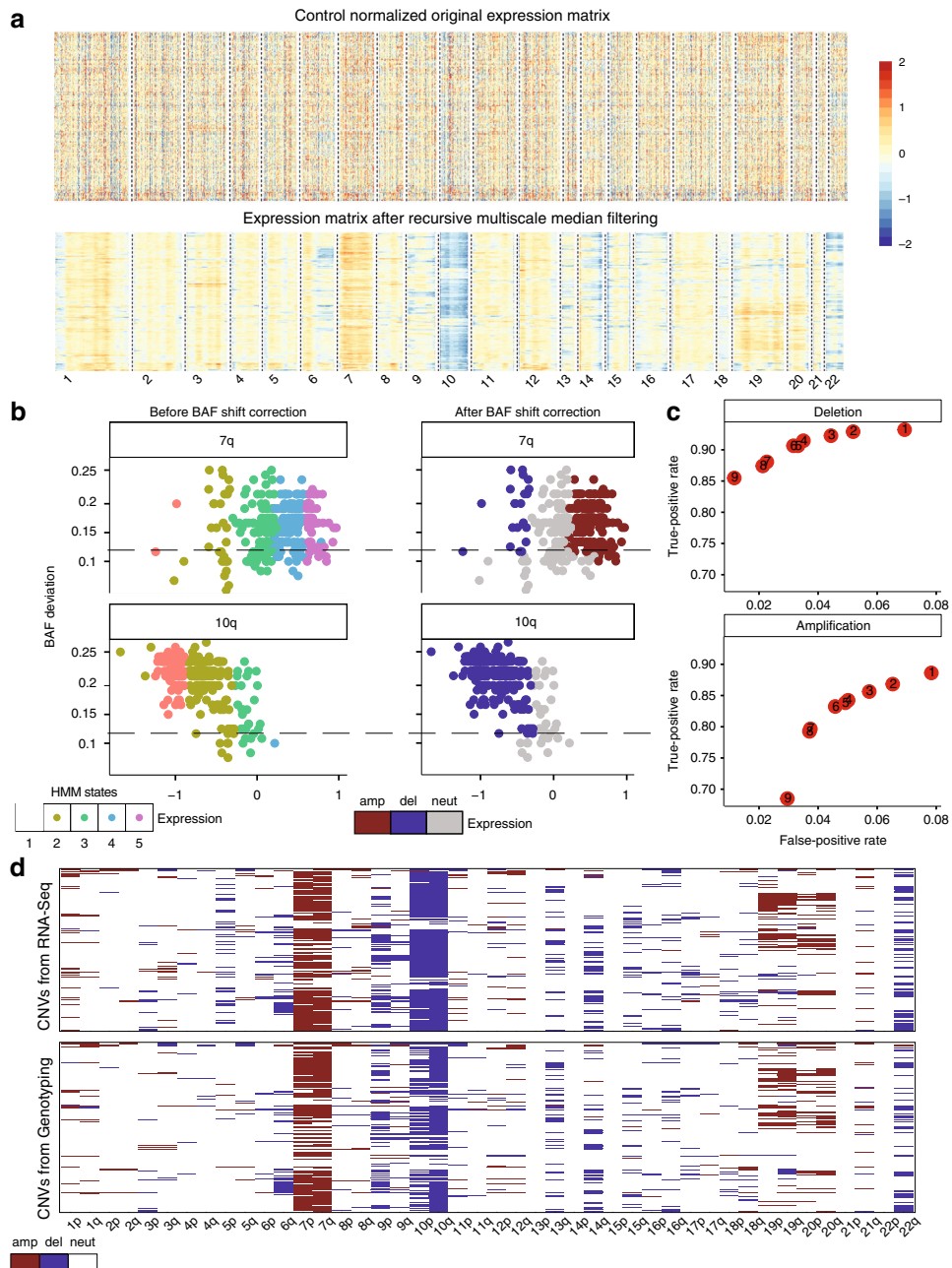

**Fig. 4 CaSpER algorithm applied to bulk TCGA-GBM RNA-seq dataset. a** Heatmap of normalized expression values of all the genes across all TCGA-GBM samples (~170 samples) is shown in the top panel. The smoothed expression signal by recursive iterative median filtering is shown in the bottom panel. **b** The correlation of expression values and BAF values in recurrently amplified chromosome 7q arm and deleted chromosome 10q arm is plotted (chr7p, $r^2 = 0.16$, $P = 0.0004$; chr10q, $r^2 = -0.60$, $P < 2.2E^{-16}$). Regions with CNV states 2 and 4 that are below the BAF shift threshold (0.12) are corrected to be neutral. The color codes are explained on the bottom. *X*-axis is the expression value and *Y*-axis is the BAF deviation. Each dot represents one CNV segment. **c** TPR and FPR values for bulk TCGA-GBM RNA-seq data with varying $\gamma$ thresholds. The plot shows the large-scale event performance, which is assessed using a genotyping array. Labels in red points represent the $\gamma$ threshold. *X*-axis is the FPR value and *Y*-axis is the TPR value. **d** Heatmap of large-scale CNV events identified from RNA-seq is shown in the top panel whereas the heatmap of large-scale CNV events identified from genotyping is shown in the bottom panel. The color codes are explained on the bottom.

insight into the analysis of the clonal architecture of cancer genomics datasets. CaSpER can extend the utility of RNA-seq datasets beyond just transcriptional profiling.

There are several aspects of CaSpER that we would like to point out. First, CaSpER combines genomewide allelic shift signal, which measures the LOH at a nucleotide resolution, and expression signal to accurately estimate CNV events. While doing this, CaSpER generates allelic shift signal profile directly from

mapped reads, without the need for variant calls. This is very useful for two reasons: First, the variant calling (SNVs and indels) from mapped reads requires high computational resources and long compute times. This means that before calling CNVs, it is necessary to complete the arduous process of variant calling. CaSpER lifts this restriction by computing the allele shift profile directly from the mapped reads. Our results show that we do not need a high-quality variant call set to detect the BAF shift signal.

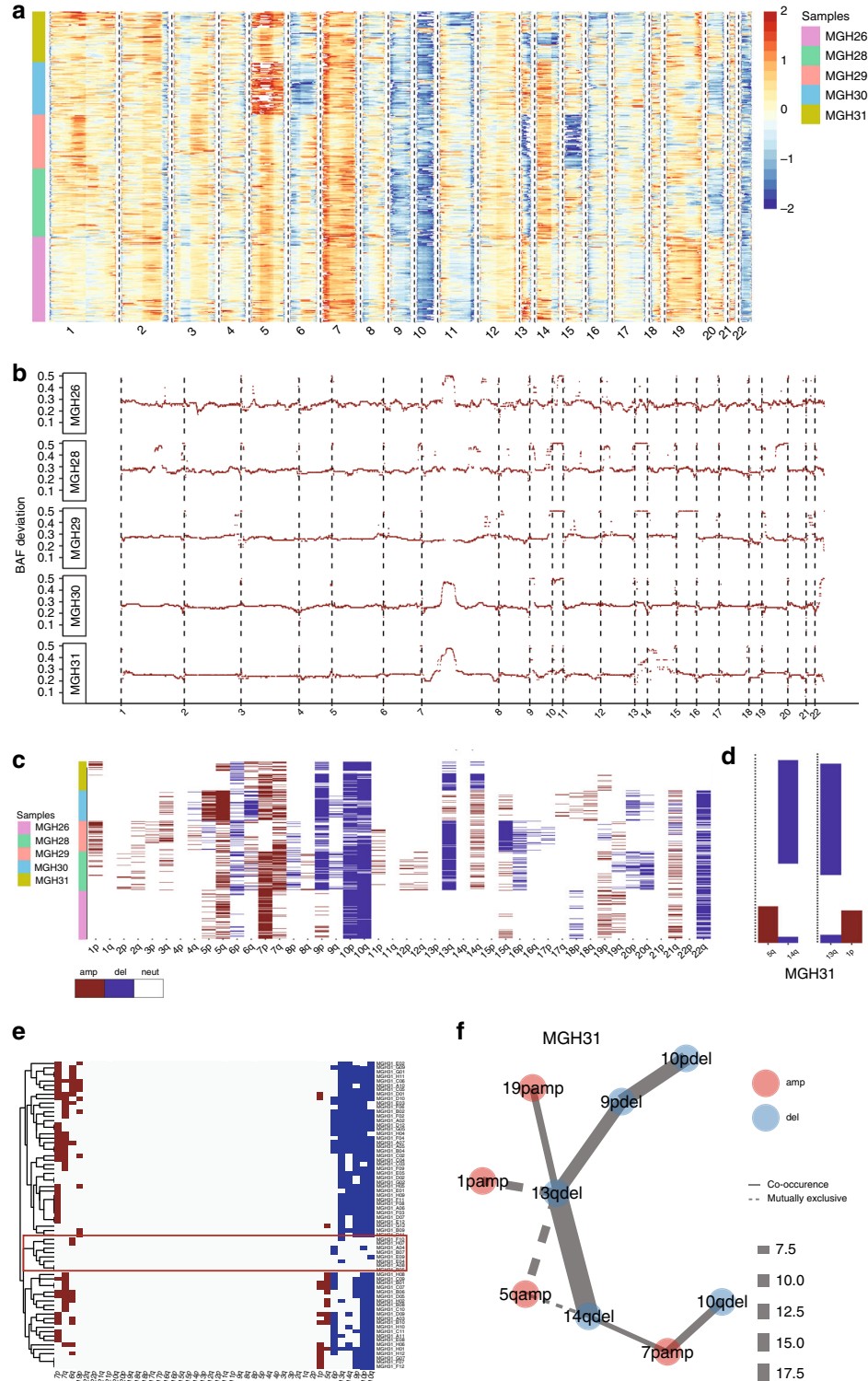

Unlike other tools that depend on each SNV to be of high quality, CaSpER aims to detect a collective shift in the allele frequencies of consecutive variants. To do this, CaSpER analyzes many potential variants at the same time. The multiscale smoothing of the BAF shift signal of these variants is vital in this process because it enables CaSpER to clear the noise from the collective shifts in the allele fractions of these variants. Second, the power of variant detection can be affected by CNV events. This is especially important in cancer sequencing experiments where CNVs can span very long genomic regions and may affect the accuracy of variant calling. Therefore, identifying CNVs before calling SNVs can give very useful information for correct identification of SNVs. Although we did not explore this thoroughly in this paper, several previous studies have demonstrated this[30].

Another aspect of CaSpER is the analysis of CNV events at multiscale resolution. With the diverse length characteristics of CNV events, we believe that it is very important to be able to analyze CNV events in multiple length scales. CaSpER makes

**Fig. 5 CaSpER algorithm applied to single-cell GBM RNA-seq dataset. a** Heatmap of smoothed expression signal of all the genes across all samples is shown in the top panel. The color codes are explained on the right. **b** Smoothed BAF signal from the pooled patient-specific reads is shown in the plot. The smoothed patient-specific BAF signal shows shifts in deleted and amplified chromosomes. **c** The heatmap of summarized large-scale CNV events using the common events in all scale pairs is plotted. Columns represent chromosome arms whereas rows represent cells. The color codes represent the patient id. **d** MGH31 consists of two mutually exclusive sub-clones where one sub-clone contains chromosome 5q amplification whereas the other sub-clone contains chromosome 14q deletion. Additionally, one sub-clone contains 1p amplification and the other sub-clone contains 13q deletion. **e** Clustering of large-scale events generated by CaSpER in patient MGH31. Normal cells are clustered separately with a different CNV profile. Cells within the red rectangle correspond to normal cells. Rows correspond to cells whereas columns correspond to chromosome arms. Clustering separates cells harboring 1p and 5q amplification from cells harboring 13q and 14q deletion. **f** Mutually exclusive and co-occurring CNV events are plotted as a graph. Red colored events are amplified whereas blue colored events are deleted. The solid lines represent co-occurring events, whereas dashed lines represent mutually exclusive events. Edge width increases with event significance, which is assessed using Fisher's Exact test $p$-value (edge width $= -\log2(p$-value)). The mutually exclusive 1p:13q and 5q:13q, 5q:14q event pairs for patient MGH31 is significant.

available the multiscale smoothed genomewide expression signal and allelic shift signal profiles and the CNV calls that can be used for downstream analysis and visualization. For smoothing, CaSpER utilizes a non-linear median-based filtering of RNA-seq expression and allele-frequency signal. The median filtering preserves the edges of the signal much better compared to the kernel-based linear filters[24]. We also demonstrated that the signal profiles that are smoothed at multiple scales are useful for visualization of the copy number events that are detectable from the RNA-seq datasets. In addition to identifying CNV events, CaSpER also visualizes and performs integrative analysis of CNV events such as inferring clonal evolution, discovering mutual-exclusive and co-occurring CNV events and identifying gene expression signatures of the identified clones. The mutual exclusivity and co-occurrence of variants have been used by many studies to study bulk sequencing of tumor samples with respect to different genetic backgrounds and processes[31,32]. For example, CBioPortal website is a popular website dedicated to analyzing large cancer cohorts for analysis of co-occurrence and exclusivity patterns in somatic variants among multiple tumor samples by generating OncoPrint plots[33]. The application of mutual exclusivity and co-occurrence in the context of single-cell RNA and DNA sequencing has not been studied extensively but it has been analyzed by different studies. We believe CaSpER fills an important gap by providing these plots so that single-cell experiments can be evaluated in terms of co-occurrence and exclusivity patterns.

There are many technical factors that may impact CaSpER's performance. The data normalization is one of the first important factors. Interestingly, CaSpER does not have a specific requirement for the inter-sample distribution of the gene expression levels, e.g. quantile normalization. This is because multiscale decomposition works on a sample by sample basis and does not expect expression levels to be normalized across samples. Secondly, the BAF shift signal profiles are generated for each sample independently for bulk sequencing, i.e., the BAF shift is computed in terms of the fraction of reads that support the B-alleles. Thus, the BAF shift signal normalizes itself at each SNV location and does not require a normalization across samples. One issue is that for single-cell datasets, the BAF shift signal is computed initially by pooling the reads from all the cells of a sample. If some cells have a substantially large number of reads, they may dominate the BAF shift signal and may potentially create biases. As long as each cell contributes a similar number of reads to the BAF shift generation, this should not be a major biasing factor.

Another major factor is the underlying sequencing technology that generated the data. Especially the scRNA-seq technologies can be classified broadly into two groups. First is full-transcript sequencing (SMART-Seq[34]) and second is 3′-only sequencing (such as DropSeq[35]). The data used in the analyses are from full transcript sequencing technologies. The data from 3′-only

technologies are biased such that the reads are enriched along the 3′ end of the genes compared to the 5′ ends of the genes. While 3′-based reads can be used to generate expression levels that are of fair quality[36], the BAF shift signal may be adversely impacted because BAF shift will be measured for only the variants that are located close to the 3′ ends of the genes. We, therefore, expect that the BAF shift signal will be much sparser when data from these technologies are used (Supplementary Figs. 21 and 22, Supplementary Note 6). As CaSpER pools all the reads from all the cells while generating the BAF shift signal. Therefore, we believe that the pooling will soothe the adverse effects of the 3′ bias as long as a decent number of reads are accumulated along with the transcripts. We assessed the suitability of CaSpER 3′ transcript sequencing data using the multiple myeloma 10× RNA-seq data (MM135) ($n = 947$ single cells) presented in HONEYBADGER study. Even though the exact ground truth CNV calls are unknown for this study, we show the concordance of CaSpER CNV calls with HONEYBADGER (Supplementary Figs. 23–25, Supplementary Notes 7 and 8).

Having noted these, we believe that CaSpER can benefit a model for BAF shift generation step where it takes into account the 3′ bias of the sample preparation assays. It should also be noted that factors such as RNA editing[37], PCR duplication artefacts[38], and sequencing errors[39] may impact and bias detection of CNVs as they may impact the BAF shift signal in a complex manner. Among these, RNA editing is specific to the detection of CNVs from RNA-seq datasets and must be considered carefully while analyzing RNA-seq data from the tumor (Supplementary Note 9).

In our analyses, CaSpER discovered ssCNV regions in TCGA bulk RNA-seq data, which represent both focal and broad CNV events, and we showed the utility of these events in the analysis of scale-specific co-occurrence and mutual exclusivity of the CNV events. Analyzing single-cell GBM RNA-seq data using CaSpER, unraveled mutual exclusive and co-occurring CNV subclones. Gene expression signatures of the identified clones gave us insight into the phenotype of the clones such as invasiveness and survival. Moreover, we identified potential therapeutic targets for the clones. In conclusion, our study demonstrates the significance and feasibility of CNV calling using either single or bulk RNA-seq data.

## Methods

**Bulk and scRNA-seq expression quantification**. Yale meningioma bulk RNA-seq data reads were aligned using STAR[40]. Expression-level quantification was performed using DESeq2 R package[41]. TCGA-GBM and TCGA-BRCA bulk RNA-seq normalized expression matrix were downloaded using TCGABiolinks R package[42]. The corresponding bam files were downloaded from GDC data portal[43].

Single-cell GBM RNA-seq data reads were aligned with Hisat2 using ENCODE V28 transcriptome annotation[44]. We pooled single cells from the same patient into a single bam file using bamtools merge function[45]. The aligned bam files were later used for allele-based frequency signal calculation.

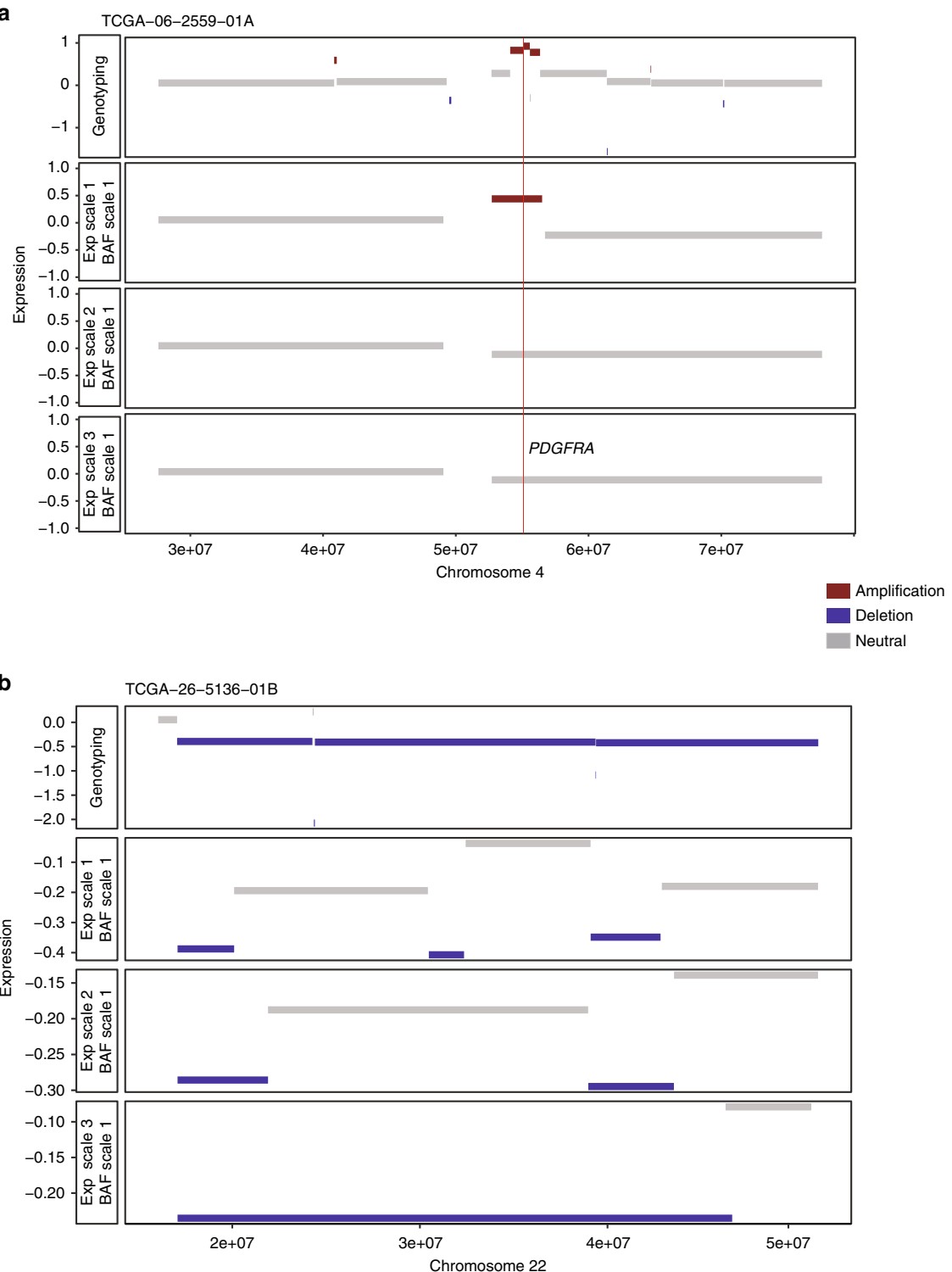

**Fig. 6 CaSpER identifies scale-specific CNV (SSCNV) regions. a** CNV segments identified from different expression scales and genotyping is plotted for *PDGFRA* region. *PDGFRA* focal amplification is identified using a lower scale length. **b** CNV segments identified from different expression scales and genotyping are plotted for chromosome 22. Broad chromosome arm-level deletion in chromosome 22 is identified using a higher scale length.

We used the normalized expression matrix provided in the paper both for GBM and for the DR-Seq study.

**CNV data processing for DR-Seq study**. In DR-Seq study, we processed the absolute copy numbers provided by the authors to build the relative copy numbers (i.e., amplification, deletion, neutral) in each cell. This is necessary because both CaSpER and HoneyBADGER assign relative copy numbers to the detected

segments where copy numbers are assigned with respect to the average DNA content in each cell as we discuss further below.

To generate the ground truth relative CNV calls for each cell, we first computed the ploidy in each cell using the following:

$$\text{ploidy}_i = \sum_j \frac{((\text{Absolute CN of segment } j \text{ in cell } i) \times (\text{Length of segment } j \text{ in cell } i))}{\sum_k (\text{Length of segment } k \text{ in cell } i)},$$

(1)

where ploidy$_i$ indicates the average DNA content in cell $i$. We next assigned amplification and deletion to each cell using ploidy$_i$:

$$\text{CNV}_j = \begin{cases} \text{Amplification, if (absolute CN of segment } j) > \text{ploidy}_i \\ \text{Deletion, if (absolute CN of segment } j) < \text{ploidy}_i \\ \text{Neutral, if (absolute CN of Segment } j) = \text{ploidy}_i \end{cases} \quad (2)$$

where CNV$_j$ denotes the relative copy number value assigned to segment $j$. We need to perform the relative copy number assignments because both CaSpER and HoneyBADGER assign copy numbers relative to the ploidy of the cell. In other words, these tools do not assign absolute copy numbers. Thus, we needed to convert the absolute copy numbers to relative copy numbers. We next used the relative copy numbers to assign the large-scale FPR and TPR to the CNV calls generated by CaSpER and HoneyBADGER using the scRNA-seq data for seven cells.

**Gene expression data processing**. The inputs to the algorithm are the normalized read counts such as FPKM or TPM. Before multiscale decomposition of the signals, we perform two-step gene expression centering. First, gene centric expression centering is performed. the gene expression levels are centered around the mid-point. For each gene, the mid-point of expression level is computed among all the cells (or samples in bulk RNA-seq), then the mid-point expression level is subtracted from the expression levels. Next cell-centric expression centering is performed. For each cell (or sample), we compute the mid-point of the expression level, then we subtract the mid-point expression from the expression levels of all the genes for the corresponding cell. This is done to reduce the effect of complexity. After expression centering steps, the control normalization is performed by subtracting reference expression values from the tumor expression values.

**Generation of the allele-based frequency signal**. We generate allele-based frequency signals from RNA-seq bam files using our in-house written C++ code, which can be downloaded at https://github.com/akdess/BAFExtract. Our method takes a bam file as an input and outputs allelic content estimation through fast SNP calling. We first perform pileup where we summarize the base calls of aligned reads to a reference sequence. For each SNP, we report the total count of reads supporting non-reference and reference nucleotide after applying the following filters: (1) reads should have mapping quality of at least 50, (2) minimum number of total reads per each SNP position should be 20, (3) minimum number of total reads supporting SNP should be 4. In bulk tumor RNA-seq, we considered SNPs that are most likely to be heterozygous with a BAF value more than 0.2 and less than 0.8 whereas in scRNA-seq BAF value more than 0.2 are considered (refer to below for detailed motivation of the thresholds) (Supplementary Note 10, Supplementary Fig. 26). Our fast BAF generation method speeds up the process of estimating BAF shift regions compared to using GATK for calling variants from RNA-seq data. After generating the allelic content for each SNP, we next apply recursive median filtering to remove noise from the signal. As explained previously in the Results section, filtering the reads according to mapping quality is very critical for correctly estimating BAF shift regions. In addition to estimating BAF shift regions, our method is also very useful in identifying allele-specific expression. The BAFExtract method does not use deduplicated reads. This was necessary to ensure that we had enough reads to generate the BAF shift generation, especially for the scRNA-seq datasets.

**BAF generation, SNV filtering, and related thresholds**. The BAF shift signal for bulk and scRNA-seq data are affected differently by the existence of healthy cells differently in bulk and single-cell samples.

*Bulk RNA-seq SNV filtering and related thresholds*: In principle, the BAF shift in cancer sample must be computed using the SNVs in the matching normal tissue such as blood or tissue surrounding the tumor. This way, the BAF shift in tumor samples are computed with respect to exactly the normal tissue's baseline allele frequency:

$$\text{BAF}_{\text{shift}}(\text{SNV}_i) = |\text{BAF}_{\text{tumor}}(\text{SNV}_i) - \text{BAF}_{\text{normal}}(\text{SNV}_i)|, \quad (3)$$

where SNV$_i$ denotes the $i$th SNV and BAF$_{\text{tumor}}$(SNV$_i$) denotes the alternate allele frequency of SNV$_i$ for the tumor sample:

$$\text{BAF}_{\text{tumor}}(\text{SNV}_i) = \frac{\text{Number of tumor reads supporting alternate allele of SNV}_i}{\text{Number of all tumor reads at SNV}_i}. \quad (4)$$

This is the ratio of the number of RNA-seq reads that support the alternate allele divided by the number of total reads that cover the SNV's locus. For the SNVs that reside on a copy neutral LOH or a CNV, this value will show deviation from 0. Otherwise, it will be distributed around 0.

We cannot compute the above quantity directly because we do not have access to the normal RNA-seq dataset. For whole-exome and whole-genome sequencing of tumor, the normal tissue is sequenced regularly (almost as a standard protocol) but RNA-seq of matching normal tissue is currently not established as standard procedure. Since we do not have the matching normal sample, we focus on the SNV candidates that are most likely heterozygous (hence, BAF$_{\text{normal}}$(SNV$_i$) = 0.5)

in the healthy tissue. We also have to modify the definition of shift as:

$$\text{BAF}_{\text{shift}}(\text{SNV}_i) = |\text{BAF}_{\text{tumor}}(\text{SNV}_i) - 0.5|. \quad (5)$$

As it can be seen, without the normal sample, we just subtract 0.5 from the observed tumor SNV allele frequency (and take absolute value) to compute the BAF shift signal. The most vital component of this computation is to ensure that the SNVs we use are most likely heterozygous in the normal tissue.

The thresholds we select aim at ensuring that the selected variants are likely heterozygous variants in the normal tissue. If we do not apply these thresholds, we observed that the allele frequency shift signals are non-informative to detect the LOH and BAF shifts associated with CNVs. We describe the motivation for selecting these thresholds below:

For bulk samples, there is generally the high concentration of healthy cells in the tumor tissues (up to 20–30% as shown in the literature[46]). Therefore, if an SNV has a high observed allele frequency (>95%) in the tumor, it is very likely that it is a homozygous SNV in the healthy tissue too. The existence of normal tissue in tumor samples will cause a shift in the observed allele frequency of SNVs in the bulk tumor RNA-seq data.

Assume that there is a one-copy deletion of a segment and there is 30% normal tissue in the bulk RNA-seq sample: 70% of the cells are tumor cells and they contain only one copy of the segment; 30% of the cells are normal cells and they contain two copies. For an SNV on this segment who is heterozygous in the normal cells, the observed allele frequency will be:

$$\frac{70 \times 1 + 30 \times 1}{70 \times 1 + 30 \times 2} = 0.769. \quad (6)$$

In the above equation, if the impurity is smaller than 30%, the observed allele frequency increases. To be cautious, we choose to act conservatively, and we assume that there may be around 30% impurity in the tumor samples which is consisted of TCGA cohort (Supplementary Fig. 27). Thus, we assume that any SNV with allele frequency higher than around 80% is likely a homozygous variant. We, therefore, exclude the variants that have higher than 80% allele frequencies (found by rounding 0.769) and smaller than 20% allele frequencies (the other side of the AF spectrum) since these are likely homozygous events in the normal matching tissue (Supplementary Fig. 28). Note that by doing this filtering, we are likely removing variants that inform us about the existence of important LOH and CNV events, but we still remove these for being conservative and using only the SNVs that are likely heterozygous in the normal sample.

In the above computation, we assume for simplicity that there is a clonal deletion of the segment. If the deletion is subclonal (less than 100% of the tumor cells contain the deletion), the observed allele frequencies will be closer to 50% and they will still be selected in the thresholding process. Similarly, for uneven amplification events, the above ratio will decrease. Thus, for these events, the thresholding will capture the useful SNVs appropriately. In addition, for very high amplifications events, allele frequency will be close to 50% (i.e. shift close to 0). In these cases, CaSpER weighs the expression signal much more than the BAF shift signal. Thus, the thresholding affects these events less.

*scRNA-seq SNV filtering and related threshold*: SNV selection procedure for single-cell samples is motivated mainly by technical factors around the detection of the SNVs. Compared to the bulk RNA-seq samples, there is a much smaller number of reads from each cell. As described in the text, we tackle this issue by pooling the reads from all the cells and generate a bulk RNA-seq sample from all the cells. Even though we pool the samples, BAFExtract still detects a relatively small number of SNVs that we can use to compute the BAF shift signal.

In addition, for scRNA-seq samples, the impurity of RNA-seq samples, i.e. fraction of normal cells in the sample, is much smaller (compared to bulk tumor RNA-seq samples) since the normal cell infiltration can be controlled (using for example FACS) and it is generally much smaller. Thus, the scRNA-seq data affected much less from the impurity of the samples. Because the sample is almost pure, we cannot use the observed SNV allele frequencies directly to infer a maximum allele frequency threshold (similar to the threshold we derived in the above equation) for selecting SNVs that are most likely heterozygous in matching normal cells (Supplementary Fig. 29). We, therefore, include all the SNVs in the analysis except the SNVs whose alternate allele frequencies are smaller than 20%. The reasoning for this lower allele frequency threshold is that these SNVs may be manifestations of technical artifacts such as PCR duplications[47]. This threshold value is a conservative value that is selected by trial-and-error to optimize the quality of the observed BAF shift signal and its consistency with the expression signal. We illustrated the spectrum of allele frequencies of SNVs that are used for bulk and scRNA-seq data (Supplementary Fig. 28).

**Multiscale smoothing of expression by median filtering**. Unlike linear filtering methods, median filtering preserves the edges while removing noise in smooth regions of signal[24]. CaSpER uses recursive median filtering for removing noise from expression and allele-based frequency signal.

Let $\mathbf{X}^0 = \{x_1^0, x_2^0, \ldots, x_i^0, \ldots, x_n^0\}$ be the expression signal vector, where $x_i^0$ is the original signal value at iteration 0 in position $i$. Given the window length $l$, at scale $s$ median filtering can be formulated as:

$$x_i^s = \text{med}\left(\{x_a^{s-1}\}_{a \in [i - \frac{l_s}{2}, \ i + \frac{l_s}{2}]}\right), \quad (7)$$

where $x_i^s$ is the $i$th value of the median filtered expression signal at scale $s$. In $x_a^{s-1}$, $a$ is defined as smoothing region of each $i$ formulated as $\left\{ i - \frac{l}{2}, \ldots, i, \ldots, i + \frac{l}{2} \right\}$ and the input expression signal $\mathbf{X}^{s-1}$ is the smoothed expression signal in the previous iteration, $s - 1$.

Similar to expression signal, we also apply recursive median filtering to an allele-based frequency signal. Let $\mathbf{Y}^0 = \left\{ y_1^0, y_2^0, \ldots, y_i^0, \ldots, y_n^0 \right\}$ be the expression signal vector, where $y$ is the original allele-based frequency signal value at iteration 0 in position $i$. Given the window length $l$, at scale $s$ median filtering can be formulated as:

$$y_i^s = \mathrm{med}\left( \left\{ y_a^{s-1} \right\}_{a \in \left[ i - \frac{l}{2}, \, i + \frac{l}{2} \right]} \right), \tag{8}$$

where $y_i^s$ is the $i$th value of the median filtered allele-based frequency signal at scale $s$. In $y_a^{s-1}$, $a$ is defined as smoothing region of each $i$ formulated as $\left\{ i - \frac{l}{2}, \ldots, i, \ldots, i + \frac{l}{2} \right\}$ and the input allele-based frequency signal $\mathbf{Y}^{s-1}$ is the smoothed allele-based frequency signal in the previous iteration, $s - 1$.

CaSpER uses filter function in signal R package for median filtering implementation.

**Window length $l$ parameter selection**. We used different starting window sizes l for decompositions. We computed the segment-based accuracy (TPR) of the CNV calls made by CaSpER using the GBM scRNA-seq data in which we simulated introduction of deletions at 1 MB and 10 MB. To introduce deletions into the datasets, we first selected target regions where we are sure that there no CNV events. Next, we sampled the expression signals from the regions with known deletions and replaced the expression levels of the target regions of length 1 MB and 10 MB with these "deleted" expression signals. We ran CaSpER with varying window length $l$ parameter; 10, 30, 50, 100, 150, and 200. For small smoothing window sizes, segments show relatively low concordance with the ground truth (Supplementary Fig. 30). As the window size increases, the concordance increases and saturates around 80% at the around window length of 50 for both event lengths. Another important point to keep in mind is that as the starting window size increases, it takes longer to process the data. Putting this together, the above results indicate that the starting window length of 50 enables a fair tradeoff between computational time and accuracy.

**Pairwise comparisons from the multiscale decomposition**. Step 1. Integration of the HMM segment states with Allele Frequency Shift Information. CaSpER algorithm outputs CNV calls using all the pairwise comparisons of expression signal and BAF signal decompositions. We describe below step-by-step details of the comparison procedure.

CaSpER uses two sources of information. First is the genomewide expression signal and the other is the genomewide allelic shift signal. The expression signal refers to the vector whose elements are the expression levels of all the genes along the genome; $a$th element in expression signal corresponds to the $a$th gene along the genomic coordinates starting from the beginning of the genome. Similarly, allelic shift signal refers to the vector whose elements are the absolute value of the BAF shift of all SNVs identified by CaSpER such that the SNVs are sorted with respect to genomic coordinates. In bulk RNA-seq data, the BAF shift signal is computed independently for each sample. For scRNA-seq data, the BAF shift signal is computed using the reads pooled from all the cells. The reason for the difference in computation of the BAF shift signal in bulk and scRNA-seq is that the scRNA-seq is very sparse. In order to increase the power to detect BAF shift events (i.e. LOH events), we pool the reads and use the pooled reads to compute the BAF shift signal.

CaSpER starts by computing the multiscale decomposition (smoothing) of the expression signal and the BAF shift signal. After the smoothing, we get the scale-by-scale smoothed expression signals and scale-by-scale smoothed BAF shift signals. For brevity, we refer to the scales in the decomposition of the expression signal by "expression scale". We denote the expression scales with $n$. Similarly, we refer to the scales in the decomposition of the BAF shift signal by the "BAF scale" and denote them with $m$.

The smoothed expression signals are segmented using the HMM. The HMM detects the segment boundaries in expression signal and assigns one state to each segment. The segments detected by the HMM-based segmentation of expression signal (at scale $n$) are filtered with respect to concordance with the BAF shift signal. For this, we assign the segments whose HMM states are 1 and 5 the deletion and amplification calls, respectively. In addition, the segments with assigned states of 2 or 4 are assigned deletion and amplification calls, respectively, given that there are accompanying BAF shifts. The details of the BAF shift test are explained in the Methods Section "Calculation of BAF shift threshold using Gaussian mixture models".

To be more concrete, we summarize this procedure below:

Let us assume that the HMM segments from the smoothed expression signal at scale $n$ consist of $K$ segments that are denoted by $X_i^{(n)}$ ($i = 1, \ldots, K$). $X_i^{(n)}$ is the $i$th segment and $n$ is an integer in $[1, N]$, where $N$ denotes the index for the highest smoothing scale for expression signal. We denote the set of all segments at scale $n$

with $\mathbf{X}^{(n)}$. Each $X_i^{(n)}$ is assigned by the HMM one of the 5 integer copy number states. We denote these states with $S\left( X_i^{(n)} \right) \in \{1, 2, 3, 4, 5\}$ such that

1: Homozygous deletion,
2: Heterozygous deletion,
3: Neutral,
4: One-copy amplification,
5: Multi-copy amplification.

The deletion/amplification call for the segment $X_i^{(n)}$, denoted by $CNV^{(m)}\left( X_i^{(n)} \right)$, at expression scale $n$ and BAF scale $m$ is computed as:

$$CNV^{(m)}\left( X_i^{(n)} \right) \begin{cases} 1, & \text{if } \left( S\left( X_i^{(n)} \right) = 5 \right) \text{ or } \left( S\left( X_i^{(n)} \right) = 4 \text{ and median} \left( BAF^m\left( X_i^{(n)} \right) \right) > t \right) \\ -1, & \text{if } \left( S\left( X_i^{(n)} \right) = 1 \right) \text{ or } \left( S\left( X_i^{(n)} \right) = 2 \text{ and median} \left( BAF^m\left( X_i^{(n)} \right) \right) > t \right) \\ 0, & \text{otherwise} \end{cases} \tag{9}$$

where $CNV^{(m)}\left( X_i^{(n)} \right)$ denotes the CNV call for $X_i^{(n)}$ such that $-1$, $1$, $0$ stands for deletion, amplification, and neutral event states, respectively; $m$ denotes the smoothing scale for BAF shift signal and it is an integer in $[1, M]$. Finally, $\mathrm{median}\left( BAF^m\left( X_i^{(n)} \right) \right)$ denotes the median value of the BAF shift signal (smoothed at scale $m$) on the segment $X_i^{(n)}$. The median is computed as the median value of the smoothed B-allele shift signal on all the SNVs within $X_i^{(n)}$. From the above computation, it can be seen that CaSpER assigns deletion or amplification to a segment when the HMM state is 1 or 5 without looking at the BAF signal. When the segment state is 2 or 4, an accompanying BAF shift on the segment is required. As we indicated above, the calculation of the BAF shift threshold $t$ is explained in "Calculation of BAF shift threshold using Gaussian mixture models" in the Methods section. CaSpER also reports the LOH regions where assigned HMM assigns a neutral state (i.e., state 3) and the BAF shift signal is significantly high, i.e., the BAF shift signal is higher than the threshold $t$.

From the above computation, CaSpER assigns CNV calls to the segments for all the pairwise comparisons of BAF and expression signal scales, which is in total ($N \times M$) comparisons. The final step is the harmonization of the CNV calls from all the pairwise comparisons.

Step 2. Harmonization and Summarization of CNV call from multiple scales and from multiple pairwise comparisons of BAF and Expression Signals. The pairwise comparison and assignment of CNV call generate a large number of per-scale information that must be summarized such that each position of the genome is assigned a final call about its CNV status, i.e., deletion/amplification/neutral. We use a consistency-based approach for harmonizing the pairwise comparisons: the events are put together and we assign the final CNV for a gene or large-scale event if the CNV calls are consistent among at least a certain number of pairwise scale comparisons. CaSpER harmonizes and summarizes the CNV calls by dividing them into large-scale, gene-based, and segment-based CNV calls as described below:

1. Large-Scale CNV Summarization. For each pairwise comparison of expression smoothing scale $n$ and BAF shift signal smoothing scale $m$, $CNV^{(m)}(X_i^{(n)})$, the union of the deletion (and amplification) events in every chromosome arm are computed. Next, from the union, we identify the chromosome arms for which the deletions (or amplification) are affecting more than one-third of the chromosomal arm[48]. This way we assign a large-scale CNV call to every chromosome arm for each of the $N \times M$ pairwise scale comparisons. Each chromosome arm gets assigned $N \times M$ large-scale CNV calls.

Next, for each chromosome arm, we ask whether the large-scale CNV call is consistent among at least $\gamma$ of the $N \times M$ large-scale CNV calls. If there is consistency in more than $\gamma$ comparisons, we assign the final large-scale CNV call of the chromosome arm as the consistent call. If there is no consistency, we assign a neutral CNV call for the chromosome arm. This procedure is repeated for every sample (or cell). We summarize the consistency-based large-scale CNV calls in a matrix where rows are the samples (or cells) and columns are the chromosome arms. The matrix entry of 0 corresponds to no alteration, 1 corresponds to amplification and $-1$ corresponds to deletion.

2. Gene-Based Summarization. Similar to the large-scale summarization, we generate a matrix where rows are the samples (cells) and columns are the genes. The matrix entry of 0 corresponds to no alteration, 1 corresponds to amplification and $-1$ corresponds to deletion. If an alteration is consistent in more than $\gamma$scale comparisons (out of $N \times M$ comparisons), we report that alteration event for that sample.

3. Segment-Based Summarization. The segments-based summarization aims at generating a final set of CNV calls for a final set of segments that are computed by a comparison of scales. We first compare the segments from different expression scales and generate a consistent set of segments. For this, we first identify the final set of segments. To generate the final set of segments, we first pool the ends of all the segments, $\mathbf{X}^{(1)}, \ldots, \mathbf{X}^{(N)}$ from all the scales l to N. We then sort the segment ends with respect to genomic coordinates, then we generate a new final segments as the regions between consecutive segment ends (Supplementary Fig. 31). We denote these final set of segments by $\mathbf{Y}$. Note that there is no scale notation in the final set of segments since these are identified using segments from all the scales. A

hypothetical example for detection of **Y** using segments from three different scales is shown in (Supplementary Fig. 31).

Note that for any final segment $Y_j \in \mathbf{Y}$, there is exactly one segment in $\mathbf{X}^{(n)}$ that intersects with $Y_j$. With this in mind, the next step is the assignment of CNV calls to the segments in the final segment set. For each segment $Y_j \in \mathbf{Y}$, we assign $N \times M$ CNV calls using $\mathrm{CNV}^{(m)}(\mathrm{X}_i^{(n)})$. For this, we take $Y_j$, identify the $N$ scale-specific segments, i.e., $\mathbf{X}^{(n)}$, $n = 1, 2, .., N$, that intersect with it, then pool $M$ CNV calls from each scale-specific segment. At the end of this procedure, we assign $N \times M$ CNV calls to each $Y_j$. The final step is assignment of the final CNV call for $Y_j$. This is performed similar to the consistency-based assignment we used before: For each $Y_j$, if there are more than $\gamma$ consistent CNV calls among $N \times M$ CNV calls, we assign the consistent CNV call to $Y_j$. When there is no consistency among the calls, we assign a neutral CNV state to $Y_j$.

Selection of $\gamma$ parameter. $\gamma$ represents the minimum number of consistent CNV calls (out of $N \times M$ comparisons of expression scales and BAF scales) while assigning a final CNV (amp/del/neutral) call to a segment/gene/chromosome arm. To visualize the effect of changing $\gamma$ on different datasets, we computed the large-scale and gene-based event summaries of CNV calls for several datasets and calculated TPR and FPR using genotyping arrays as the gold standard. We describe TPR and FPR calculations in "Performance Metrics" in the Methods section. $\gamma$ parameter tunes the tradeoff between FPR and TPR rates where low (high) $\gamma$ implies high (low) TPR and high (low) FPR. In the manuscript, to balance FPR and TPR rates we used 7 as a $\gamma$ threshold. The user can change $\gamma$ threshold to be more stringent (higher $\gamma$) or relaxed (lower $\gamma$) in terms of consistency among the pairwise scale comparisons.

**Gaussian mixture models.** CaSpER models the allele-based frequencies as a mixture of Gaussian distributions for identification and classification of genotype clusters. For example, in a normal chromosomal region with two copies, we expect to observe three BAF genotype clusters represented as AA, AB, and BB whereas, in heterozygous deletions, we expect to observe two clusters which can be represented as A and B.

Let $\mathbf{X} = \{x_1, x_2, \ldots, x_i, \ldots, x_n\}$ be the allele-based frequency signal vector, where $x_i$ is the signal value at position $i$. The distribution of every value is specified by a probability density function through a finite mixture model of $G$ classes:

$$f(x_i; z) = \sum_{k=l}^{G} \pi_k f_k(x_i; \theta_k),\qquad(10)$$

where $z = \left\{\pi_{1, \ldots, \pi_{G-1}}, \theta_{1, \ldots, \theta_G}\right\}$ is the parameters of the mixture model and $f_k(x_i; \theta_k)$ is the $k$th component density, which assumes to follow Gaussian distribution $f_k(x_i; \theta_k) \sim N(\mu_k, \sigma_k)$. $\{\pi_{1, \ldots, \pi_{G-1}}\}$ is the vector of probabilities, non-negative values which sum to 1, known as the mixing proportions. Mixing proportions, $\pi$, follows a multinomial distribution.

The model $z$ parameters are estimated by maximizing log-likelihood function via the EM algorithm. The log-likelihood function is formulated as:

$$l(z; x) = \sum_{i=1}^{n} \log f_k(x_i; z).\qquad(11)$$

The number of classes, $G$, are estimated using the Bayesian Information Criteria (BIC). The class with the lowest mean value corresponds to alleles without any BAF shift. We choose the class with second lowest mean value, called 'class 2' to identify the BAF shift threshold. In bulk sequencing data, we set the BAF shift threshold to mean allele-based frequency signal in 'class 2'. In scRNA-seq data, we set the BAF shift threshold to a minimum allele-based frequency signal in 'class 2'. CaSpER uses mclust R package for GMM implementation[49].

**Hidden Markov model.** CaSpER uses a modified version of HMMCopy R package[50] for HMM implementation to (1) segment the copy number profile in regions predicted to be generated by the same copy number event and (2) predict the CNV event for each segment. In HMM, we use a hierarchical Bayesian model where the posterior estimates are calculated using an exact likelihood function.

Our HMM model contains 5 CNV states, where the states represent homozygous deletion, heterozygous deletion, neutral, one-copy gain and multiple-copy gain. The initial transition matrix is defined as:

$$\begin{pmatrix} 1-t & t & t & t & t \\ t & 1-t & t & t & t \\ t & t & 1-t & t & t \\ t & t & t & 1-t & t \\ t & t & t & t & 1-t \end{pmatrix},$$

where $t$ is equal to 1e−07. While HMM segmentation is performed, the model performs an expectation–maximization procedure (i.e., Baum–Welch Algorithm) and iteratively updates the transition probabilities. Consequently, the final segmentations are computed using the updated transition probabilities that are fit to the data. The final transition matrices are not symmetric (unlike initial matrices). It is also worth noting that the final transition matrix is data dependent as the EM-step fits the transition matrix to the data.

Emission probabilities are represented by a normal distribution with means and variance derived from normalized expression data. We believe that using more data by pooling all the samples would give better initial HMM parameter estimates. For each dataset, we estimate the mean values by pooling all the samples in that dataset. For each dataset, mean values are calculated using sample quantiles corresponding to 0.01 (homo-del), 0.05 (het-del), 0.5 (neutral), 0.95 (amp), 0.99 (high-amp) probabilities. In the segmentation process, the transition and emission parameters are updated by the HMM model in the segmentation to ensure that the emission probabilities (and transitions) of the CNV states are tuned exactly to the data.

Single-cell GBM data initial emission probability normal distribution mean values are much different than bulk RNA-seq datasets (Supplementary Fig. 32). Moreover, the accuracy of CNV calls increases with the number of cells (Supplementary Note 11, Supplementary Figs. 33–36).

**Assessing the performance of CaSpER on CNVs of varying clonality.** In general, we denote the clonality of a deletion by $c$, where

$$c = \frac{\text{Number of cells with deletion}}{\text{Total number of cells}},\qquad(12)$$

and $c$ denotes the fraction of cells that harbor the deletion.

To study how clonality impacts the detected CNVs, we simulated gene expression levels of scRNA-seq experiments by controlled introduction of deletions into a certain set of target regions. To introduce deletions into the datasets, we first selected target regions where we are sure that there no CNV events. Next, we sampled the expression signals from the regions with known deletions and replaced the expression levels of the target regions with these "deleted" expression signals in $P \times c$ cells where $P$ denotes the total number of cells. This way, we simulate deleted expression signals at the target regions on the $c$ cells that harbor deletions. After the expression signals are simulated, we added a sample of $P \times (1-c)$ normal cells into the population. These cells represent the fraction of cells that do not harbor the deletion. We finally created the "bulk" expression signal by taking the averages among the $P \times (1 - c)$ normal and $P \times c$ simulated expression signals. This procedure generates a simulated bulk expression signal data on $P$ cells.

Next, we simulated the BAF shift signal in the target regions. This is accomplished by using the following formulation:

$$\mathrm{BAF_{shift}}(\text{target deletion}) = \frac{P \times c \times 1 + P \times (1-c) \times 1}{P \times c \times 1 + P \times (1-c) \times 2} - 0.5,\qquad(13)$$

where $c$ denotes the clonal fraction of the deletion. The numerator in the above formula computes the total amount of major allele at the deletion site: 1 copy that is originating from the cells with deletion ($P \times c$ cells) and the 1 copy that is originating from the cells that do not have the deletion ($P \times (1 - c)$ cells). The denominator computes the total DNA from all the cells at the deletion site: 1 copy originating from the cells that harbor the deletion and the 2 total copies that are originating from the cells that do not harbor the deletion. The shift computed using the above formula is simply added to the existing BAF shift signal in the target regions.

Thus, we modified both the expression signal (using expression signals of regions with known deletions) and the BAF shift signal at the target regions. In the simulations, we also divided the segments with respect to lengths 1 mb, 10 mb, and 100 mb.

**Calculating mutual-exclusive and co-occurring CNV events.** We calculated the distance between cells using the Jaccard distance metric. We next used this distance matrix to build the phylogenetic tree of the CNV events. A phylogenetic tree is constructed using the Fitch–Margoliash method implemented in the Rfitch R package. We finally plotted the tree using phydataplot function in ape R package. The co-occurrence and mutual exclusivity of CNV events were assessed using one-sided Fisher's exact test.

**Identifying gene expression signatures and enrichment analysis.** Differentially expressed genes were identified using an empirical Bayesian method ebayes implemented in limma R package[51]. Genes were considered differentially expressed with adjusted $P$-value < 0.05. GO term enrichment analysis was performed using the GOStats R package[52].

**Genotyping data.** TCGA-GBM genotyping data were downloaded using TCGA-Biolinks R package[42]. CNV segments with a mean log ratio value > 0.3 were defined as amplification whereas segments with mean log ratio value < 0.3 were defined as deletion. Large-scale chromosomal deletion or amplification was defined as affecting more than one-third of the chromosomal arm, whereas focal event deletion or amplification was defined as affecting less than one-third and more than one-tenth of the chromosomal arm with accompanying log ratio of signal intensities < −0.1 or >0.1 and BAFs at heterozygous sites deviating from 0.5 by at least 0.05 units.

In meningioma genotyping data, CNVs were detected by comparing the normalized signal intensity between a tumor and matched blood or a tumor and the average of all blood samples[26]. Segmentation was performed on log intensity ($R$) ratios using DNACopy algorithm[53].

**Performance assessment**. We assessed the performance of CaSpER by comparing the CNV calls identified from RNA-seq with genotyping data. Thus, the TPR is the percentage of large-scale or gene-level CNV events which are correctly identified by CaSpER, while the FPR is the percentage of falsely rejected true CNV events. To clarify these definitions, we summarized them in the Supplementary Table 3.

Based on these definitions, the TPR and the FPR are calculated as: TPR = TP/(TP + FN), FPR = FP/FP + TN.

When computing the accuracy at the segment level, we compare the segments from each method with the segments in the ground truth dataset, i.e. genotyping array segments. We illustrate the false-negative, false-positive, and true-positive portions of the segments as in Supplementary Fig. 37. In this figure, X represents the true CNV segment (from genotyping array) and Y represents the detected segment by CaSpER.

1. FN denotes the length of the true segment that does not overlap with the detected segment.
2. TP denotes the length of the detected segment that overlaps with the true segment.
3. FP denotes the length of the detected segment that does not overlap with the true segment.

In order to compute the accuracy metrics over all the segments, we take the weighted average of the TP, FP, and FN computed for each segment by appropriate length. Let us assume that the final CaSpER CNV segments are denoted by $Y_i$: $i = 1, \dots, N$ and the genotyping array CNV segments are denoted by $X_j: j = 1, \dots, M$. We define Precision (or PPV) using the following fractions:

$$\text{PPV (precision)} = \frac{\sum_i \text{TP}(Y_i)}{\sum_i |Y_i|}, \tag{14}$$

$$\text{FPR} = \frac{\sum_i \text{FP}(Y_i)}{\sum_i |Y_i|}, \tag{15}$$

$$\text{TPR(sensitivity)} = \frac{\sum_i \text{TP}(Y_i)}{\sum_j |X_j|}, \tag{16}$$

$$\text{FNR} = \frac{\sum_i \text{FN}(Y_i)}{\sum_j |X_j|}. \tag{17}$$

**Reporting Summary**. Further information on research design is available in the Nature Research Reporting Summary linked to this article.

## Data availability

All the data used in this study are from previously published studies. Meningioma data are downloaded from GEO (accession: GSE85133). Single-cell GBM data are downloaded from GEO (accession: GSE57872 (https://www.ncbi.nlm.nih.gov/sra?term=SRP042161)). TCGA-GBM and TCGA-BRCA RNA-seq data are downloaded from TCGA data portal (https://portal.gdc.cancer.gov/). DR-Seq study is downloaded from GEO (accession: GSE62952 (https://www.ncbi.nlm.nih.gov/sra?term=SRP049500)). 10x mouse scRNA-seq is downloaded from GEO (accession: GSE121861 (https://www.ncbi.nlm.nih.gov/sra?term=SRP166967)). scRNA-seq data for the MM cells are downloaded from GEO (accession: GSE110499 (https://www.ncbi.nlm.nih.gov/sra?term=SRP132719)).

## Code availability

CaSpER source code and documentation are publicly available at https://github.com/akdess/CaSpER.

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

## Acknowledgements

This study is partially funded by the NIH grants (R01GM123037, R01CA241930, U01AR69393, U01CA166866) and STARs award. The results shown here are in part based upon data generated by the TCGA Research Network: http://cancergenome.nih.gov/. The authors acknowledge the Texas Advanced Computing Center (TACC) at The University of Texas at Austin for providing HPC resources that have contributed to the research results reported within this paper: http://www.tacc.utexas.edu.

## Author contributions

A.S.H and A.O.H. initiated and designed the study, implemented the code, performed the analysis, and wrote the manuscript. X.Z. conceived the study. All authors edited and approved the final manuscript.

## Competing interests

The authors declare no competing interests.
