## [Peer Review File · Nature Communications]

Reviewers' comments:

Reviewer #1 (Remarks to the Author):

The manuscript describes a method (CaSpER) to analyze copy number variations (CNVs) in single cell or bulk RNA sequencing data. There is a need for new tools in this area and CaSpER takes an interesting approach. For example, the median filtering approach seems quite nice. However, several critical components of the method are not described well or at all, it is not clear how well the data support conclusions in the manuscript, and insufficient comparison is made to other methods.

Major comments:

- 1) Several critical components of the method are really not described. For example:
 - a. "CaSpER performs a pairwise comparison of all scales from BAF and expression signals to ensure a coherent set of CNV calls" How is this done? What thresholds are used and why?
 - b. What determines the emission probabilities of the HMM? Do these differ for bulk data (which can be impure) vs single cells (which cannot)?
 - c. Are raw expression values used, or are the data normalized somehow? Raw expression values would seem to vary by gene simply because of the strength of the promoter etc.
 - d. BAF signals are interpreted differently for bulk vs single cell data. What is the justification for the different thresholds used?
- 2) Conclusions are drawn that do not seem well-supported by data. For example:
 - a. "The smoothed signal distribution shows that data does not contain many amplification events (Figure 4B)". I do not see this in the figure.
 - b. "CaSpER achieves 95% TPR and 0.3% FPR for detecting deletion events (Figure 4F)." All events, or in a certain size regime?
 - c. Why do the authors claim the BAF shift in chromosome 14 for MGH31 is subclonal? No rationale is provided
 - d. For MGH31, the overall decomposition doesn't seem to make sense. For example, I see 9p loss both among 1p amp and 13q loss cells, and other 1p amp and 13q loss cells that do not have 9p loss. This doesn't seem to accord with an evolutionary tree, or with the results indicated in Figure 5E. Indeed, the organization of the data in Figure 5C make assessments of correlation and mutual exclusivity difficult to assess. Figure 5C has colors at the left that are not described; I imagine these refer to subclone assignments but it is not clear how those assignments were made. Figure 5E shows edges for co-occurring events, but the significance levels associated with an edge of a certain width is not clear—no legend is provided.
- 3) How do the TPRs and FPRs for CaSpER compare to existing methods? Note that meningiomas as a test set is not ideal: it tends to have few CNVs, and few focal CNVs. CaSpER would better be tested in a more complex tumor type, such as breast or ovarian cancer. In particular, GBM and meningioma tend to be diploid, and it will be important to test CaSpER in cancers with more frequent tetraploidy/genome doubling events
- 4) The use of pooled BAF data seems to counter the whole point of the method to evaluate single-cell changes. If this needs to be done, why not use pooled data to assess a baseline shared by most cells, and then assess expression data explicitly to detect deviations from that baseline?

Minor comments:

- 1) The introduction is too long.
- 2) In general, the figures are difficult to understand and not well-labeled. For example, what does the y-axis refer to in 3b-c? What do the dots in 3E represent? The existing labels are hard to decipher

Reviewer #3 (Remarks to the Author):

Overview

Beyond simply quantifying expression levels, single-cell RNA-seq data contain other potentially important signals such as from copy-number alterations that can be elucidated with the appropriate bioinformatics tools. Harmanci et al. present a computational approach called CaSpER that strives to enable identification, visualization, and integrative analysis of focal and large-scale CNVs using either bulk or single-cell RNA-seq data. Although the authors acknowledge previous methods for CNV inference using single-cell RNA-seq data, including inferCNV and HoneyBADGER, comparisons with these previously published approaches are not provided. The suitability of CaSpER for identifying smaller focal alterations, multi-copy amplifications, small subclonal alterations (affecting a small percentage of cells) and clonal alterations (affecting 100% of cells) remain to be convincingly shown, as benchmarks have focused on chromosome-arm-level single-copy deletions and amplifications affecting large subclones. Similarly, the sensitivity and accuracy of the method at different resolutions (e.g. for different CNV sizes and clonality) remain unclear. As such, the manuscript is not suitable for publication at this stage.

Major comments

1. The authors state that "CaSpER performs well in accuracy compared to gold-standard SNP genotyping arrays." How does CaSpER compare to other sequencing-based CNV inference approaches, in particular to other single-cell RNA-seq CNV inference approaches such as inferCNV and HoneyBADGER?

2. The authors measure the accuracy of CaSpER by comparing identified CNVs with those called from genotyping arrays such that "the true positive rate is the percentage of large-scale CNV events who are correctly identified by CaSpER while the false positive rate is the percentage of falsely rejected true CNV events." This is incorrect. The false positive rate is the percentage of falsely identified CNV events (ie. CaSpER identifies a CNV that the genotyping array does not). Quantification of the false positive rate by this definition is currently lacking and should be provided. The false negative rate is the percentage of falsely rejected true CNV events (ie. CaSpER does not identify a CNV but the genotyping array says a CNV is present). Still, it is unclear whether this accuracy assessment is done on the CNV level or base-pair level. For example, if a true CNV affects chromosome 1 bases 1 to 5e6, but CaSpER calls a CNV affecting chromosome 1 bases 1 to 6e6, is this considered a true positive on chromosome 1? Or are the base pair resolution of the CNV also considered?

3. The authors apply CaSpER to single-cell GBM RNA-seq data from 5 patients. A small number of cells from one of these patients, MGH31, was shown in the original publication to be normal immune cells. It is unclear from Figure 5 whether these normal cells were properly identified by CaSpER, as they should have harbor no CNVs. Please comment on this discrepancy. Similarly, the authors identify a number of new CNVs in the GBM data, previously undetected in the original publication. Please comment on this discrepancy. What are the sizes of these alterations? Were they missed previously due to the resolution limitations of previous methods? How do we know they are not false positives? Are there any validations available for these new identified CNVs such as cytogenetics or bulk DNA sequencing?

4. The authors suggest that CaSpER is suitable to identifying focal alterations, but most of the alterations identified are chromosome to chromosome-arm level. In Figure 6, the authors do show that CaSpER is able to identify a focal amplification in PDGFRA using small-scale lengths that is consistent with gold-standard genotyping. However, in the same figure, gold-standard genotyping appears to identify another focal amplification of comparable size (around base pair 1e8) that CaSpER fails to identify. Is CaSpER capable of reliably identifying CNVs on the scale of 1 or 2 megabases? A more thorough quantification of CaSpER's ability to identify CNVs as a function of CNV size is needed.

5. Similarly, the authors note how "lower clonality rates in meningiomas lead to better deletion CNV event identification. Similarly, high clonality rates in GBM tumors lower the detection accuracy of low-level amplification events." Please provide a more quantitative basis for this observation and quantify CaSpER's performance as a function of clonality. As CaSpER uses B-allele frequencies from RNA-seq reads to identify CNV events does not require heterozygous variants to be called using a normal reference, it is unclear how B-allele frequencies can be derived for clonal heterozygous deletions, since only 1 allele can be observed. Similarly, for clonal homozygous deletions, no allelic information will be available. Is CaSpER suitable for detecting clonal alterations?

6. The authors present CaSpER as a tool for visualizing genome-wide RNA-seq signals. However, it is unclear which visualization(s) they are referring to that are unique to CaSpER and what these new visualizations enable that previous visualizations, such as from inferCNV, do not. Similarly, the authors suggest that "these visualizations are especially useful for visually confirming the significance of the results." Please clarify how these visualizations enable users to confirm "significance of the results."

7. The CaSpER R package currently lacks a manual and is generally poorly documented. The authors should document each function using Roxygen notation and export a manual as well as provide outputted vignettes with sample usage instructions to facilitate users.

Minor comments

1. As 3'-only RNA sequencing techniques such as Drop-seq, In-drop, 10X Genomics, and others are rapidly becoming more popular, please comment on whether CaSpER is suitable for these 3' only RNA sequencing data. How will its accuracy in identifying CNVs of different sizes or clonality be affected?

2. As CaSpER relies on a sliding window-based median filtering, how well can CaSpER call the precise boundaries of CNVs? Does this depend on the window size used?

3. In the HMM model, the authors use a symmetric transition matrix where there is equal likelihood of transitioning from every state to every other state. For example, given an HMM model initialized in the copy neutral state (state 3), it is equally likely (with probability $t=1e-6$) to transition to a homozygous deletion, heterozygous deletion, one-copy gain or multiple-copy gain. Similarly, given an HMM model at a multiple-copy gain state, it is equally likely to transition to a homozygous deletion, heterozygous deletion, neutral or one-copy gain state. How did the authors arrive at this symmetric transition matrix?

4. As CaSpER's HMM model distinguishes between one-copy gain and multiple-copy gain, how well does CaSpER distinguish between the two states? Similarly, would CaSpER be able to identify multi-copy duplication events where both alleles have been amplified equally? In this case, the B-allele frequency would remain comparable to a neutral region's.

5. Similarly, as the authors mention LOH in addition to CNVs, can CaSpER identify copy-neutral LOH events?

6. The authors note that "identifying CNVs before calling SNVs can give very useful information for correct identification of SNVs." On the flip side, point mutations will also affect the B-allele frequency, potentially impacting CNV detection. Other sources of noise unique to RNA such as RNA-editing, amplification errors, and sequencing errors may also contribute to noise in the B-allele frequency. Please comment on the impact of point-mutations, RNA-editing, amplification errors, and sequencing errors on CNV identification with CaSpER.

7. To generate the appropriate allele-based frequency signals from RNA-seq bam files, do reads need to be filtered for duplicates? Or is the minimum number of total reads supporting a SNP including potential duplicates?
8. The authors note that "for 5q:14q event pair in patient MGH31, we discovered GFPT2 gene to be highly expressed in 5q amplified clone." And "for 5q:19q event pair in patient MGH28, we discovered NOS2 gene to be highly expressed in 19q deletion clone." Are these genes located within the amplified regions? What proportion of affected genes constitute cis effects (within affected region) vs. trans effects (outside affected region)?
9. The motivation for calling CNVs from bulk RNA-seq is unclear. It is very common to do both bulk DNA and bulk RNA sequencing on the same sample, particularly in cancer. So it is unclear why inferring CNVs from bulk RNA-seq would be desirable.
10. The authors note that "it remains technically challenging to assay both the genome and transcriptome from the same cell." However, a number of approaches for simultaneously assaying DNA and RNA information from the same single cell has been published, including G&T-Seq by Macaulay et al (Nature Methods 2015), and BioMark by Wang et al (Genome Research 2017) to name a few. While it may still be true that these approaches are too technically challenging for broad-scale implementation, they should be acknowledged.
11. The authors note that "many algorithms have been developed for detecting CNV events from DNA sequencing using depth of coverage." Many algorithms have also been developed for detecting CNV events from DNA sequencing using B allele frequency, including PennCNV by Lima et al (BMC Bioinformatics 2017). Please acknowledge these previous works.
12. The authors claim that "loss-of-heterozygosity...has previously shown to be extremely useful for identifying CNVs." Previous provide a citation.
13. Similarly, the authors note that "unlike most other tools, CaSpER does not require heterozygous variant calls to generate the allelic shift profile." Please provide a citation for these "other tools".

We sincerely thank the reviewers for their insightful comments and constructive criticisms, which have helped to substantially improve our manuscript. Based on their input, we have performed numerous additional experiments and revised the manuscript. We provide below our detailed point-by-point response to their valuable comments.

Reviewer #1 (Remarks to the Author):

The manuscript describes a method (CaSpER) to analyze copy number variations (CNVs) in single cell or bulk RNA sequencing data. There is a need for new tools in this area and CaSpER takes an interesting approach. For example, the median filtering approach seems quite nice. However, several critical components of the method are not described well or at all, it is not clear how well the data support conclusions in the manuscript, and insufficient comparison is made to other methods.

We thank the reviewer for the constructive comments. As the reviewer points out, several critical parts of the method description were not clearly presented. We have now revised several sections in detail for which the reviewer asked for clarification. Moreover, we have included performance comparisons with the existing tools in the Results Section.

Major comments:

1) Several critical components of the method are really not described. For example:

a. "CaSpER performs a pairwise comparison of all scales from BAF and expression signals to ensure a coherent set of CNV calls" How is this done? What thresholds are used and why?

Authors' Response: We appreciate the reviewer's critical input that helps to clarify this important point. We agree that the pairwise comparison of the scales in BAF and expression signals were not clearly explained. Based on this suggestion, we included the details of the pairwise comparison of scales from BAF and expression signals in Methods section (**new text line 618-737**). We also include the details of comparison here:

Step 1. Integration of the HMM segment states with Allele Frequency Shift Information. CaSpER algorithm outputs CNV calls using all the pairwise comparisons of expression signal and BAF signal decompositions. We describe below step-by-step details of the comparison procedure.

CaSpER uses two sources of information. First is the genomewide expression signal and other is the genomewide allelic shift signal. The expression signal refers to the vector whose elements are the expression levels of all the genes along the genome. a^{th} element in expression signal corresponds to the a^{th} gene along the genomic coordinates starting from the beginning of the genome. Similarly, allelic shift signal refers to the vector whose elements are the absolute value of the B-allele frequency (BAF) shift of all SNVs identified by CaSpER such that the SNVs are sorted with respect to genomic coordinates. In bulk RNA-seq data, the BAF shift signal is computed independently for each sample. For single cell RNA-seq data, the BAF shift signal is computed using the reads pooled from all the cells (See Response for the Comment 1. d. below for more detailed explanation of BAF shift computation). The reason for the difference in computation of the BAF shift signal in bulk and single cell RNA-seq is that the single cell RNA-seq is very sparse. In order to increase the power to detect BAF shift events (i.e. LOH events), we pool the reads and use the pooled reads to compute the BAF shift signal.

CaSpER starts by computing the multiscale decomposition (smoothing) of the expression signal and the BAF shift signal. After the smoothing, we get the scale-by-scale smoothed expression signals and scale-by-scale smoothed BAF shift signals. For brevity, we refer to the scales in the decomposition of the expression signal by “expression scale”. We denote the expression scales with n . Similarly, we refer to the scales in the decomposition of the BAF shift signal by “BAF scale” and denote them with m .

The smoothed expression signals are segmented using the hidden Markov Model (HMM). The HMM detects the segment boundaries in expression signal and assigns one state to each segment. The segments detected by the HMM-based segmentation of expression signal (at scale n) are filtered with respect to concordance with the BAF shift signal. For this, we assign the segments whose HMM states are 1 and 5 the deletion and amplification calls, respectively. In addition, the segments with assigned states of 2 or 4 are assigned deletion and amplification calls, respectively, given that there are an accompanying BAF shifts. The details of the BAF shift test are explained in revised Methods Section Named “*Calculation of BAF shift threshold using Gaussian mixture models*”.

To be more concrete, we summarize this procedure below:

Let us assume that the HMM segments from the smoothed expression signal at scale n consists of K segments that are denoted by $X_i^{(n)}$ ($i = 1, \dots, K$). $X_i^{(n)}$ is the i^{th} segment and n is an integer in $[1, N]$, where N denotes the index for the highest smoothing scale for expression signal. We denote the set of all segments at scale n with $X^{(n)}$. Each $X_i^{(n)}$ is assigned by the HMM one of the 5 integer copy number states. We denote these states with $S(X_i^{(n)}) \in \{1, 2, 3, 4, 5\}$ such that

- 1: Homozygous deletion,
- 2: Heterozygous deletion,
- 3: Neutral,
- 4: One-copy amplification,
- 5: Multi-copy amplification.

The deletion/amplification call for the segment $X_i^{(n)}$, denoted by $CNV^{(m)}(X_i^{(n)})$, at expression scale n and BAF scale m is computed as:

$$CNV^{(m)}(X_i^{(n)}) = \begin{cases} 1, & \text{if } (S(X_i^{(n)}) = 5) \text{ or } (S(X_i^{(n)}) = 4 \text{ and } \widehat{BAF^{(m)}}(X_i^{(n)}) > t) \\ -1, & \text{if } (S(X_i^{(n)}) = 1) \text{ or } (S(X_i^{(n)}) = 2 \text{ and } \widehat{BAF^{(m)}}(X_i^{(n)}) > t) \\ 0, & \text{otherwise} \end{cases}$$

where $CNV^{(m)}(X_i^{(n)})$ denotes the CNV call for $X_i^{(n)}$ such that -1, 1, 0 stands for deletion, amplification, and neutral event states respectively. m denotes the smoothing scale for BAF shift signal and it is an integer in $[1, M]$. Finally, $\widehat{BAF^{(m)}}(X_i^{(n)})$ denotes the median value of the BAF shift signal (smoothed at scale m) on the segment $X_i^{(n)}$. The median is computed as the median value of the smoothed B-allele shift signal on all the SNVs within $X_i^{(n)}$. From above computation, it can be seen that CaSpER assigns deletion or amplification to a segment when the HMM state is 1 or 5 without looking at the BAF signal. When the segment state is 2 or 4, an accompanying BAF shift on the segment is required. As we indicated above, the calculation of the BAF shift

threshold t is explained in “*Calculation of BAF shift threshold using Gaussian Mixture Models*” in the Methods Section.

From above computation, CaSpER assigns CNV calls to the segments for all the pairwise comparisons of BAF and expression signal scales, which is in total $(N \times M)$ comparisons. Final step is the harmonization of the CNV calls from all the pairwise comparisons.

Step 2. Harmonization and Summarization of CNV calls from multiple scales and from multiple pairwise comparison of BAF and Expression Signals. The pairwise comparison and assignment of CNV calls generates a large number of per-scale information that must be summarized such that each position of the genome is assigned a final call about its CNV status, i.e., deletion/amplification/neutral. We use a consistency-based approach for harmonizing the pairwise comparisons: The events are put together and we assign the final CNV for a gene or large-scale event if the CNV calls are consistent among at least a certain number of pairwise scale comparisons. CaSpER harmonizes and summarizes the CNV calls by dividing them into **large-scale**, **gene-based**, and **segment-based** CNV calls as described below:

1. Large-Scale CNV Summarization. For each pairwise comparison of expression smoothing scale n and BAF shift signal smoothing scale m , $CNV^{(m)}(X_i^{(n)})$, the union of the deletion (and amplification) events in every chromosome arm are computed. Next, from the union, we identify the chromosome arms for which the deletions (or amplifications) are affecting more than one-third of the chromosomal arm¹. This way we assign a large-scale CNV call to every chromosome arm for each of the $N \times M$ pairwise scale comparisons. Each chromosome arm gets assigned $N \times M$ large-scale CNV calls.

Next, for each chromosome arm, we ask whether the large-scale CNV call is consistent among at least γ of the $N \times M$ large-scale CNV calls. If there is consistency in more than γ comparisons, we assign the final large-scale CNV call of the chromosome arm as the consistent call. If there is no consistency, we assign a neutral CNV call for the chromosome arm. This procedure is repeated for every sample (or cell). We summarize the consistency based large-scale CNV calls in a matrix where rows are the samples (or cells) and columns are the chromosome arms. The matrix entry of 0 corresponds to no alteration, 1 corresponds to amplification and -1 corresponds to deletion.

2. Gene-Based Summarization. Similar to the large-scale summarization, we generate a matrix where rows are the samples (cells) and **columns are the genes**. The matrix entry of 0 corresponds to no alteration, 1 corresponds to amplification and -1 corresponds to deletion. If an alteration is consistent in more than γ scale comparisons (out of $N \times M$ comparisons), we report that alteration event for that sample.

3. Segment-Based Summarization. The segments-based summarization aims at generating a final set of CNV calls for a final set of segments that are computed by comparison of scales. We first compare the segments from different expression scales and generate the consistent set of segments. For this, we first identify the final set of segments. To generate the final set of segments, we first pool the ends of all the segments, $X^{(1)}, \dots, X^{(N)}$ from all the scales 1 to N . We then sort the segment ends with respect to genomic coordinates, then we generate a new final segments as the regions between consecutive segment ends (See Figure below). We denote these final set of segments by Y . Note that there is no scale notation in the final set of segments since these are identified using segments from all the scales. A hypothetical example for detection of Y using segments from 3 different scales is shown below:

Updated Supplementary Figure 24. A toy example for detection of final consistent segments.

Note that for any final segment $Y_j \in Y$, there is exactly one segment in $X^{(n)}$ that intersects with Y_j . With this in mind, next step is assignment of CNV calls to the segments in the final segment set. For each segment $Y_j \in Y$, we assign $N \times M$ CNV calls using $CNV^{(m)}(X_i^{(n)})$. For this, we take Y_j , identify the N scale-specific segments, i.e., $X^{(n)}, n = 1, 2, \dots, N$, that intersect with it, then pool M CNV calls from each scale-specific segment. At the end of this procedure, we assign $N \times M$ CNV calls to each Y_j . The final step is assignment of the final CNV call for Y_j . This is performed similar to the consistency-based assignment we used before: For each Y_j , if there are more than γ consistent CNV calls among $N \times M$ CNV calls, we assign the consistent CNV call to Y_j . When there is no consistency among the calls, we assign a neutral CNV state to Y_j .

Selection of γ parameter. γ represents minimum number of consistent CNV calls (Out of $N \times M$ comparisons of expression scales and BAF scales) while assigning a final CNV (amp/del/neutral) call to a segment/gene/chromosome arm. To visualize the effect of changing γ on different datasets, we computed the large-scale and gene-based event summaries of CNV calls for several datasets and calculated true positive rate (TPR) and false positive rate (FPR) using genotyping arrays as gold standard. We describe TPR and FPR calculation in “Performance Metrics” in the Methods section. As it is seen in the below plots, γ parameter tunes the tradeoff between FPR and TPR rates where low (high) γ implies high (low) TPR and high (low) FPR. In the manuscript, to balance FPR and TPR rates we used 7 as a γ threshold. The user can change γ threshold to be more stringent (higher γ) or relaxed (lower γ) in terms of consistency among the pairwise scale comparisons.

Updated Figure 4C. TPR and FPR values for bulk TCGA-GBM RNA-Seq data with varying γ thresholds. Plot shows the large scale event performance which is assessed using genotyping array. Labels in red points represent the γ threshold.

Updated supplementary Figure 3. TPR and FPR values for bulk TCGA-GBM RNA-Seq data with varying γ thresholds. Plot shows the gene based performance which is assessed using genotyping array. Labels in red points represent the γ threshold.

Updated supplementary Figure 9. TPR and FPR values for bulk TCGA-BRCA RNA-Seq data with varying γ thresholds. Plot shows the large scale event performance which is assessed using genotyping array. Labels in red points represent the γ threshold.

Updated supplementary Figure 10. TPR and FPR values for bulk TCGA-BRCA RNA-Seq data with varying γ thresholds. Plot shows the gene based performance which is assessed using genotyping array. Labels in red points represent the γ threshold.

Updated supplementary Figure 1. TPR and FPR values for bulk Meningioma deletion CNV events with varying γ thresholds. Plot shows the large scale event performance which is assessed using genotyping array. Labels in red points represent the γ threshold.

b. What determines the emission probabilities of the HMM? Do these differ for bulk data (which can be impure) vs single cells (which cannot)?

Authors' Response: We thank the reviewer for this comment regarding the details of hidden Markov Model (HMM) parameters. The HMM is supplied with an initial set of parameters that are derived from the data. Estimation of the emission probabilities from the data enables CaSpER to tune its parameters to the data itself specifically, rather than using a preselected set of parameters, which may bias the results globally.

Emission probabilities are represented by a normal distribution with means and variance derived from normalized expression data. For each dataset, we estimate the mean values by pooling all the samples in that dataset. For each dataset, mean values are calculated using sample quantiles corresponding to 0.01 (State 1: homo-del), 0.05 (State 2: het-del), 0.5 (State 3: neutral), 0.95 (State 4: amp), 0.99 (State 5: high-amp) probabilities. In the segmentation process, the transition and emission parameters are updated by the HMM model (Expectation-Maximization based updates) in the segmentation to ensure that the emission probabilities (and transitions) of the CNV states are tuned exactly to the data.

We included below the plot of mean values of the normal distributions corresponding to 5 copy number states (homozygous deletion, heterozygous deletion, neutral, amplification, high-level amplification) that are derived from the datasets; i.e. single cell GBM, bulk TCGA GBM, bulk TCGA-BRCA and bulk meningioma datasets. As it can be seen from the plot, initial emission probability distributions' mean values for single-cell GBM data are much different than bulk RNA-Seq datasets. In particular, the single cell RNA-seq data are assigned lower expression distributions at the deletion states (states 1, 2) and higher expression distributions at the amplification states (states 4, 5). As the reviewer points out, this may be associated with factors related to the purity of the samples. The impure samples (i.e., bulk RNA-seq samples) may contain normal cells without any somatic alternations and the existence of the normal cells may be buffering the observed expression levels associated with the deletion and amplification states. Another factor that may be driving this behavior is the inherent noise in the gene expression quantifications: The single cell RNA-seq data exhibits higher noise levels such that the deviation of the expression levels may be higher. Therefore, the expression quantiles

associated with the amplification and deletion events may be more extreme when they are estimated from the single cell data.

Based on reviewer's suggestion we updated the Methods "Hidden Markov Model (HMM)" section (**new text line 776-785**).

Updated supplementary Figure 25. Mean values of the normal distributions corresponding to 5 copy number states (homozygous deletion, heterozygous deletion, neutral, amplification, high-level amplification) that are derived from the datasets; i.e. single cell GBM, bulk TCGA GBM, bulk TCGA-BRCA and bulk meningioma datasets. X-axis shows the HMM states and Y-axis shows the mean of the Gaussian emission probability for each state.

c. Are raw expression values used, or are the data normalized somehow? Raw expression values would seem to vary by gene simply because of the strength of the promoter etc.

Authors' Response: The reviewer raises an important point that must be clarified. We agree that the raw expression levels may vary based on factors such as promoter strength. However, CaSpER does not have a specific requirement for the inter-sample distribution of the gene expression levels. For example, it is not necessary to normalize the expression levels so that distributions of gene expressions are matched among samples (e.g. quantile normalization). This is because multiscale decomposition works on a sample by sample basis and does not expect expression levels to be normalized across samples. Secondly, the BAF shift signal profiles are generated for each sample independently for bulk sequencing. Therefore, the BAF shift is computed in terms of the fraction of reads that support the B-alleles. Thus, BAF shift signal normalizes itself at each SNV location and does not require a normalization across samples. One issue is that for single cell datasets, the BAF shift signal is computed initially by pooling the reads from all the cells. If some cells have substantially large number of reads, they

may dominate the BAF shift signal and may potentially create biases. As long as each cell contributes similar number of reads to the BAF shift generation, this should not be a major biasing factor.

For clarification, we briefly describe the preprocessing steps that CaSpER performs. The inputs to the algorithm are the normalized read counts such as FPKM or TPM. Before multiscale decomposition of the signals, we perform two-step gene expression centering. First, **gene centric expression centering is performed.** the gene expression levels are centered around the mid-point. For each gene, the mid-point of expression level is computed among all the cells (or samples in bulk RNA-seq), then the mid-point expression level is subtracted from the expression levels. **Next, cell centric expression centering is performed.** For each cell (or sample), we compute the mid-point of the expression level then we subtract the mid-point expression from the expression levels of all the genes for the corresponding cell. After expression centering steps, **the control normalization** is performed by subtracting reference expression values from the tumor expression values. Note that these data centering steps generally increase the quality of the filtered and segmented data and are also performed by methods such as InferCNV and HoneyBadger.

We added a clarification in methods “Gene expression data processing” section and also discussed the above points in the Discussion Section (**new text line 426-437, 475-485**).

d. BAF signals are interpreted differently for bulk vs single cell data. What is the justification for the different thresholds used?

Authors’ Response: Reviewer raises an important point that we must clarify. The main motivation for selecting different thresholds for bulk and single cell data is that difference in the effect of impurity (i.e., non-cancer cells in the sequenced sample) on single cell and bulk RNA-seq data. More specifically, the BAF shift signal for bulk and single cell RNA-seq data are affected differently by the existence of healthy cell differently in bulk and single cell samples.

Bulk RNA-Seq SNV Filtering and Related Thresholds: In principle, the BAF shift in cancer sample must be computed using the SNVs in the matching normal tissue such as blood or tissue surrounding the tumor. This way, the BAF shift in tumor samples are computed with respect to exactly the normal tissue’s baseline allele frequency:

$$BAF_{shift}(SNV_i) = |BAF_{tumor}(SNV_i) - BAF_{normal}(SNV_i)|$$

where SNV_i denotes the i^{th} SNV and $BAF_{tumor}(SNV_i)$ denotes the alternate allele frequency of SNV_i for the tumor sample. For the SNVs that reside on a copy neutral LOH or a CNV, this value will show deviation from 0. Otherwise it will be distributed around 0.

We cannot compute the above quantity directly because we do not have access to normal RNA-sequencing dataset. For whole exome and whole genome sequencing of tumor, the normal tissue is sequenced regularly (almost as a standard protocol) but RNA-sequencing of matching normal tissue is currently not established as standard procedure. Since we do not have the matching normal sample, we focus on the SNV candidates that are most likely heterozygous (Hence, $BAF_{normal}(SNV_i) = 0.5$) in the healthy tissue. We also have to modify the definition of shift as:

$$BAF_{shift}(SNV_i) = |BAF_{tumor}(SNV_i) - 0.5|$$

As it can be seen, without the normal sample, we just subtract 0.5 from the observed tumor SNV allele frequency (and take absolute value) to compute the BAF shift signal. The most vital

component of this computation is to ensure that the SNVs we use are most likely heterozygous in the normal tissue.

The thresholds we select aim at ensuring that the selected variants are likely heterozygous variants in the normal tissue. If we do not apply these thresholds, we observed that the allele frequency shift signals are non-informative to detect the loss-of-heterozygosity and BAF shifts associated with CNVs. We describe the motivation for selecting these thresholds below:

For bulk samples, there is generally high concentration of healthy cells in the tumor tissues (up to 20-30% as shown in the literature²). Therefore, if an SNV has a high observed allele frequency (>95%) in the tumor, it is very likely that it is a homozygous SNV in the healthy tissue too. The existence of normal tissue in tumor samples will cause a shift in the observed allele frequency of SNVs in the bulk tumor RNA-seq data.

Assume that there is a 1-copy deletion of a segment and there is 30% normal tissue in the bulk RNA-seq sample: 70% of the cells are tumor cells and they contain only 1 copy of the segment; 30% of the cells are normal cells and they contain 2 copies. For an SNV on this segment who is a heterozygous in the normal cells, the observed allele frequency will be:

$$\frac{70 \times 1 + 30 \times 1}{70 \times 1 + 30 \times 2} = 0.769$$

In the above equation, if the impurity is smaller than 30%, the observed allele frequency increases. To be cautious, we choose to act conservatively, and we assume that there may be around 30% impurity in the tumor samples. Thus, we assume that any SNV with allele frequency higher than around 80% is likely a homozygous variant. We therefore exclude the variants that have higher than 80% allele frequencies (Found by rounding 0.769) and smaller than 20% allele frequencies (the other side of the AF spectrum) since these are likely homozygous events in the normal matching tissue. Note that by doing this filtering, we are likely removing variants that inform us about existence of important LOH and CNV events but we still remove these for being conservative and using only the SNVs that are likely heterozygous in the matching normal sample.

In the above computation, we assume for simplicity that there is a clonal deletion of the segment. If the deletion is subclonal (less than 100% of the tumor cells contain the deletion), the observed allele frequencies will be closer to 50% and they will still be selected in the thresholding process. Similarly, for uneven amplification events, the above fraction will decrease. Thus, for these events, the thresholding will capture the useful SNVs appropriately. In addition, for very high amplifications events, allele frequency will be close to 50% (i.e. shift close to 0). In these cases, CaSpER weighs the expression signal much more than the BAF shift signal. Thus, the thresholding affects these events less.

Single Cell RNA-seq SNV Filtering and Related Threshold: SNV selection procedure for single cell samples is motivated mainly by technical factors around detection of the SNVs. Compared to the bulk RNA-seq samples, there is much smaller number of reads from each cell. As described in the text, we tackle this issue by pooling the reads from all the cells and generate a bulk RNA-seq sample from all the cells. Even though we pool the samples, BAFExtract still detects a relatively small number of SNVs that we can use to compute the BAF shift signal.

In addition, for single cell RNA-seq samples, impurity of RNA-seq sample, i.e. fraction of normal cells in the sample, is much smaller (compared to bulk tumor RNA-seq samples) since the normal cell infiltration can be controlled (using for example FACS) and it is generally much smaller. Thus, the single cell RNA-seq data affected much less from the impurity of the samples. Because the sample is almost pure, we cannot use the observed SNV allele frequencies directly to infer a maximum allele frequency threshold (similar to the threshold we derived in the above equation) for selecting SNVs that are most likely heterozygous in matching normal cells. We therefore include all the SNVs in the analysis except the SNVs whose alternate allele frequencies are smaller than 20%. The reasoning for this lower allele frequency threshold is that these SNVs may be manifestations of technical artefacts such as PCR duplications³. This threshold value is a conservative value that is selected by trial-and-error to optimize the quality of the observed BAF shift signal and its consistency with the expression signal. We also included below figure to illustrate the spectrum of allele frequencies of SNVs that are used for bulk and single cell RNA-seq data.

Updated supplementary Figure 21. The illustration of the observed allele frequency ranges (or spectrums) that are used to select the SNVs for computing the BAF shift signal. For bulk RNA-seq, the SNVs whose observed allele frequencies are in 20%-80% range (Illustrated with red box) are selected for computing the BAF shift signal. For single cell RNA-seq, the SNVs whose observed allele frequencies are in the range 20%-100% (Illustrated with blue box) are used for **computing the BAF shift signal**.

We updated in methods “*BAF generation, SNV filtering, and related thresholds*” section (**new text line 505-582**).

2) Conclusions are drawn that do not seem well-supported by data. For example:

a. “The smoothed signal distribution shows that data does not contain many amplification events (Figure 4B)”. I do not see this in the figure.

Authors’ Response: We agree with the reviewer that we did not clearly explain the observation with respect to the data. Figure 4B shows the median expression signal distribution of the segments **before the integration of the BAF shift signal** as stratified by the assigned HMM states (indicated by the colors). The segments are assigned 5 different states by the HMM: 1: homozygous deletion, 2: heterozygous deletion, 3: neutral, 4: one-copy amplification, 5: multi-copy amplification, indicating high-amplifications. It can be seen from the distribution that there

are no segments with assigned state 5 (high-amplifications). Secondly, after BAF shift integration, the most of the segments whose assigned states are 4 (one-copy amplification) are assigned to neutral CNV calls because they are not accompanied by a BAF shift. Because of the above observations, we originally stated that data do not contain many amplification events.

We now realize that this figure may create unnecessary confusion for the readers as it is generated from intermediate data and we removed the figure and the sentence that created confusion.

b. "CaSpER achieves 95% TPR and 0.3% FPR for detecting deletion events (Figure 4F)." All events, or in a certain size regime?

Authors' Response: We apologize that we did not make this clear. Initially, in this sentence "CaSpER achieves 95% TPR and 0.3% FPR for detecting deletion events (Figure 4F)." We reported TPR and FPR values **for the large-scale events only**. To clarify and to make the results more comprehensive, we now also calculated TPR and FPR values for gene level CNV call summarizations. We have now revised the manuscript to clarify TPR and FPR calculation and included the gene level TPR and FPR calculations (**in new text line 202, 839-868**) (**Methods "Performance Assessment" section**).

c. Why do the authors claim the BAF shift in chromosome 14 for MGH31 is subclonal? No rationale is provided.

Authors' Response: We thank the reviewer for pointing out the unclear statement that might cause confusions. We agree with the reviewer that the BAF shift subclonality is difficult to anticipate from the Figure 5B. The basic rationale for this claim was the visual inspection of the BAF shift signal on chromosome 14. In the current form of the figure, it is fairly hard to visually assess the shift. We included below an enlarged version of the BAF shift signal on chromosome 14. It can be seen that the BAF shift wiggles around 0.2 and 0.3 (away from 0.5), which indicates that this signal stems from subclonal cell populations with mixture of different BAF shift levels on this chromosome. The reason for this claim is that if there was a clonal event, the shift generally stays around 0.5 (See below enlarged figure for chromosome 7). An alternative explanation for this observation could be that the infiltration of normal cells but since the BAF shifts signal is generated from the single cell RNA-seq data there is none or very small infiltrating normal cells. Therefore, this deviation cannot be resulting from the infiltrating normal cells in the tumor. After evaluating the figure again, we also observed that there are very small number of SNVs in this figure. In order to avoid further confusion from this visual observation, we removed the statement from the manuscript.

Figure 1. BAF shift signal on chromosome 14 for patient MGH31.

d. For MGH31, the overall decomposition doesn't seem to make sense. For example, I see 9p loss both among 1p amp and 13q loss cells, and other 1p amp and 13q loss cells that do not have 9p loss. This doesn't seem to accord with an evolutionary tree, or with the results indicated in Figure 5E. Indeed, the organization of the data in Figure 5C make assessments of correlation and mutual exclusivity difficult to assess. Figure 5C has colors at the left that are not described; I imagine these refer to subclone assignments but it is not clear how those assignments were made. Figure 5E shows edges for co-occurring events, but the significance levels associated with an edge of a certain width is not clear—no legend is provided.

Authors' Response: We apologize that we did not make this clear. The colors on the left of Figure 5C indicates the 5 patients from which the cells originate from, namely MGH26, MGH28, MGH29, MGH30, and MGH31. In the figure, we have failed to convey that these colors do not represent subclone assignments. We have now added the patient identifiers on the colors to clarify this point.

We also updated Figure 5F (previously Figure 5E) and included the explanation of legend width for Figure 5F.

3) How do the TPRs and FPRs for CaSpER compare to existing methods? Note that meningiomas as a test set is not ideal: it tends to have few CNVs, and few focal CNVs. CaSpER would better be tested in a more complex tumor type, such as breast or ovarian cancer. In particular, GBM and meningioma tend to be diploid, and it will be important to test CaSpER in cancers with more frequent tetraploidy/genome doubling events

Authors' Response: We appreciate the reviewer's input. Based on this suggestion, we have now included TCGA bulk breast cancer RNA-Seq data for performance assessment of

CaSpER. TCGA contains around ~1000 breast cancer RNA-seq samples, therefore we randomly selected 150 samples from the cohort. TCGA Ovarian cancer RNA-Seq cohort did not include any control normal blood samples therefore we focused on breast cancer as a representative of more complex tumor type. We also tested our method on bulk RNA-Seq dataset from the TCGA-GBM cohort where we randomly selected 75 samples.

Regarding the accessibility of genome to RNA-seq based CNV calling. Before jumping into the accuracy benchmarks, we would like to highlight an important aspect of CNV calling using RNA-seq data. RNA-seq data is mainly concentrated on the genic regions on the genome. Therefore, we can only identify the CNVs that overlap with a reasonable number of genes. We have performed some simulations and hypothesize that RNA-seq based CNV calling tools cannot access CNVs shorter than 1 megabases reasonably because these CNVs cover a small number of genes (7.54 genes on average) and it is hard to distinguish the signal on these very short regions from the background. Please see Response to “Reviewer 3: General Remarks” below for a more detailed discussion on this.

Regarding genome doubling. The reviewer points out a very important element that is directly related to CNV calling. CaSpER (and many other CNV detection tools such as HoneyBADGER) do not assign absolute copy numbers. Assignment of absolute copy numbers and detection of genome doubling events are very challenging problems even when DNA-seq data is used with matching controls. For instance, the authors of ABSOLUTE⁴, which is the most popular tool for assignment of absolute copy numbers in DNA-seq data, highlight the challenging nature of the estimation of absolute copy numbers and detection of genome doubling and tetraploidy events:

“...Accurate calibration of both the SCNA-fit [Somatic Copy Number Alternation fitting] and karyotype models to the true level of certainty implied by the data would allow for assignment of probabilities to each candidate solution; we do not believe that the models we have presented here sufficiently capture the complexity of cancer genomes to allow for such interpretations. Even with manual review, analysis with ABSOLUTE may occasionally result in incorrect interpretations, for example genome-doubling without subsequent detectable gains or losses would result in a solution implying half the true ploidy value, which in some cases may correspond to a plausible karyotype model...”, from ⁴.

ABSOLUTE uses karyotype models as reference datasets to compare a new dataset against so that the new dataset can be classified in terms of genome doubling events. Even in the existence of a comprehensive karyotype model database, the authors are limited by the fact that cancer genomes are utterly complex. Generally speaking, it is not possible to distinguish a genome doubling event and a genome quadrupling event without making any heuristic assumptions, as the last sentence implies in the above paragraph.

The main reason for the hardness of absolute copy number assignment is that the RNA-seq data needs to be normalized using RPKM based normalization and after this it is not possible to directly perform absolute quantification of copy number. If we have proper controls, such as spike-ins that are frequently generated with RNA-seq data, the genome doubling may potentially be detected. (For more details on this point, please see Comment 3 of Reviewer 3) We have updated the Discussion Section of the manuscript to highlight the above points (**new text line 415-425**)

We calculated TPRs and FPRs on gene and large-scale CNV assignments (See **Methods Section line 839-869** for details of performance assessment, also the Response to the

Comment 2 of Reviewer 3). (Table 1-2). As it is seen in Table 1-2, CaSpER outperforms HoneyBADGER. One main observation is that HoneyBADGER is extremely specific and not very sensitive for detection of CNV events. HoneyBADGER clusters expression and lesser allele frequency (LAF) data separately. Then, for each expression and LAF cluster, it calculates the mean the expression and LAF signal. Then an HMM is used to segment the mean expression signal. Unlike CaSpER, joint analyses and integration of the LAF and expression is not performed. The authors claim “Gene expression-based approach of HoneyBADGER is able to identify whole-chromosome-level alterations with high accuracy and precision. The allele-based approach can achieve high accuracy and precision even for small deletions. Allele-based model is also much more sensitive at identifying deletions than the expression-based model alone.”

		Large Scale Level		Gene Level	
		HoneyBADGER	CaSpER	HoneyBADGER	CaSpER
TPR	Deletion	6%	85%	5.9%	77.6%
	Amplification	25%	71%	22%	57.5%
FPR	Deletion	0.07%	3%	0.27%	3.5%
	Amplification	0.14%	1.5%	0.23%	2%

Updated Supplementary Table 1. TPR and FPR values for bulk RNA-Seq TCGA-GBM

		Large Scale Level		Gene Level	
		HoneyBADGER	CaSpER	HoneyBADGER	CaSpER
TPR	Deletion	6.4%	79%	3.2%	62.8%
	Amplification	8%	60.3%	10.5%	56.6%
FPR	Deletion	0.46%	7.7%	0.75%	6%
	Amplification	0.1%	3.8%	0.09%	7%

Updated Supplementary Table 2. TPR and FPR values for bulk RNA-Seq TCGA-BRCA

When we applied HoneyBADGER on bulk TCGA-GBM RNA-Seq data, expression-based model of HoneyBADGER discovered only two events; deletion in **chr22:19941607-50783663** and amplification in **chr7:1815793-151080866**. Allele based model of HoneyBADGER discovered six focal deletion events; **chr1:109745618:109795026**, **chr6: 3255327: 3285286**, **chr9:134384277:134395628**, **chr14:102467847:103059220**, **chr20: 60904853: 60909060**, **chr22: 26862153: 26895337**.

When we applied HoneyBADGER on bulk TCGA-BRCA RNA-Seq data, expression-based model of HoneyBADGER discovered amplification events in **chr1:148808181:248849517**, **chr17: 19770829:82945922**, **chr19:531712: 58558960**, **chr20:296968:64287821** and deletion events in **chr16:31489471:89968060**, **chr11:87037844:117204782**, **chr6:107867756:170584692**, **chr18:158383:214629**. Allele based model of HoneyBADGER discovered the following focal deletion events;

chr	start	end
chr3	88189341	88199179
chr7	149427409	149522366
chr9	130670652	130710726
chr10	135076523	135180430
chr14	24910210	25101589
chr16	23521643	23563501
chr22	50277835	50317049

From the above results, it can be seen that HoneyBADGER is not suitable for detection of CNVs from the bulk RNA-seq datasets. To compare the visualizations of the methods, we also include the expression plots of HoneyBADGER, inferCNV and CaSpER for TCGA-GBM data. All the methods generate the visualizations that can be compared. It should be noted, however, that CaSpER generates a range of other visuals that can be useful for more comprehensive characterization of the RNA-seq data and these visuals are not available from other methods. We believe that CaSpER can provide novel insight into analyzing the CNV architecture of single cell and bulk RNA-seq samples compared to other methods.

We have updated the the manuscript to highlight the above points (**new text line 318-328**).

Updated Supplementary Figure 16. Expression plots of HoneyBADGER, inferCNV and CaSpER for TCGA-GBM data.

HONEYBADGER

inferCNV

CaSpER

Updated Supplementary Figure 17. Expression plot of HoneyBADGER, inferCNV and CaSpER for TCGA-BRCA data.

Updated Supplementary Figure 18. Large scale event summary plots of HoneyBADGER (A), CaSpER (B) and Genotyping (C) for TCGA-GBM data. Rows correspond to samples whereas columns correspond to chromosome arms.

Updated Supplementary Figure 19. Large scale event summary plots of HoneyBADGER (A), CaSpER (B) and Genotyping (C) for TCGA-BRCA data. Rows correspond to samples whereas columns correspond to chromosome arms. Red represents amplification whereas blue represents deletion.

4) The use of pooled BAF data seems to counter the whole point of the method to evaluate single-cell changes. If this needs to be done, why not use pooled data to assess a baseline shared by most cells, and then assess expression data explicitly to detect deviations from that baseline?

Authors' Response: The reviewer is proposing a very good and insightful idea. While we were studying the usage of BAF shift signal from single cell RNA-seq samples, one of the major obstacles was having enough reads to detect SNVs that can be used to produce informative BAF shift signal. As the number of reads (and cells) decrease, we observed a fast decay in the quality of the BAF shift signal. From one single cell study around ~200 SNVs are detected using BAFExtract and BAF the signal is very sparse. We therefore face a tradeoff: Accurate detection of informative SNVs versus the accurate detection of the subclonal BAF shift events. To include more informative SNVs, we chose to use pool reads from all the cells. After pooling the reads, we are able to detect around 140k SNVs per patient. As the single cell RNA-sequencing technologies progress and generate more data for each cell, we foresee that the BAF shift estimation can be performed in each cell independently. This way, every cell can be treated as its own sample. And BAF shift generation does not need to be done by pooling. Finally, we also acknowledge that pooling is certainly not the only way to generate the BAF shift signals but we are aiming to include as many informative SNVs as possible.

Regarding the idea of using pooled BAFs as baseline. We assume that the reviewer suggests using the pooled BAF shift signal to detect regions with no CNV events because the pooled signal is coming from all the cells and it may not represent events that can be found in individual cells.

In fact, this idea is very similar to what CaSpER already does. CaSpER analyzes the pooled BAF shift signal distribution **to identify the baseline events where no BAF shift is observed**. These events indicate locations where there are likely no allelic shifts. After these regions are identified, CaSpER uses these regions while assigning the final CNV calls after HMM segmentation. We found that this to be the most conservatively useful way to make use of the pooled BAF shift signal profiles.

It is worth noting that using the whole pooled BAFs directly as baseline may cause issues because **we can clearly see BAF shift signal at the clonal (i.e., events happening at majority of the cells) and some of the subclonal CNV events**. If we used the pooled BAF shift signal as baseline that indicates no events, we would have to assume that these events are baseline and mark them as neutral events. In fact, the main motivation behind using BAF shift signal jointly with expression signal is the observation that the pooled BAF shift signal has very good concordance with expression signal from individual cells and it helps with identifying and elucidating the CNV events. Rather, the expression signal from each cell is segmented by the HMM without regard to the pooled BAF shift signal. We then evaluate the pooled BAF shift signal on the segments detected in expression signal and final CNV calls are assigned on each segment. This way, the pooled BAF shift signal is used as auxiliary and supporting information to identify CNVs.

Minor comments:

1) *The introduction is too long.*

Authors' Response: We appreciate the reviewer's input. We have made the introduction long to ensure that the method is motivated and the previous literature is covered well. Based on this suggestion, we have removed portions of the introduction.

2) *In general, the figures are difficult to understand and not well-labeled. For example, what does the y-axis refer to in 3b-c? What do the dots in 3E represent? The existing labels are hard to decipher*

Authors' Response: We apologize sincerely for lack of clarity in the figures labels and we appreciate the reviewer's careful reading. We have now reviewed all the figure legends and made sure that they are well-labeled, including Figure 4 (previously Figure 3).

Reviewer #3 (Remarks to the Author):

Overview

Beyond simply quantifying expression levels, single-cell RNA-seq data contain other potentially important signals such as from copy-number alterations that can be elucidated with the appropriate bioinformatics tools. Harmanci et al. present a computational approach called CaSpER that strives to enable identification, visualization, and integrative analysis of focal and large-scale CNVs using either bulk or single-cell RNA-seq data. Although the authors acknowledge previous methods for CNV inference using single-cell RNA-seq data, including inferCNV and HoneyBADGER, comparisons with these previously published approaches are not provided. The suitability of CaSpER for identifying smaller focal alterations, multi-copy amplifications, small subclonal alterations (affecting a small percentage of cells) and clonal alterations (affecting 100% of cells) remain to be convincingly shown, as benchmarks have focused on chromosome-arm-level single-copy deletions and amplifications affecting large subclones. Similarly, the sensitivity and accuracy of the method at different resolutions (e.g. for different CNV sizes and clonality) remain unclear. As such, the manuscript is not suitable for publication at this stage.

Authors' Response: We thank the reviewer for their insightful comments about the manuscript. We have updated the manuscript to include comparisons with HoneyBADGER and inferCNV and accuracy estimations with respect to event size and clonality. We show that HoneyBADGER is very specific and has lower sensitivity. It is also designed primarily for single cell RNA-sequencing datasets. InferCNV is primarily for visualization of the copy number events and does not provide a set of CNV events. We tested CaSpER with simulated data that has controlled introduction of events with differing amount of clonalities and sizes. We evaluate the sensitivity of CaSpER for these different types of events.

An important factor in identifying CNVs from RNA-seq (using CaSpER and HoneyBADGER) data is that **the methods work at the level of genes, which means that the smallest resolution for segment coordinates will be at the gene boundaries. In other words, the regions in the genome that do not contain any genes will not be accessible for calling CNVs on them.** In order to assess the accessibility of CNVs by RNA-Sequencing based methods, we randomly generated 50,000 regions in varying sizes; 100 kilobases (kB), 1 megabases (mb), 5 mb, 10 mb, 50 mb, 100 mb and computed the number of genes that overlap with them (See below figures). **Interestingly, we found that for single cell RNA-Seq datasets which cover on average 6000 genes genomewide, we found on average 0.33 genes in 100 kb segments, 2.12 genes in 1 mb segments, 10.08 genes in 5 mb segments, 20.09 genes in 10 mb segments, 102 genes in 50 mb segments, and 168 genes in 100mb.**

The bulk RNA-seq datasets cover much larger portion of the genome as they measure all 20,000 protein coding genes. **In bulk RNA-Seq dataset 100 Kb segments cover on average 1.15 genes, 1 mb segments cover 7.54 genes, 5 Mb segments 35.09 genes, 10 mb segments cover 68.9 genes, 50 mb segments cover 343.29 genes, and 100 mb segments cover 534.98 genes.** It can be seen from these statistics that most of the segments that are

shorter than 1 megabases will be very hard to detect (unless in gene dense regions) because they cover very few genes. We believe that these should be kept in mind while interpreting the accuracy results.

This has several implications: We cannot identify the CNV segments that are solely in the intergenic space (no gene overlap) and solely in the introns of genes because there is very little or no RNA-seq signal in them. Therefore, we must consider the regions in the genome without any genes in them as ***inaccessible for RNA-seq based CNV calling methods*** because the methods cannot identify CNVs there. This impacts the way that the accuracy is estimated for methods like CaSpER and HoneyBADGER because inclusion of inaccessible regions in sensitivity estimation will adversely (and unfairly) decrease the sensitivity of the methods. It is worth noting that this is similar to detection of CNVs using the whole exome sequencing (WES) data except that WES can quantify the coverage on the exons so it may be able to detect breakpoints within genes and introns. But, similar to RNA-seq, WES cannot be used to detect the CNVs primarily in the intergenic space.

With these considerations in mind, we now incorporated the accuracy estimates at the gene level and at the segment level information to characterize the accuracy of CaSpER more coherently. The segment level sensitivity, however, must be carefully interpreted with above considerations in mind.

Updated Supplementary Figure 6. Accessibility of CNVs by bulk RNA-Sequencing based methods. X-axis corresponds to regions in varying sizes; 100 kilobases (kB), 1 megabases (mb), 5 mb, 10 mb, 50 mb, 100 mb and Y-axis corresponds to the number of genes that overlap with these 50000 simulated regions with varying sizes.

Updated Supplementary Figure 7. Accessibility of CNVs by single-cell RNA-Sequencing based methods. X-axis corresponds to regions in varying sizes; 100 kilobases (kB), 1 megabases (mb), 5 mb, 10 mb, 50 mb, 100 mb and Y-axis corresponds to the number of genes that overlap with these 50000 simulated regions with varying sizes.

Major comments

1. The authors state that "CaSpER performs well in accuracy compared to gold-standard SNP genotyping arrays." How does CaSpER compare to other sequencing-based CNV inference approaches, in particular to other single-cell RNA-seq CNV inference approaches such as inferCNV and HoneyBADGER?

Authors' Response: We agree that a detailed comparison will highlight how CaSpER performs compared to the other methods. To perform a thorough comparison, we downloaded TCGA bulk breast cancer RNA-Seq data for performance assessment of CaSpER and other methods. TCGA contains around ~1000 breast cancer RNA-seq samples, therefore we randomly selected 150 samples from the cohort. We focused on breast cancer as a representative of a complex tumor type. We also tested our method on bulk RNA-Seq dataset from the TCGA-GBM cohort.

We next updated the accuracy metrics. While we initially compute the large-scale accuracies, we have since updated to compute gene-based accuracies, and segment-based accuracies. Please refer to our response to the 2nd comment (and revised Methods Section) for the new accuracy metrics (**Methods Section line 839-869** for details of performance assessment).

We calculated TPRs and FPRs on gene and large-scale CNV assignments (**Methods Section line 839-869** for details of performance assessment). (Table 1-2). As it is seen in Table 1-2, CaSpER outperforms HoneyBADGER. One main observation is that HoneyBADGER is extremely specific and not very sensitive for detection of CNV events. HoneyBADGER clusters expression and lesser allele frequency (LAF) data separately. Then, for each expression and LAF cluster, it calculates the mean the expression and LAF signal. Then an HMM is used to segment the mean expression signal. Unlike CaSpER, joint analyses and integration of the LAF

and expression is not performed. The authors claim “Gene expression-based approach of HoneyBADGER is able to identify whole-chromosome-level alterations with high accuracy and precision. The allele-based approach can achieve high accuracy and precision even for small deletions. Allele-based model is also much more sensitive at identifying deletions than the expression-based model alone.”

When we applied HoneyBADGER on bulk TCGA-GBM RNA-Seq data, expression-based model of HoneyBADGER discovered only two events; deletion in **chr22:19941607-50783663** and amplification in **chr7:1815793-151080866**. Allele based model of HoneyBADGER discovered six focal deletion events; **chr1:109745618:109795026**, **chr6: 3255327: 3285286**, **chr9:134384277:134395628**, **chr14:102467847:103059220**, **chr20: 60904853: 60909060**, **chr22: 26862153: 26895337**.

		Large Scale Level		Gene Level	
		HoneyBADGER	CaSpER	HoneyBADGER	CaSpER
TPR	Deletion	6%	85%	5.9%	77.6%
	Amplification	25%	71%	22%	57.5%
FPR	Deletion	0.07%	3%	0.27%	3.5%
	Amplification	0.14%	1.5%	0.23%	2%

Updated Supplementary Table 1. TPR and FPR values for bulk RNA-Seq TCGA-GBM

		Large Scale Level		Gene Level	
		HoneyBADGER	CaSpER	HoneyBADGER	CaSpER
TPR	Deletion	6.4%	79%	3.2%	62.8%
	Amplification	8%	60.3%	10.5%	56.6%
FPR	Deletion	0.46%	7.7%	0.75%	6%
	Amplification	0.1%	3.8%	0.09%	7%

Updated Supplementary Table 2. TPR and FPR values for bulk RNA-Seq TCGA-BRCA

When we applied HoneyBADGER on bulk TCGA-BRCA RNA-Seq data, expression-based model of HoneyBADGER discovered amplification events in **chr1:148808181:248849517**, **chr17: 19770829:82945922**, **chr19:531712: 58558960**, **chr20:296968:64287821** and deletion events in **chr16:31489471:89968060**, **chr11:87037844:117204782**, **chr6:107867756:170584692**, **chr18:158383:214629**. Allele based model of HoneyBADGER discovered the following focal deletion events;

chr	start	end
chr3	88189341	88199179
chr7	149427409	149522366
chr9	130670652	130710726
chr10	135076523	135180430
chr14	24910210	25101589
chr16	23521643	23563501
chr22	50277835	50317049

From the above results, it can be seen that HoneyBADGER is not suitable for detection of CNVs from the bulk RNA-seq datasets. To compare the visualizations of the methods, we also include the expression plots of HoneyBADGER, infeCNV and CaSpER for TCGA-GBM data. All the methods generate the visualizations that can be compared. It should be noted, however, that CaSpER generates a range of other visuals that can be useful for more comprehensive characterization of the RNA-seq data and these visuals are not available from other methods. We believe that CaSpER can provide novel insight into analyzing the CNV architecture of single cell and bulk RNA-seq samples compared to other methods.

We have updated the manuscript to highlight the above points (**new text line 318-328**).

Updated Supplementary Figure 16. Expression plots of HoneyBADGER, inferCNV and CaSpER for TCGA-GBM data.

HONEYBADGER

inferCNV

CaSpER

Updated Supplementary Figure 17. Expression plot of HoneyBADGER, inferCNV and CaSpER for TCGA-BRCA data.

Updated Supplementary Figure 18. Large scale event summary plots of HoneyBADGER (A), CaSpER (B) and Genotyping (C) for TCGA-GBM data. Rows correspond to samples whereas columns correspond to chromosome arms.

Updated Supplementary Figure 19. Large scale event summary plots of HoneyBADGER (A), CaSpER (B) and Genotyping (C) for TCGA-BRCA data. Rows correspond to samples whereas columns correspond to chromosome arms. Red represents amplification whereas blue represents deletion.

2. "The authors measure the accuracy of CaSpER by comparing identified CNVs with those called from genotyping arrays such that "the true positive rate is the percentage of large-scale CNV events who are correctly identified by CaSpER while the false positive rate is the percentage of falsely rejected true CNV events." This is incorrect. The false positive rate is the percentage of falsely identified CNV events (ie. CaSpER identifies a CNV that the genotyping array does not). Quantification of the false positive rate by this definition is currently lacking and should be provided. The false negative rate is the percentage of falsely rejected true CNV events (ie. CaSpER does not identify a CNV but the genotyping array says a CNV is present). Still, it is unclear whether this accuracy assessment is done on the CNV level or base-pair level. For example, if a true CNV affects chromosome 1 bases 1 to 5e6, but CaSpER calls a CNV affecting chromosome 1 bases 1 to 6e6, is this considered a true positive on chromosome 1? Or are the base pair resolution of the CNV also considered?"

Authors' Response: We agree with the reviewer that the performance metrics are not defined properly. We have updated the manuscript to include performance metrics (**new text line 838-869**).

We have now corrected this sentence:

"the true positive rate is the percentage of large-scale CNV events who are correctly identified by CaSpER while the false positive rate is the percentage of falsely rejected true CNV events."

as following:

"the true positive rate is the fraction of the genotyping array CNVs which are detected by CaSpER and the false positive rate is the fraction of unaltered genotyping array segments which overlap with CaSpER CNVs"

In other words, TPR measures how well CaSpER can correctly detect the CNV

We describe the performance metrics below in detail.

Performance Assessment.

Large scale (chromosome arm) and gene level accuracy metrics. We assessed the performance of CaSpER's large scale (i.e., chromosome arm) and gene level CNV calls by comparing these CNV calls with the ground truth calls generated from genotyping data. In this comparison, we use the true positive rate (TPR) and false positive rate (FPR) as the main metrics. TPR is the percentage of large-scale (or gene level) CNV calls that are correctly identified by CaSpER while the FPR is the fraction of unaltered chromosome arms and genes that are predicted as having altered CNV calls by CaSpER. (For more details, please see Response to Reviewer 1's Comment #1). To clarify these definitions, we summarized them in the following table:

		True events (Genotyping Array)	
		Alteration Event (Amp. or Del.)	No Alteration Event (Neutral)
Predicted Events (CaSpER, HoneyBADGER)	Predicted Alteration Event	TP	FP
	Predicted no Alteration Event	FN	TN

Based on these definitions, the true positive rate and false positive rate is calculated as shown below.

$$TPR = TP/(TP+FN)$$

$$FPR = FP/(FP+TN)$$

Segment based accuracy metrics. When computing the accuracy at the segment level, we compare the segments from each method with the segments in the ground truth data set, i.e. genotyping array segments. For quantifying the segment-based accuracies, we use TPR and positive predictive value (or precision). Before defining these, we first illustrate the false

negative, false positive, and true positive portions of two segments as in the above figure. In this figure, X represents the true CNV segment (from genotyping array) and Y represents the segment detected by CaSpER.

1. **FN** denotes the length of the true segment (in bps) that do not overlap with the detected segment.
2. **TP** denotes the length of the detected segment (in bps) that overlaps with the true segment.
3. **FP** denotes the length of the detected segment (in bps) that does not overlap with the true segment.

In order to compute the accuracy metrics over all the segments, we use the weighted average of the TP, FP, and FN computed for each segment by appropriate length. Let us assume that the final CaSpER CNV segments are denoted by $Y_i: i = 1, \dots, K$. We define Precision (or Positive predictive value -- PPV) and sensitivity to measure the segment-based accuracy of CNVs using the following fractions:

$$PPV (precision) = \frac{\sum_i TP(Y_i)}{\sum_i |Y_i|}$$

$$TPR (sensitivity) = \frac{\sum_i TP(Y_i)}{\sum_j |X_j|}$$

where $|X_j|$ denotes the length of X_j in base pairs. From the above equations, PPV and TPR are computed by taking the lengths into consideration. Finally, based on the above definitions, the segment-based accuracy metrics using $\gamma > 6$:

For Deletions PPV: 70%; TPR 76%

For Amplifications PPV: 68.5%; TPR 66%

Updated Supplementary Figure 4. TPR and PPV values for deletion events estimated from bulk TCGA-GBM with varying γ thresholds and varying segment mean thresholds for genotyping array (0.1, 0.2 and 0.3). Plot shows segment based performance which is assessed using genotyping array.

Updated Supplementary Figure 5. TPR and PPV values for amplification events estimated from bulk TCGA-GBM with varying γ thresholds and varying segment mean thresholds for genotyping array (0.1, 0.2 and 0.3). Plot shows segment based performance which is assessed using genotyping array.

We have updated the Methods “Performance Assessment” section (**new text line 838-869**) and Results section to include the above definitions and corrections.

3. “The authors apply CaSpER to single-cell GBM RNA-seq data from 5 patients. A small number of cells from one of these patients, MGH31, was shown in the original publication to be normal immune cells. It is unclear from Figure 5 whether these normal cells were properly identified by CaSpER, as they should have harbor no CNVs. Please comment on this discrepancy.

Similarly, the authors identify a number of new CNVs in the GBM data, previously undetected in the original publication. Please comment on this discrepancy. What are the sizes of these alterations? Were they missed previously due to the resolution limitations of previous methods? How do we know they are not false positives? Are there any validations available for these new identified CNVs such as cytogenetics or bulk DNA sequencing?”

Authors’ Response: We thank the reviewer for the insightful comment. Based on the reviewer’s suggestion, we performed PCA analysis on the expression levels of MGH31’s cells so that we identify the cells whose expression levels are outliers. This plot reveals 8 normal (oligodendrocytes) cells clearly separating from other cells:

Updated Supplementary Figure 11. PCA plot shows the separation of 8 normal oligodendrocytes from other tumor cells. Cells within blue circle corresponds to normal cells.

We next focused on these cells and evaluated their CNV calls. The heatmap below shows the CNVs detected in each cell. It can be seen that the normal cells show a very different CNV profile compared to the other cells in the sample. In total, there are 8 large-scale events in these cells. In addition, the CNVs do not exhibit the general co-occurrence pattern of CNVs in other cells. While CaSpER does not explicitly identify the infiltrating healthy cells, the users can identify these cells in the visualization.

Updated Figure 3C. Clustering of large scale events generated by CaSpER in patient MGH31. Normal cells are clustered separately with a different CNV profile. Cells within the red rectangle corresponds to normal cells. Rows correspond to cells whereas columns correspond to chromosome arms.

Regarding the novelty of CNVs in GBM data, the authors of the original study primarily make use of inferCNV plots to detect patterns of CNV co-occurrence and exclusivity. Since inferCNV enables a visual inspection of the data and does not perform analytical detection of the CNVs, it is likely that the authors might have missed the other visually less obvious events. In fact, the authors also point out that the “panoramic view” of chromosomal landscape does not capture all the information in the signal but nevertheless it reveals some visually obvious patterns:

From Patel et al⁵: *“Gain of chromosome 7 and loss of chromosome 10, the two most common genetic alterations in glioblastoma (20), were consistently inferred in every tumor cell. Chromosomal aberrations were relatively consistent within tumors, with the exception that MGH31 appears to contain two genetic clones with discordant copy-number changes on chromosomes 5, 13, and 14. Although these data suggest largescale intratumoral genetic homogeneity, we recognize that heterogeneity generated by focal alterations and point mutations will*

be grossly underappreciated using this method. **Nevertheless, such panoramic analysis of chromosomal landscapes effectively separated normal from malignant cells.**"

Validation of exclusion patterns. Unfortunately, we do not have access to the cells that were analyzed (or the bulk DNA-sequencing data) and we cannot perform experimental wet-lab

validation of the observed patterns. However, as supportive evidence of the existence of the reported patterns and to clarify the patterns generated by inferCNV and by CaSpER, we included below the heatmap plots of the expression signal as they are generated by both methods. We believe that these patterns can be confirmed by visual inspection.

Updated Supplementary Figure 14. Heatmap plots of the expression signal generated by CaSpER and inferCNV are plotted. Rows correspond to cells whereas columns correspond to genes ordered by chromosomal locations. Rows are ordered by the significant mutually exclusive event pairs identified by CaSpER.

It can be seen that the patterns that we are reporting can be seen in inferCNV plot but they are not as visually catchy. The computational analysis, however, enables a more coherent detection of these patterns compared to a panoramic analysis.

We have updated the Results section (**new text line 273-276, 292-302**) and included the above points.

4. The authors suggest that CaSpER is suitable to identifying focal alterations, but most of the alterations identified are chromosome to chromosome-arm level.

In Figure 6, the authors do show that CaSpER is able to identify a focal amplification in PDGFRA using small-scale lengths that is consistent with gold-standard genotyping. However, in the same figure, gold-standard genotyping appears to identify another focal amplification of comparable size (around base pair $1e8$) that CaSpER fails to identify.

Authors' Response: We thank the reviewer for touching on an important point. To analyze the CNV detection performance of CaSpER, we use the segments detected by CaSpER in the bulk TCGA-GBM bulk RNA-seq data and we computed the accuracy of CNV calls for different segment sizes. Specifically, we extracted sets of focal segments with respect to the segment length, using segments in 5 length ranges: **Shorter than 1mb, between 1mb-5mb, between 5mb-10mb, between 10mb-50mb, between 50mb-100mb**. This way, we evaluate the accuracy of focal segments at different length thresholds. We computed the segment-based PPV for the segments in these length ranges. The genotyping array data is used as ground truth while computing segment-based accuracies. These accuracies are plotted in the below figure. In general, we observed that the focal deletions and amplifications are detected with lower PPV (As low as 50% PPV) compared to the longer the segments (As high as 95% PPV).

Updated Supplementary Figure 8. Segment based accuracy measures for bulk TCGA-GBM with varying segment size intervals (<1mb, 1mb-5mb, 5mb-10mb, 10mb-50mb, 50mb-100mb). Performance is assessed using genotyping array.

We have updated the Results section (**new text line 239-248**) and included the above points.

Regarding Visualization of PDGFRA and Neighboring Region. We agree with the reviewer that this example may be confusing. We have investigated why the region neighboring PDGFRA is not detected by CaSpER. First, while genotyping array signal shows relatively strong signal, we saw that the RNA-seq expression signal is very noisy before and after the signal smoothing. In comparison, PDGFRA region has a much more distinct and high expression signal that is easily detected by CaSpER after smoothing (See figures below).

In addition, even the genotyping array signal does not show a high amplification in the neighboring region as PDGFRA region and the signal increase manifests on a much shorter

length scale: The neighboring region has an amplification level of (~1.5) and event size ~3MB; to compare, PDGFRA region's expression level is around 3 and event size is approximately 9MB (See figures below).

Figure 2. RNA-Seq signal around the PDGFRA gene locus

Figure 3. RNA-Seq signal in the region neighboring the PDGFRA gene locus

Consequently, the non-detection of the neighboring region stems from the fact that the RNA-seq expression signal is much noisier in the neighboring region (compared to PDGFRA gene) and does not show a good amplification signal as genotyping array data (See above RNA-Seq expression signal figures). We believe that one of the confusing factors might have been that we placed the genotyping array data in the Figure to the top row and we placed the segmented multiscale decomposition of expression signal right below the genotyping array plot. This makes it look like the decompositions are computed from the first row, but this is not the case. The first row (Genotyping array data) is placed for comparison.

We hope that this clarifies the issue of detection of the focal events around PDGFRA region. Regardless of our explanation we feel that this figure does not convey the message we meant to present. Therefore, we decided to replace this region with another sample and replaced the figure so that it conveys our message clearly.

Figure 4. Focal amplification events detected by genotyping events are shown.

5. Similarly, the authors note how "lower clonality rates in meningiomas lead to better deletion CNV event identification. Similarly, high clonality rates in GBM tumors lower the detection accuracy of low-level amplification events." Please provide a more quantitative basis for this observation and quantify CaSpER's performance as a function of clonality.

As CaSpER uses B-allele frequencies from RNA-seq reads to identify CNV events does not require heterozygous variants to be called using a normal reference, it is unclear how B-allele frequencies can be derived for clonal heterozygous deletions, since only 1 allele can be observed. Similarly, for clonal homozygous deletions, no allelic information will be available. Is CaSpER suitable for detecting clonal alterations?

Authors' Response: The reviewer is raising an important aspect of CNV calling, which is related to the clonality of CNV calls. In general, we denote the clonality of a deletion by c , where

$$c = \frac{\text{Number of cells with deletion}}{\text{total number of cells}}$$

and c denotes the fraction of cells that harbor the deletion.

To study how clonality impacts the detected CNVs, we simulated gene expression levels of single cell RNA-seq experiments by controlled introduction of deletions into certain set of target regions. To introduce deletions into the datasets, we first selected target regions where we are sure that there no CNV events. Next, we sampled the expression signals from the regions with known deletions and replaced the expression levels of the target regions with these “deleted” expression signals in $P \times c$ cells where P denotes the total number of cells. This way, we simulate deleted expression signals at the target regions on the c cells that harbor deletions. After the expression signals are simulated, we added a sample of $P \times (1 - c)$ normal cells into the population. These cells represent the fraction of cells that do not harbor the deletion. We finally created the “bulk” expression signal by taking the averages among the $P \times (1 - c)$ normal and $P \times c$ simulated expression signals. This procedure generates a simulated bulk expression signal data on P cells.

Next, we simulated the BAF shift signal in the target regions. This is accomplished by using the following formulation:

$$BAF_{shift}(\text{target deletion}) = \frac{P \times c \times 1 + P \times (1 - c) \times 1}{P \times c \times 1 + P \times (1 - c) \times 2} - 0.5$$

where c denotes the clonal fraction of the deletion. The numerator in the above formula computes the total amount of major allele at the deletion site: 1 copy that is originating from the cells with deletion ($P \times c$ cells) and the 1 copy that is originating from the cells that do not have the deletion ($P \times (1 - c)$ cells). The denominator computes the total DNA from all the cells at the deletion site: 1 copy originating from the cells that harbor the deletion and the 2 total copies that are originating from the cells that do not harbor the deletion. The shift computed using the above formula is simply added to the existing BAF shift signal in the target regions.

Thus, we modified both the expression signal (using expression signals of regions with known deletions) and the BAF shift signal at the target regions. In the simulations we also divided the segments with respect to lengths 1mb, 10mb, and 100mb. We show below the plot of the sensitivity of the detected events with respect to the clonality. As expected, the sensitivity increases significantly with the increasing clonality. We did not observe a strict relation between the sensitivity and the deletion length.

Updated Supplementary Figure 15. Plot of the sensitivity of the detected events with respect to the clonality. Performance of CaSpER as a function of clonality. X-axis corresponds to varying levels of clonality; 10%, 30%, 50% and 70%. Y-axis corresponds to TPR value.

We have updated the Results section (**new text line 307-317**) and Methods section (**new text line 786-813**) included the above points.

6. The authors present CaSpER as a tool for visualizing genome-wide RNA-seq signals. However, it is unclear which visualization(s) they are referring to that are unique to CaSpER and what these new visualizations enable that previous visualizations, such as from inferCNV, do not. Similarly, the authors suggest that "these visualizations are especially useful for visually confirming the significance of the results." Please clarify how these visualization enable users to confirm "significance of the results."

Authors' Response: Reviewer is asking clarification about an important aspect of CaSpER. The genomewide visualization and clustering of the CNV events is vital to ensure that the RNA-seq data is explored to the fullest extent possible. This is why we are putting much effort for generating a large number of visual items that combine expression signal, BAF shift signal, and clustering of the samples. At the same time, we also visualize the multiscale smoothed signals whenever possible to make use of the decompositions. As CaSpER performs multiscale smoothing of the expression and BAF shift signals, it generates a very large amount of processed data and visuals become especially important to comprehensively represent the processed data. **These visualizations aim at complementing the detected CNV events so that the users can manually review the visualizations and evaluate any events that**

CaSpER (or other tools) may have missed. We believe that this is vital because there are many factors (clonality, impurity, sequencing depth and coverage, etc.) that affect the accuracy of the CNV detection tools.

Manual review of the visuals can be performed before or after the quantitative analysis of the data. In order to comprehensively visualize the CNV events, CaSpER first generates visuals that combine all the samples so that user can view the events jointly over all samples and co-occurrence and mutually exclusive patterns can be visually inspected. To characterize these patterns, CaSpER performs clustering of samples and clustering of detected CNVs and generates clustering plots. We summarize these visualizations below. To include detailed expression and BAF shift signal for each sample, CaSpER generates the genomewide signal the plots for each sample separately such that each chromosome is plotted on a separate page. This way the user can assess the detailed view of the expression and BAF shift signal for each sample separately.

To our best knowledge InferCNV is the only other popular tool that is focused on generating visualization of the CNV events using single cell RNA-seq data. As we explained before, inferCNV is used mainly for panoramic view of the datasets to qualitatively study the copy number events **using only expression signal and not the BAF shift signal**. InferCNV is used to explore tumor single cell RNA-Seq data to identify evidence for gains or deletions of entire chromosomes or large segments of chromosomes. The only output of inferCNV is the heatmap of expression intensity of genes across positions of the genome in comparison to a set of reference 'normal' cells. An example output of inferCNV is shown below:

Figure 2. An example inferCNV output is shown.

In comparison, CaSpER aims at generating a more comprehensive set of visualizations than inferCNV does. We summarize different visualizations (with the corresponding options) that CaSpER generates below:

plotHeatmap: Visualization of the genomewide gene expression signal plot at different smoothing scales

CaSpER outputs the expression signal at different scales. In these plots, each row is a sample and the columns are the chromosomes. As we exemplified before, these can be useful for comparison of outputs from different scales using the panoramic inspection of the expression signal. Examples are included below. Scale 3 is more about large scale events whereas scale 1 and 2 shows more focal events.

plotLargeScaleEvent: Visualization of the detect large-scale CNV events among all the samples/cells

Large scale event summarization is useful for summarizing the detected large-scale CNV events (deletions and amplifications) over multiple samples. This plot summarizes the large scale CNVs and may reveal the patterns that may otherwise be missed when data is visualized at smaller scales.

plotBAFAISamples: Visualization of BAF shift signal for all samples together

The inspection of BAF shift signal is useful especially when compared to the expression signal to analyze the CNV and LOH events. **inferCNV does not provide any visualizations regarding the genomewide BAF shift signal and we believe that this visualization can enable a more comprehensive analysis of the above-mentioned events.** The BAF shift plots show the BAF shift signal such that each row is the genomewide BAF shift signal profile. An example is included below for reference.

plotBAFOneSample: Visualization of BAF shift signal in different scales for one sample

This option plots the BAF shift signal for one sample at different scales. Similar to the multiscale smoothing of expression signal, this information enables panoramic assessment and identification of CNV and LOH events.

plotSingleCellLargeScaleEventHeatmap: Visualization of large scale event summary for selected samples and chromosomes

This option generates a heatmap view of only the detected large-scale CNV events over all samples. In this plot, the rows correspond to the samples and columns correspond to the detected large scale CNV events. This plot aims at providing the user with a way to visually inspect the large-scale event summaries, i.e. co-occurrence and exclusivity patterns. An example of this heatmap is shown below.

plotMUAndCooccurrence: Visualization of mutually exclusive and co-occurring events

This plot shows the co-occurrence and mutually-exclusive patterns of the detected large-scale CNV events. Each event is represented by a circle where red circles represent amplifications and blue circles represent deletions. Each circle is clearly labeled with the event it represents. The events are connected to each other with lines such that the line thickness represents the significance of the observed pattern. The line type is changed to indicate the type of pattern between different events. An example of this plot is included above in which the large-scale CNV events from the MGH31 single cell RNA-sequencing data are visualized.

plotSCCellCNVTree: Pyhlogenetic tree-based clustering and visualization of the cells based on the CNV events from single cell RNA-seq Data

This option generates a phylogenetic-tree visualization of the CNV events. The tree building uses Fitch–Margoliash method to compare and cluster the cells in terms of the detected CNV events. This plot provides user a way to visually inspect the clonal and architecture of the cells.

plotBAFInSeperatePages: Visualization of BAF deviation for each sample in separate pages

This option creates the BAF shift signal for each sample separately. This way the user can visualize each sample by itself. An example is shown below.

plotGEAndBAFOneSample: Gene expression and BAF signal for one sample in one plot

This option generates the visualization of the gene expression signal and BAF shift signal together so that the user can jointly assess them in one page. An example figure is included below. This plot shows the original data and the smoothed data for all the scales to include a complete visualization of the multiscale decomposition.

We updated our vignettes and included above visualization examples.

7. The CaSpER R package currently lacks a manual and is generally poorly documented. The authors should document each function using Roxygen notation and export a manual as well as provide outputted vignettes with sample usage instructions to facilitate users.

Authors' Response: We thank the reviewer for pointing out an important issue about reusability of CaSpER. Documentation is one of the most vital components of the developed bioinformatics tools. We have now updated the source code and included Roxygen based documentation of CaSpER with an exported manual. Also, we have provided vignettes with sample usages so that users can easily run their analysis using CaSpER.

Minor comments

1. As 3'-only RNA sequencing techniques such as Drop-seq, In-drop, 10X Genomics, and others are rapidly becoming more popular, please comment on whether CaSpER is suitable for these 3' only RNA sequencing data. How will its accuracy in identifying CNVs of different sizes or clonality be affected?

Authors' Response: This is a very important question about the suitability of CaSpER (and other single cell-based CNV detection tools) for usage with 3' capture-based scRNA-seq technologies. CaSpER's applicability relies mainly on the quality of the gene expression signal and the BAF shift signal. As the reviewer points out, the data from 3'-only technologies are biased such that the reads are enriched along the 3' end of the genes compared to the 5' ends of the genes. While 3' based reads can be used to generate expression levels that are of fair quality⁶, the BAF shift signal may be adversely impacted because BAF shift will be measured for only the variants that are located close to the 3' ends of the genes. We therefore expect that BAF shift signal will be much sparser when data from these technologies are used.

While 3' technologies are expected to impact the BAF shift signal adversely, it is also important to point out that the 3' bias of these methods do not necessarily mean that the reads come only from 3' ends of the genes. Recent publications point out that there is an enrichment of reads on 3' ends of transcripts and there are still decent number of reads from the 5' ends of genes. Secondly, CaSpER pools all the reads from all the cells while generating the BAF shift signal. Therefore, we believe that the pooling will soothe the adverse effects of the 3' bias as long as decent number of reads are accumulated along the transcripts. Having noted these, we believe that CaSpER can benefit a new model for BAF shift generation step where it takes into account about the 3' bias of the sample preparation assays.

We added a discussion of the limitations of CaSpER and the above points in the revised manuscript (**new text line 438-455**).

2. As CaSpER relies on a sliding window-based median filtering, how well can CaSpER call the precise boundaries of CNVs? Does this depend on the window size used?

Authors' Response: The reviewer raises an important point about the smoothing window sizes in computation of multiscale decomposition. To address this point, we used different starting window sizes for decompositions. We next computed the segment-based accuracy (TPR) of the CNV calls made by CaSpER using the GBM single cell RNA-seq data in which we simulated introduction of deletions at 1MB and 10MB. The details of simulated data generation are explained in the revised version of the Methods "Window length I parameter selection" Section. The results are included in the figure below. It can be seen that for small smoothing window sizes, segments show relatively low concordance with the ground truth. As the window size increases, the concordance increases and saturates around 80% at the around window length of 50 for both event lengths. Another important point to keep in mind is that as the starting window size increases, it takes longer to process the data. Putting this together, above results indicate that the starting window length of 50 enables a fair tradeoff between computational time and accuracy.

We have updated the Methods section (**new text line 604-617**) included the above points.

Updated Supplementary Figure 23. TPR values of the CNV calls made by CaSpER using the GBM single cell RNA-seq data in which we simulated introduction of deletions at 1MB and 10MB. For small smoothing window sizes, segments show relatively low concordance with the ground truth. As the window size increases, the concordance increases and saturates around 80% at the around window length of 50 for both event lengths.

3. In the HMM model, the authors use a symmetric transition matrix where there is equal likelihood of transitioning from every state to every other state. For example, given an HMM model initialized in the copy neutral state (state 3), it is equally likely (with probability $t=1e-6$) to transition to a homozygous deletion, heterozygous deletion, one-copy gain or multiple-copy gain. Similarly, given an HMM model at a multiple-copy gain state, it is equally likely to transition to a homozygous deletion, heterozygous deletion, neutral or one-copy gain state. How did the authors arrive at this symmetric transition matrix?

Authors' Response: The reviewer is pointing out a critical detail which we did not clarify well in the current manuscript. **It must be pointed out that the matrix that the reviewer is referring to is the initial transition matrix that is supplied to the HMM.** The initial transition matrix is setup to uniform distributions because we wish to ensure that there are no biases towards any of the (non-self) state transitions. We believe that this is reasonable because cancer cells (unlike healthy genomes) can contain complex CNV patterns along their genomes and the

transition probabilities may be fairly diverse when different cells and cancer types are compared. Thus, we initialize transition probability matrix to be uniform among all the transitions. While HMM segmentation is performed, the model performs an expectation-maximization procedure (i.e., Baum-Welch Algorithm) and iteratively updates the transition probabilities. Consequently, the final segmentations are computed using the updated transition probabilities that are fit to the data. We have included examples of final transition matrices. It can be seen these matrices are not symmetric (unlike initial matrices). It is also worth noting that the final transition matrix is data dependent as the EM-step fits the transition matrix to the data (Table 3).

We clarify the above points in the Methods Section of the manuscript.

	1	2	3	4	5
1	0.999999	7.16E-07	1.15E-07	2.50E-08	2.50E-08
2	9.05E-07	0.999997	1.69E-06	5.09E-08	1.18E-07
3	1.05E-07	2.03E-06	0.999996	1.53E-06	2.31E-07
4	2.54E-08	2.50E-08	1.81E-06	0.999998	4.22E-07
5	2.50E-08	2.50E-08	2.50E-08	7.24E-07	0.999999

Table 3. An example final transition matrix. Rows and columns represent the originating and target states, respectively. Each cell contains the transition probability from the originating state to the target state. Yellow highlighted probabilities show asymmetric transition matrix values.

We have updated the Methods section (**new text line 770-775**) included the above points.

4. As CaSpER's HMM model distinguishes between one-copy gain and multiple-copy gain, how well does CaSpER distinguish between the two states?

Similarly, would CaSpER be able to identify multi-copy duplication events where both alleles have been amplified equally? In this case, the B-allele frequency would remain comparable to a neutral region's.

Authors' Response: We thank the reviewer for bringing up an important point. The difference between the one-copy gain (State 4) and multi-copy gain (State 5) in the HMM model is reflected in their emission probabilities of the expression signal. While these are distinct states, CaSpER does not distinguish these events in the final CNV calls for segments: One-copy and multi-copy gains are both assigned an “amplification” call. Therefore, the distinction between these two events is only intermediary in CaSpER such that HMM models one-copy gains and multiple-copy gains. Finally, it is fairly hard to quantify how well CaSpER distinguishes between these two amplification events since we do not have a reliable gold standard that distinguishes one-copy gains and multiple-copy gains.

The balanced multi-copy gains, as the reviewer points out, generally correspond to the high amplification states (with 4, 6, 8, ... copies). **When a region is assigned to a high amplification state (State 5 in HMM state nomenclature), the BAF shift is not used for correction and CaSpER automatically assigns them an amplification call.** This behavior is by design because, as the reviewer points out, the high-level amplifications may not have a high BAF shift associated with them. So if we enforced the BAF shift consistency with high amplifications, it would be highly likely that we would miss these events. The BAF shift signal is used to evaluate the low deletion state (State 2) and amplification state (State 4), where the

expression signal shows some decrease that must be corroborated with the BAF shift in order to make a call. The users can distinguish the regions with balanced amplification by first selecting the high copy amplification regions whose BAF shift is low.

We are hoping that our explanation clarifies the reviewer's insightful question. We have incorporated the above discussion into the Results Section (**new text line 131-146**) of the manuscript.

5. Similarly, as the authors mention LOH in addition to CNVs, can CaSpER identify copy-neutral LOH events?

Authors' Response: We thank the reviewer, we have now updated CaSpER to report the LOH regions where assigned HMM assigns a neutral state (i.e., state 3) and the BAF shift signal is significantly high, i.e., the BAF shift signal is higher than the threshold t (See Methods for details of selection of t).

6. The authors note that "identifying CNVs before calling SNVs can give very useful information for correct identification of SNVs." On the flip side, point mutations will also affect the B-allele frequency, potentially impacting CNV detection. Other sources of noise unique to RNA such as RNA-editing, amplification errors, and sequencing errors may also contribute to noise in the B-allele frequency. Please comment on the impact of point-mutations, RNA-editing, amplification errors, and sequencing errors on CNV identification with CaSpER.

Authors' Response: The reviewer is highlighting a very insightful aspect of the technical challenges around our idea about calling CNVs before identification of SNVs. The factors that the reviewer is bringing up will create biases. While amplification errors and sequencing errors impact other methods such as DNA-seq based detection, RNA-editing impacts specifically RNA-seq based detection of CNVs and it introduces potentially adverse biases.

Point mutations impacting B-allele frequency. As the reviewer points out, the statement becomes circular when we use the B-allele frequency of SNVs to identify CNVs and claim that the detecting CNVs before detecting SNVs is useful for correct identification of SNVs. We must therefore clarify this. While we agree that the individual point mutations affect the B-allele frequency, we are not using the B-allele frequency of individual variants; **we are using a collective shift in the observed B-allele frequencies of consecutive variants.** This is the main reason why we use the multiscale smoothing B-allele shift signals while assigning the CNV calls. The smoothing summarizes the estimated B-allele shift signal among potential consecutive SNVs. We are therefore claiming that one can detect the B-allele shift without the existence of a high-quality set of SNVs. We believe that the concordance between our B-allele shift plots (Generated without a high quality SNV call set) and the expression signals is convincing evidence that support this claim.

RNA-editing is a general mechanism where RNA is edited post-transcriptionally by cells. The nucleotide sequence of the transcripts are changed as the result. As the RNA sequence is changed, the sequenced RNA reads will reflect the nucleotide modifications. While these edited nucleotides may not affect the gene expression quantifications (Assuming they do not interfere with read mapping), they may potentially impact the BAF shift information because the modified nucleotides may look like SNVs while reads are being processed. The RNA editing events are mainly dominated by A-to-I modifications. While the exact number of RNA editing sites are not

known, the common RNA editing events is hypothesized to be rare, on the order of 10,000-100,000 sites^{7,8} and many of the detected editing events overlap with repeat regions and do not overlap with coding sequences.

Since the number of the events is low, we believe that these events will not have major impact on the BAF shift generation. Secondly, the RNA-seq data is predominantly enriched in the coding and exonic sequences. Since there is high association of RNA-editing events with repetitive non-coding elements, we believe that they will have almost unnoticeable impact on the BAF shift signal measured on the exonic regions. However, the cancer cells may still have high RNA editing just because the editing pathways may be malfunctioning. Therefore, we believe the impact of RNA editing should be considered carefully any time RNA-Seq data is used to identify CNVs.

Amplification errors stem from the PCR amplifications that are used to increase the cDNA content in the sequencing libraries. PCR amplification may introduce errors that get amplified within any cycle. Most alarmingly, these errors are correlated among different reads and it is hard to correct for these errors. These amplified errors may seem like novel mutations. Especially in the lack of a matching control data, these PCR errors may adversely impact BAF shift signal generation. We hope that the other less biased PCR-free amplification assays will be more available to decrease the amplification errors. In addition, molecular barcoding such as UMIs may be useful to detect duplicates and remove them effectively.

Sequencing errors are introduced in the sequencing step when the sequencing machine wrongly assigns a base. These errors are less of a concern than PCR amplifications since they are generally independently introduced in different reads but this may not always be the case because sequencing errors can be context specific. Nevertheless, the sequencing error rates of short read sequencing technologies such as Illumina is very low; around 0.1% per base⁹. We therefore think the sequencing errors will impact BAF shift generation slightly.

We incorporated above points in the Discussion Section (new text line 451-455) and Supplementary Information.

7. To generate the appropriate allele-based frequency signals from RNA-seq bam files, do reads need to be filtered for duplicates? Or is the minimum number of total reads supporting a SNP including potential duplicates?

Authors' Response: As before, the reviewer points out an important detail related to BAF shift generation. BAFExtract method does not use deduplicated reads. This was necessary to ensure that we had enough reads to generate the BAF shift generation. We employed a filter on the number of supporting reads for any candidate SNV (default: 4 supporting reads for candidate SNV). We also initially filter the reads based on mapping quality (default: 50). These parameters can be tuned by the user in BAFExtract program. We revised the manuscript to clarify the details of BAF generation.

We incorporated above points in the Methods Section (**new text line 502-504**).

8. The authors note that "for 5q:14q event pair in patient MGH31, we discovered GFPT2 gene to be highly expressed in 5q amplified clone." And "for 5q:19q event pair in patient MGH28, we discovered NOS2 gene to be highly expressed in 19q deletion clone." Are these genes located within the amplified regions? What proportion of affected genes

constitute cis effects (within affected region) vs. trans effects (outside affected region)?

Authors' Response: We thank the reviewer for raising an important point. *GFPT2* gene is located in chr5:179,727,690-179,780,387 it is within the affected region. *NOS2* gene is located in chr17:26,083,792-26,127,555, it is outside the effected region.

For 5q:14q event pair in patient MGH31, we have discovered 7 genes to be differentially expressed and 2 out of 7 is within affected region. For 5q:19q event pair in patient MGH28 we have discovered 34 genes to be differentially expressed and 3 out of 34 is within affected region.

Differentially expressed genes and their locations within the chromosome are listed in SupplementaryData1.

9. The motivation for calling CNVs from bulk RNA-seq is unclear. It is very common to do both bulk DNA and bulk RNA sequencing on the same sample, particularly in cancer. So it is unclear why inferring CNVs from bulk RNA-seq would be desirable.

Authors' Response: We thank the reviewer for raising an important point related to the motivation of CaSpER about application on the bulk RNA-seq data. As the reviewer points out, RNA-seq and WGS is performed regularly on same samples. We foresee that new experimental designs can first start by performing RNA-seq rather than starting with WGS. This will enable the researchers to study the samples simultaneously from the genetic and transcriptomic perspective. After this evaluation, the researchers can focus on the uncharacterized samples, for example samples that do not harbor any of the known large scale CNV event associated with known subtypes of a cancer. In principle, the RNA-seq can be used to call somatic SNVs and short indels and can potentially identify these known markers. After selecting the uncharacterized samples, the researchers can continue with performing WGS of these uncharacterized samples.

This "sequential" or "RNA-seq first" design is advantageous for several reasons. First, cost can be decreased significantly. WGS is currently much more expensive in terms of sequencing, analysis, and storage compared to the costs associated with RNA-seq. If the researchers can decrease the samples for which WGS is performed by focusing on the most interesting samples, this can decrease the costs substantially. Additionally, RNA-seq can be used to identify much more information such as the transcript fusion events that may have important clinical implications. Therefore, we believe that RNA-seq data provides a very high utility for every dollar spent. While we are optimistic with the RNA-seq first design, we are just presenting this case as a potential example and we do not want to overstate this scenario because the sample preselection may create adverse statistical biases that must be properly controlled.

On another note, with the advent of RNA-sequencing from the formalin-fixed paraffin-embedded (FFPE) tissue samples, we can now perform low-cost RNA-sequencing of these samples. This represents a low-cost alternative for simultaneous transcriptomic and genomic characterization of the FFPE samples compared to the laborious and expensive DNA-sequencing.

We incorporated above points in the Discussion Section (**new text line 367-383**).

10. The authors note that "it remains technically challenging to assay both the genome and transcriptome from the same cell." However, a number of approaches for simultaneously assaying DNA and RNA

information from the same single cell has been published, including G&T-Seq by Macaulay et al (Nature Methods 2015), and BioMark by Wang et al (Genome Research 2017) to name a few. While it may still be true that these approaches are too technically challenging for broad-scale implementation, they should be acknowledged.

Authors' Response: Reviewer has raised important references that relate directly to our study. We thank the reviewer for taking time to compile the list of references. We have now updated Introduction Section (new text line 51-53).

11. The authors note that "many algorithms have been developed for detecting CNV events from DNA sequencing using depth of coverage." Many algorithms have also been developed for detecting CNV events from DNA sequencing using B allele frequency, including PennCNV by Lima et al (BMC Bioinformatics 2017). Please acknowledge these previous works.

Authors' Response: We appreciate the reviewer's input. We acknowledged previous works (new text line 79, 92).

12. The authors claim that "loss-of-heterozygosity...has previously shown to be extremely useful for identifying CNVs." Previous provide a citation.

Authors' Response: We again appreciate greatly careful reading of the manuscript. We have now updated the manuscript to include citations to support this claim. This also directly relates to the reviewer's previous comment (new text line 92).

13. Similarly, the authors note that "unlike most other tools, CaSpER does not require heterozygous variant calls to generate the allelic shift profile." Please provide a citation for these "other tools".

Authors' Response: We thank the reviewer for pointing out the vague statement that must be clarified. We updated this sentence and acknowledged these previous previous works (new text line 94).

1. Clark, V. E. *et al.* Genomic analysis of non-NF2 meningiomas reveals mutations in TRAF7, KLF4, AKT1, and SMO. *Science* (80-.). **339**, 1077–1080 (2013).
2. Aran, D., Sirota, M. & Butte, A. J. Systematic pan-cancer analysis of tumour purity. *Nat. Commun.* **6**, (2015).
3. Piskol, R., Ramaswami, G. & Li, J. B. Reliable identification of genomic variants from RNA-seq data. *Am. J. Hum. Genet.* **93**, 641–651 (2013).
4. Carter, S. L. *et al.* Absolute quantification of somatic DNA alterations in human cancer. *Nat. Biotechnol.* **30**, 413–421 (2012).
5. Patel, A. P. *et al.* Single-cell RNA-seq highlights intratumoral heterogeneity in primary glioblastoma. *Science* (80-.). **344**, 1396–1401 (2014).
6. Ziegenhain, C. *et al.* Comparative Analysis of Single-Cell RNA Sequencing Methods. *Mol. Cell* **65**, 631–643.e4 (2017).

7. Porath, H. T., Knisbacher, B. A., Eisenberg, E. & Levanon, E. Y. Massive A-to-I RNA editing is common across the Metazoa and correlates with dsRNA abundance. *Genome Biol.* **18**, 185 (2017).
8. Bahn, J. H. *et al.* Accurate identification of A-to-I RNA editing in human by transcriptome sequencing. *Genome Res.* **22**, 142–150 (2012).
9. Glenn, T. C. Field guide to next-generation DNA sequencers. *Mol. Ecol. Resour.* **11**, 759–769 (2011).

Reviewers' comments:

Reviewer #1 (Remarks to the Author):

The authors have taken great efforts to address my concerns, which are now allayed.

Rameen Beroukhim

Reviewer #2 (Remarks to the Author (last round of review))

This paper presents CaSpER, a method to identify and visualize copy number variants (CNVs) from bulk RNA-seq and scRNA-seq data. I have the following 3 main concerns. Overall I think the performance of CaSpER is not convincing and the algorithmic innovation is insufficient.

1, The accuracy of CaSpER in identifying CNVs has not been compared with any existing methods. As the authors stated, there exists methods that use RNA-seq to infer CNVs, especially HoneyBADGER, which, as stated in its paper, can identify CNV in individual cells from single-cell RNA-sequencing data. Therefore, the authors should either compare with HoneyBADGER (at least on scRNA-seq data) or explain why such comparison is not possible.

2, I cannot see much methodological innovation within CaSpER. The authors stated in the paper that CaSpER employs a *novel* methodology for generation of genome-wide BAF signal profile from the reads and utilizes it in multiscale fashion for correction of CNVs calls. However, as we can see from HoneyBADGER paper, it already integrates allele and normalized expression information to infer CNVs. CaSpER uses BAF information to correct CNV events (Step 7, Figure 1). For me this is interesting: a new method that can combine these two types of information in an efficient way would be very useful. Unfortunately, in the Methods section, I could not find details about how such correction is performed.

3, In terms of biological results, CaSpER has reported mutually exclusive and co-occurring CNV events in the scRNA-seq GBM dataset (Figure 5). But why these CNV events are significant has not been convincingly discussed.

Reviewer #3 (Remarks to the Author):

In this revision, the authors have made a number of key clarifications that have substantially improved the manuscript. In particular, the authors have provided new benchmarks of CaSpER's ability to accurately identify CNVs in bulk RNA-seq data at different length-scales, clonalities, and parameter ranges. The authors also provide new analyses that compare CaSpER with previously published approaches, HoneyBADGER and inferCNV, again using bulk RNA-seq data. The authors have also extensively improved the Methods section with new details of how CaSpER works.

However, a number of critical outstanding issues remain unclear or were insufficiently addressed in the revision that preclude publication. While CaSpER's capacity for identifying CNVs in bulk RNA-seq data have been sufficiently demonstrated, it remains unclear how CaSpER performs, especially as compared to HoneyBADGER and inferCNV, with respect to single-cell RNA-seq data. The authors fail to provide thorough performance characterization of CaSpER using single-cell RNA-seq data and it remains unclear how suitable CaSpER is for detecting CNVs at different length-scales or clonalities in single-cell RNA-seq data. Likewise, the primary visualization capabilities of CaSpER appear identical to those of inferCNV and HoneyBADGER so CaSpER's utility as a visualization tool remains to be convincingly shown.

Major comments:

1. While the application of CaSpER to the bulk RNA-seq of breast cancer samples from TCGA is promising, its performance for single-cell RNA-seq data, particularly as compared to previously published approaches, still requires further clarification and fails to demonstrate improvement over previously published approaches. For example, what is the accuracy of identifying focal segments at different length scales in single-cell RNA-seq data? It cannot be assumed that CaSpER's performance using bulk RNA-seq data will translate to comparable levels of accuracy when applied to single-cell RNA-seq, particularly since different thresholds are used.

2. In the updated figure 3C as referenced in the response to reviews (5E in the manuscript - "Clustering of large scale events generated by CaSpER in patient MGH31"), CaSpER appears to identify CNVs (amplification on 6q, deletions on 5q, 6p, 10p, 10q) in the putative normal cells in patient MGH31. This seems to be a much higher false positive rate than what would be expected based on bulk-RNA-seq benchmarks (which suggest an FPR of 0.3%). Likewise, there appear to be a number of false negatives, where clonal alterations not being identified by CaSpER in the cancer cells. This suggests that CaSpER's performance in terms of TPR and FPR rates are likely different for single-cell RNA-seq data and should be characterized in order to substantiate the conclusion that CaSpER is suitable for accurate identification of CNVs in single-cell RNA-seq data.

3. To assess the CNV-calling capabilities of CaSpER compared to previously published approaches, the authors provide a new performance comparison of CaSpER to HoneyBADGER (Fan et al, Genome Research 2018), using bulk RNA-seq data. However, as the authors note, HoneyBADGER was developed for CNV-calling for single-cell RNA-seq data and thus conclude that HoneyBADGER is not suitable for detection of CNVs for bulk RNA-seq data as expected. So it is unclear why the performance comparison was done using bulk RNA-seq data when single-cell RNA-seq data was also readily available, particularly since both papers have analyzed the same GBM data from Patel et al (Science 2014).

4. The suitability of CaSpER for usage with 3'-only RNA-sequencing techniques remain unclear. While the authors provide new speculation as to how CaSpER may perform worse with 3'-only sequencing, they provide no new analysis demonstrating such decreased performance even though 3'-only sequencing data of cancer samples are available. For example, the authors speculate that BAF will be sparser from 3'-only sequencing, and fail to support such speculation with data, even though such data are available.

5. Although the authors suggest that visualization capabilities are a distinguishing feature of CaSpER compared to previously published approaches, the main heat map visualizations are essentially identical to visualizations provided by HoneyBADGER and inferCNV. For example, the new updated Supplementary Figure 16 and 17 actually highlight how similar visualizations in CaSpER and HoneyBADGER are (and to a lesser extent inferCNV) with the only appreciable difference being that CaSpER scales chromosomes by size and uses a different color scheme.

Similarly, CaSpER's plotHeatmap function visualizes genome wide gene expression signals at different smoothing scales. However, parameters in both inferCNV and HoneyBADGER can be passed in order to visualize different smoothing scales. Inclusion of these parameters doesn't justify presenting CaSpER as a visualization tool.

Likewise, HoneyBADGER provides allele-level visualizations that offers comparable information and insights to CaSpER's BAF shift plots.

The only visualization unique to CaSpER appear to be the graph of mutually exclusive and co-occurring events. However, it remains unclear how this visualization is useful and will allow users to gain additional insights that are not apparent from the heat map visualizations common to all 3 approaches. On a more fundamental level, it is unclear why users would want to visualize the

relationship among CNV events. More importantly, it is unclear how these relationships among CNV events fit onto the proposed phylogenetic relationship among cells proposed by CaSpER's plotSCellCNVTree function.

6. The authors suggest that although they do not have access to experimental validation for confirming the newly discovered CNVs in CaSpER's analysis of glioblastoma cells, supportive evidence can be seen using the visualizations from inferCNV. However, inferCNV's visualization is also based on smoothed gene expression, and thus does not offer orthogonal information for independent validation. A more orthogonal piece of information that could be used to validate these novel CNVs would be allele data. What are the allele information supporting these novel CNVs?

Furthermore, the authors note that CaSpER achieves a lower PPV and TPR rate for identifying focal alterations compared to large-scale CNVs (based on benchmarks from bulk RNA-seq). How can we be sure that these novel focal CNVs identified in the single-cell RNA-seq data are not false positives, particularly since they affect only a few cells?

The authors emphasize that visualize inspection can confirm these novel CNVs. However, looking at the updated Supplementary Figure 14, it is unclear to me why 14q was identified as a deletion when equally blue cells can be seen on chromosome 9, 16, and 22, which are not identified as a deletion. Similarly, it appears that the cells in MGH31 have been reordered to cluster the CNVs on 5q:14q versus 1p:13q. Ordering cells by CNVs patterns on 5q:14q appear to separate cells for the 1q:13q visualization. The reliance on visual inspection is very subjective and can be biased based on the ordering of cells.

Minor comments:

1. In the software's website, tutorials are provided as .R files that a user must execute. Please provide these as .Rmd files like how the main documentation is done or as compiled PDFs.
2. In MGH31, there appears to be a (few?) cells that harbor both 5q amplification and 14q deletion. Are these the same cells that also harbor the 13q deletion and 1p amplification? Or are they different? In order for this CNV pattern to occur, a 5q amplification or a 14q deletion will have had to occur in two independent sub clones, which seems unlikely. Or is this a false positive?
3. In figure 5E, a number of cells appear to harbor 7q and 6q amplifications, but both of these CNVs are not included in 5F. Yet a fewer number of cells, the putative sub clone, appear to harbor 1p and 5q amplifications, which are included in 5F. Again, it is unclear how these relationships among CNV events fit onto the proposed phylogenetic relationship among cells.

"Reviewer #2 (Remarks to the Author (last round of review))

This paper presents CaSpER, a method to identify and visualize copy number variants (CNVs) from bulk RNA-seq and scRNA-seq data. I have the following 3 main concerns. Overall I think the performance of CaSpER is not convincing and the algorithmic innovation is insufficient."

Author's Response: We sincerely thank the reviewer for careful consideration of our manuscript. The reviewer's comments have made us to re-assess the presentation of the data and the content of the analyses and have enabled us to improve the manuscript. We understand that the reviewer **finds performance of CaSpER not convincing** and that **there is not enough algorithmic innovation in CaSpER**. We provide below our responses to these comments. We also would like to point out that some of the comments have been addressed in the previous revision and we refer to the previous revision for these comments. We try and explicitly state below which comments have been addressed in current round and previous round.

"1, The accuracy of CaSpER in identifying CNVs has not been compared with any existing methods. As the authors stated, there exists methods that use RNA-seq to infer CNVs, especially HoneyBADGER, which, as stated in its paper, can identify CNV in individual cells from single-cell RNA-sequencing data. Therefore, the authors should either compare with HoneyBADGER (at least on scRNA-seq data) or explain why such comparison is not possible."

Author's Response: We thank the reviewer for bringing up important points regarding performance benchmarks. We are dividing our response to the reviewer's comment into 2 parts because portions of the comments were addressed in the previous revision (Bulk RNA-seq benchmarks) and we have also added new benchmarks (Single Cell RNA-Seq benchmarks).

1. Performance Benchmarks on Bulk RNA-Seq Data from previous Revision: As the reviewer suggests, the performance comparisons are very important to highlight where CaSpER stands within the current set of computational tools. Since the last review, we have performed several performance comparisons, including bulk and single cell RNA-sequencing datasets, with HoneyBADGER. In general, we observe that HoneyBADGER exhibits substantially lower sensitivity compared to CaSpER in terms of both large-scale level and gene-level summarizations of the CNV events. Please note that the summarizations are described in the Methods Section of the manuscript. Also, the benchmark metrics, i.e., True positive rate (TPR) and false positive rate (FPR) are described in detail in the Methods Section. For bulk RNA-Seq comparisons, we used the TCGA-GBM datasets. The ground truth CNV calls for this is generated from the genotyping arrays from the TCGA Project. We include below the performance benchmarks, as included in the last set of revisions:

		Large Scale Level		Gene Level	
		HoneyBADGER	CaSpER	HoneyBADGER	CaSpER
TPR	Deletion	6%	85%	5.9%	77.6%
	Amplification	25%	71%	22%	57.5%
FPR	Deletion	0.07%	3%	0.27%	3.5%
	Amplification	0.14%	1.5%	0.23%	2%

Updated Supplementary Table 1. TPR and FPR values for bulk RNA-Seq TCGA-GBM

		Large Scale Level		Gene Level	
		HoneyBADGER	CaSpER	HoneyBADGER	CaSpER
TPR	Deletion	6.4%	79%	3.2%	62.8%
	Amplification	8%	60.3%	10.5%	56.6%
FPR	Deletion	0.46%	7.7%	0.75%	6%
	Amplification	0.1%	3.8%	0.09%	7%

Updated Supplementary Table 2. TPR and FPR values for bulk RNA-Seq TCGA-BRCA

When we applied HoneyBADGER on bulk TCGA-GBM RNA-Seq data, expression-based model of HoneyBADGER discovered only two events; deletion in **chr22:19941607-50783663** and amplification in **chr7:1815793-151080866**. Allele based model of HoneyBADGER discovered six focal deletion events; **chr1:109745618:109795026**, **chr6: 3255327: 3285286**, **chr9:134384277:134395628**, **chr14:102467847:103059220**, **chr20: 60904853: 60909060**, **chr22: 26862153: 26895337**.

When we applied HoneyBADGER on bulk TCGA-BRCA RNA-Seq data, expression-based model of HoneyBADGER discovered amplification events in **chr1:148808181:248849517**, **chr17: 19770829:82945922**, **chr19:531712: 58558960**, **chr20:296968:64287821** and deletion events in **chr16:31489471:89968060**, **chr11:87037844:117204782**, **chr6:107867756:170584692**, **chr18:158383:214629**. Allele based model of HoneyBADGER discovered the following focal deletion events;

chr	start	end
chr3	88189341	88199179
chr7	149427409	149522366
chr9	130670652	130710726
chr10	135076523	135180430
chr14	24910210	25101589
chr16	23521643	23563501
chr22	50277835	50317049

From the above results, it can be seen that HoneyBADGER is not suitable for detection of CNVs from the bulk RNA-seq datasets.

The manuscript was updated in the last to highlight the above points (**new text line 342-355**).

2. Single Cell Data Performance Benchmark: In the current revision, we have now added benchmarking of the single cell RNA-seq data. In this benchmark, we have been challenged for finding ground truth data for copy number variants in single cells matching to the single cell RNA-seq experiments. This is because the methods for simultaneous measurement of single cell RNA-sequencing and single cell DNA sequencing (or copy number profiling) are still very much under development and they do not provide high number of cells where CNVs and RNA-seq is probed simultaneously^{1,2}. Moreover, we were constrained by the fact that we needed samples with decent number of copy number events such as tumor samples, which decreased the potential datasets that we could use for single cell benchmarks.

Among the methods, G&T-Seq and DR-Seq technologies represent the most well-developed simultaneous RNA-DNA measurement methods. Among these, we saw that G&T-Seq does not have readily publicly available the data for immediate usage. We found that the data from the original DR-Seq² study, where the simultaneous RNA and copy number measurements are publicly available for 7 cells from Breast cancer cells with many amplification and deletion events.

CaSpER shows good concordance with the Absolute CNVs in DR-Seq Data: In DR-Seq study, the authors used a computational approach and assigned absolute copy numbers to segments. We first evaluated how the copy number states from CaSpER correlate with the copy number assigned to each gene. We show below, for each cell, the distribution of the DR-Seq assigned copy numbers corresponding to the HMM copy number states. We observed a fairly high consistency between the HMM copy number state and the DR-Seq assigned copy number **within each cell**. Higher copy number states in CaSpER correspond to the amplifications and lower copy number states correspond well to the deletions.

Updated Supplementary Figure 16. Figure shows, for each cell (X3, X10, X13, X20, X25, X26, X28; $n=7$) probed by the DR-Seq study, the distribution of the absolute copy number assigned to the genes (y-axis) vs the HMM state (x-axis). Box plots indicate the distribution of absolute copy number assigned to each gene by DR-Seq study, for the corresponding copy number state.

Comparison of CaSpER and HoneyBADGER using scRNA-Seq data. We next focused on comparison of the methods. For this, we first processed the absolute copy numbers from DR-Seq study to build the relative copy numbers (i.e., amplification, deletion, neutral) in each cell.

This is necessary because both CaSpER and HoneyBADGER assigns relative copy numbers to the detected segments where copy numbers are assigned with respect to the average DNA content in each cell as we discuss further below.

To generate the ground truth relative CNV calls for each cell, we first computed the ploidy in each cell using following:

$$ploidy_i = \sum_j \frac{((Absolute\ CN\ of\ Segment\ j\ in\ cell\ i) \times (Length\ of\ Segment\ j\ in\ cell\ i))}{\sum_k (Length\ of\ Segment\ k\ in\ cell\ i)}$$

Where $ploidy_i$ indicates the average DNA content in cell i . We next assigned amplification and deletion to each cell using $ploidy_i$:

$$CNV_j = \begin{cases} Amplification, & \text{if } (Absolute\ CN\ of\ Segment\ j) > ploidy_i \\ Deletion, & \text{if } (Absolute\ CN\ of\ Segment\ j) < ploidy_i \\ Neutral, & \text{if } (Absolute\ CN\ of\ Segment\ j) = ploidy_i \end{cases}$$

where CNV_j denotes the relative copy number value assigned to segment j . We need to perform the relative copy number assignments because both CaSpER and HoneyBADGER assign copy numbers relative to the ploidy of the cell. In other words, these tools do not assign absolute copy numbers. Thus, we needed to convert the absolute copy numbers to relative copy numbers. We next used the relative copy numbers to assign the large scale false positive rate and true positive rate to the CNV calls generated by CaSpER and HoneyBADGER using the single cell RNA-seq data for 7 cells.

We used the expression quantifications from DR-Seq data as input to CaSpER and HoneyBADGER. We evaluated the TPR and FPR for the detected large-scale deletion and amplification events. CaSpER achieves 32% TPR and 1.6% FPR in deletion events and 49% TPR and 3.8% FPR in amplification events. If we relax the parameters (with $\gamma=1$), CaSpER achieves 45% TPR and 3% FPR in deletion events and 62.6% TPR and 8.6% FPR in amplification events.

HoneyBADGER could not detect any CNV events even though the ground truth data has many amplification and deletion events. We also ran HoneyBADGER with numerous combinations of parameter configurations (clustering depth, inclusion of all genes in analysis, different approaches for data normalization) but these did not change the result from HoneyBADGER (**new text line 319-342**).

In summary, the benchmarks indicate that HoneyBADGER represents a fairly specific method for detection of copy number variants from bulk and single cell RNA-sequencing datasets. Even with numerous different parameter configurations, we observed that HoneyBADGER does not perform sensitively. On the other hand, CaSpER exhibits favorable performance for bulk RNA-sequencing data and more conservative performance on the single cell RNA-sequencing datasets that we benchmarked it on.

*"2, I cannot see much methodological innovation within CaSpER. The authors stated in the paper that CaSpER employs a *novel* methodology for generation of genome-wide BAF signal profile from the reads and utilizes it in multiscale fashion for correction of CNVs calls. However, as we can see from HoneyBADGER paper, it already integrates allele and*

normalized expression information to infer CNVs. CaSpER uses BAF information to correct CNV events (Step 7, Figure 1). For me this is interesting: a new method that can combine these two types of information in an efficient way would be very useful. Unfortunately, in the Methods section, I could not find details about how such correction is performed."

Author's Response: We appreciate the reviewer's critical input that helps to clarify these important points. We address these questions below:

Innovation in BAF generation by non-dependency on reference SNP panels: We feel that we did not highlight and clarify this point well. There is a fundamental difference in the way CaSpER computes BAF information compared to how HoneyBADGER computes BAF information. HoneyBADGER requires the SNV calls from The ExAC Database by computing the BAF at each variant identified by ExAC. We believe that this creates a major obstacle for processing data because ExAC data must be downloaded and then processed by HoneyBADGER. CaSpER, however, does not require the variants to be specified a-priori. In fact, CaSpER starts simply from BAM files and simultaneously identifies the candidate SNVs and measures the BAF shift. This way, CaSpER does not rely on a predetermined set of variants. There are two innovations related to this that we would like to highlight:

1. Our BAF shift generation works well with gene expression for detection of CNVs. This implies that any BAF shift based method could utilize our approach to efficiently extract BAF shift signal without depending on a predetermined set of SNVs. This includes the methods for CNV detection from whole exome and whole genome sequencing datasets. Thus, our approach for generating BAF shift without a reference SNP panel has, in our opinion, far reaching implications that we believe can be used to further extend other CNV detection methods.
2. Generation of BAF shift signal without reference panel set makes CaSpER much more portable in certain cases. ***For example, if the genome assembly version of ExAC does not match the genome version of the BAM file, it will not be possible to run HoneyBADGER's BAF extraction because ExAC coordinates will not be matching. On the other hand, CaSpER can be easily run as it does not depend on a preselected set of regions.*** It should be noted that this is not a far-fetched scenario. The current version of ExAC database (version 1.0) is built on hg19 genome assembly. When the read mapping is performed on newer hg38 assembly, all the variants must be lifted over, which is a fairly tedious and error-prone process. In addition, there is no obvious way to make use of organisms other than human, such as mouse, for which there are no obvious reference SNP panel that can be immediately used for all the mice strains.

Clarification of BAF Integration into CNV Calls (From previous revision): We agree that the pairwise comparison of the scales in BAF and expression signals were not clearly explained in our original manuscript. Based on this suggestion, we included, in the previous revision, the details of the pairwise comparison of scales from BAF and expression signals in Methods section (**new text line 678-797**). We also include the details of comparison here:

Step 1. Integration of the HMM segment states with Allele Frequency Shift Information. CaSpER algorithm outputs CNV calls using all the pairwise comparisons of expression signal and BAF signal decompositions. We describe below step-by-step details of the comparison procedure.

CaSpER uses two sources of information. First is the genomewide expression signal and other is the genomewide allelic shift signal. The expression signal refers to the vector whose elements

are the expression levels of all the genes along the genome. a^{th} element in expression signal corresponds to the a^{th} gene along the genomic coordinates starting from the beginning of the genome. Similarly, allelic shift signal refers to the vector whose elements are the absolute value of the B-allele frequency (BAF) shift of all SNVs identified by CaSpER such that the SNVs are sorted with respect to genomic coordinates. In bulk RNA-seq data, the BAF shift signal is computed independently for each sample. For single cell RNA-seq data, the BAF shift signal is computed using the reads pooled from all the cells (See Methods for detailed explanation of BAF shift computation). The reason for the difference in computation of the BAF shift signal in bulk and single cell RNA-seq is that the single cell RNA-seq is very sparse. In order to increase the power to detect BAF shift events (i.e. LOH events), we pool the reads and use the pooled reads to compute the BAF shift signal.

CaSpER starts by computing the multiscale decomposition (smoothing) of the expression signal and the BAF shift signal. After the smoothing, we get the scale-by-scale smoothed expression signals and scale-by-scale smoothed BAF shift signals. For brevity, we refer to the scales in the decomposition of the expression signal by “expression scale”. We denote the expression scales with n . Similarly, we refer to the scales in the decomposition of the BAF shift signal by “BAF scale” and denote them with m .

The smoothed expression signals are segmented using the hidden Markov Model (HMM). The HMM detects the segment boundaries in expression signal and assigns one state to each segment. The segments detected by the HMM-based segmentation of expression signal (at scale n) are filtered with respect to concordance with the BAF shift signal. For this, we assign the segments whose HMM states are 1 and 5 the deletion and amplification calls, respectively. In addition, the segments with assigned states of 2 or 4 are assigned deletion and amplification calls, respectively, given that there are an accompanying BAF shifts. The details of the BAF shift test are explained in revised Methods Section Named “*Calculation of BAF shift threshold using Gaussian mixture models*”.

To be more concrete, we summarize this procedure below:

Let us assume that the HMM segments from the smoothed expression signal at scale n consists of K segments that are denoted by $X_i^{(n)}$ ($i = 1, \dots, K$). $X_i^{(n)}$ is the i^{th} segment and n is an integer in $[1, N]$, where N denotes the index for the highest smoothing scale for expression signal. We denote the set of all segments at scale n with $X^{(n)}$. Each $X_i^{(n)}$ is assigned by the HMM one of the 5 integer copy number states. We denote these states with $S(X_i^{(n)}) \in \{1, 2, 3, 4, 5\}$ such that

- 1: Homozygous deletion,
- 2: Heterozygous deletion,
- 3: Neutral,
- 4: One-copy amplification,
- 5: Multi-copy amplification.

The deletion/amplification call for the segment $X_i^{(n)}$, denoted by $CNV^{(m)}(X_i^{(n)})$, at expression scale n and BAF scale m is computed as:

$$CNV^{(m)}(X_i^{(n)}) = \begin{cases} 1, & \text{if } (S(X_i^{(n)}) = 5) \text{ or } (S(X_i^{(n)}) = 4 \text{ and } BAF^{(m)}(\widehat{X_i^{(n)}}) > t) \\ -1, & \text{if } (S(X_i^{(n)}) = 1) \text{ or } (S(X_i^{(n)}) = 2 \text{ and } BAF^{(m)}(\widehat{X_i^{(n)}}) > t) \\ 0, & \text{otherwise} \end{cases}$$

where $CNV^{(m)}(X_i^{(n)})$ denotes the CNV call for $X_i^{(n)}$ such that -1, 1, 0 stands for deletion, amplification, and neutral event states respectively. m denotes the smoothing scale for BAF shift signal and it is an integer in $[1, M]$. Finally, $BAF^{(m)}(\widehat{X_i^{(n)}})$ denotes the median value of the BAF shift signal (smoothed at scale m) on the segment $X_i^{(n)}$. The median is computed as the median value of the smoothed B-allele shift signal on all the SNVs within $X_i^{(n)}$. From above computation, it can be seen that CaSpER assigns deletion or amplification to a segment when the HMM state is 1 or 5 without looking at the BAF signal. When the segment state is 2 or 4, an accompanying BAF shift on the segment is required. As we indicated above, the calculation of the BAF shift threshold t is explained in “*Calculation of BAF shift threshold using Gaussian Mixture Models*” in the Methods Section.

From above computation, CaSpER assigns CNV calls to the segments for all the pairwise comparisons of BAF and expression signal scales, which is in total $(N \times M)$ comparisons. Final step is the harmonization of the CNV calls from all the pairwise comparisons.

Step 2. Harmonization and Summarization of CNV calls from multiple scales and from multiple pairwise comparison of BAF and Expression Signals. The pairwise comparison and assignment of CNV calls generates a large number of per-scale information that must be summarized such that each position of the genome is assigned a final call about its CNV status, i.e., deletion/amplification/neutral. We use a consistency-based approach for harmonizing the pairwise comparisons: The events are put together and we assign the final CNV for a gene or large-scale event if the CNV calls are consistent among at least a certain number of pairwise scale comparisons. CaSpER harmonizes and summarizes the CNV calls by dividing them into **large-scale**, **gene-based**, and **segment-based** CNV calls as described below:

1. Large-Scale CNV Summarization. For each pairwise comparison of expression smoothing scale n and BAF shift signal smoothing scale m , $CNV^{(m)}(X_i^{(n)})$, the union of the deletion (and amplification) events in every chromosome arm are computed. Next, from the union, we identify the chromosome arms for which the deletions (or amplifications) are affecting more than one-third of the chromosomal arm³. This way we assign a large-scale CNV call to every chromosome arm for each of the $N \times M$ pairwise scale comparisons. Each chromosome arm gets assigned $N \times M$ large-scale CNV calls.

Next, for each chromosome arm, we ask whether the large-scale CNV call is consistent among at least γ of the $N \times M$ large-scale CNV calls. If there is consistency in more than γ comparisons, we assign the final large-scale CNV call of the chromosome arm as the consistent call. If there is no consistency, we assign a neutral CNV call for the chromosome arm. This procedure is repeated for every sample (or cell). We summarize the consistency based large-scale CNV calls in a matrix where rows are the samples (or cells) and columns are the chromosome arms. The matrix entry of 0 corresponds to no alteration, 1 corresponds to amplification and -1 corresponds to deletion.

2. Gene-Based Summarization. Similar to the large-scale summarization, we generate a matrix where rows are the samples (cells) and **columns are the genes**. The matrix entry of 0 corresponds to no alteration, 1 corresponds to amplification and -1 corresponds to deletion. If

an alteration is consistent in more than γ scale comparisons (out of $N \times M$ comparisons), we report that alteration event for that sample.

3. Segment-Based Summarization. The segments-based summarization aims at generating a final set of CNV calls for a final set of segments that are computed by comparison of scales. We first compare the segments from different expression scales and generate the consistent set of segments. For this, we first identify the final set of segments. To generate the final set of segments, we first pool the ends of all the segments, $X^{(1)}, \dots, X^{(N)}$ from all the scales 1 to N . We then sort the segment ends with respect to genomic coordinates, then we generate a new final segments as the regions between consecutive segment ends (See Figure below). We denote these final set of segments by Y . Note that there is no scale notation in the final set of segments since these are identified using segments from all the scales. A hypothetical example for detection of Y using segments from 3 different scales is shown below:

Updated Supplementary Figure 27. A hypothetical example for detection of final consistent segments.

Note that for any final segment $Y_j \in Y$, there is exactly one segment in $X^{(n)}$ that intersects with Y_j . With this in mind, next step is assignment of CNV calls to the segments in the final segment set. For each segment $Y_j \in Y$, we assign $N \times M$ CNV calls using $CNV^{(m)}(X_i^{(n)})$. For this, we take Y_j , identify the N scale-specific segments, i.e., $X^{(n)}, n = 1, 2, \dots, N$, that intersect with it, then pool M CNV calls from each scale-specific segment. At the end of this procedure, we assign $N \times M$ CNV calls to each Y_j . The final step is assignment of the final CNV call for Y_j . This is performed similar to the consistency-based assignment we used before: For each Y_j , if there are more than γ consistent CNV calls among $N \times M$ CNV calls, we assign the consistent CNV call to Y_j . When there is no consistency among the calls, we assign a neutral CNV state to Y_j .

Selection of γ parameter. γ represents minimum number of consistent CNV calls (Out of $N \times M$ comparisons of expression scales and BAF scales) while assigning a final CNV (amp/del/neutral) call to a segment/gene/chromosome arm. To visualize the effect of changing γ on different datasets, we computed the large-scale and gene-based event summaries of CNV calls for several datasets and calculated true positive rate (TPR) and false positive rate (FPR) using genotyping arrays as gold standard. We describe TPR and FPR calculation in “Performance Metrics” in the Methods section. As it is seen in the below plots, γ parameter tunes the tradeoff between FPR and TPR rates where low (high) γ implies high (low) TPR and high (low) FPR. In the manuscript, to balance FPR and TPR rates we used 7 as a γ threshold. The user can change γ threshold to be more stringent (higher γ) or relaxed (lower γ) in terms of consistency among the pairwise scale comparisons.

Bulk RNA-Seq SNV Filtering and Related Thresholds: In principle, the BAF shift in cancer sample must be computed using the SNVs in the matching normal tissue such as blood or tissue surrounding the tumor. This way, the BAF shift in tumor samples are computed with respect to exactly the normal tissue's baseline allele frequency:

$$BAF_{shift}(SNV_i) = |BAF_{tumor}(SNV_i) - BAF_{normal}(SNV_i)|$$

where SNV_i denotes the i^{th} SNV and $BAF_{tumor}(SNV_i)$ denotes the alternate allele frequency of SNV_i for the tumor sample. For the SNVs that reside on a copy neutral LOH or a CNV, this value will show deviation from 0. Otherwise it will be distributed around 0.

We cannot compute the above quantity directly because we do not have access to normal RNA-sequencing dataset. For whole exome and whole genome sequencing of tumor, the normal tissue is sequenced regularly (almost as a standard protocol) but RNA-sequencing of matching normal tissue is currently not established as standard procedure. Since we do not have the matching normal sample, we focus on the SNV candidates that are most likely heterozygous (Hence, $BAF_{normal}(SNV_i) = 0.5$) in the healthy tissue. We also have to modify the definition of shift as:

$$BAF_{shift}(SNV_i) = |BAF_{tumor}(SNV_i) - 0.5|$$

As it can be seen, without the normal sample, we just subtract 0.5 from the observed tumor SNV allele frequency (and take absolute value) to compute the BAF shift signal. The most vital component of this computation is to ensure that the SNVs we use are most likely heterozygous in the normal tissue.

The thresholds we select aim at ensuring that the selected variants are likely heterozygous variants in the normal tissue. If we do not apply these thresholds, we observed that the allele frequency shift signals are non-informative to detect the loss-of-heterozygosity and BAF shifts associated with CNVs. We describe the motivation for selecting these thresholds below:

For bulk samples, there is generally high concentration of healthy cells in the tumor tissues (up to 20-30% as shown in the literature⁴). Therefore, if an SNV has a high observed allele frequency (>95%) in the tumor, it is very likely that it is a homozygous SNV in the healthy tissue too. The existence of normal tissue in tumor samples will cause a shift in the observed allele frequency of SNVs in the bulk tumor RNA-seq data.

Assume that there is a 1-copy deletion of a segment and there is 30% normal tissue in the bulk RNA-seq sample: 70% of the cells are tumor cells and they contain only 1 copy of the segment; 30% of the cells are normal cells and they contain 2 copies. For an SNV on this segment who is a heterozygous in the normal cells, the observed allele frequency will be:

$$\frac{70 \times 1 + 30 \times 1}{70 \times 1 + 30 \times 2} = 0.769$$

In the above equation, if the impurity is smaller than 30%, the observed allele frequency increases. To be cautious, we choose to act conservatively, and we assume that there may be around 30% impurity in the tumor samples. Thus, we assume that any SNV with allele frequency higher than around 80% is likely a homozygous variant. We therefore exclude the variants that have higher than 80% allele frequencies (Found by rounding 0.769) and smaller than 20% allele frequencies (the other side of the AF spectrum) since these are likely homozygous events in the normal matching tissue. Note that by doing this filtering, we are likely removing variants that inform us about existence of important LOH and CNV events but we still remove these for being conservative and using only the SNVs that are likely heterozygous in the matching normal sample.

In the above computation, we assume for simplicity that there is a clonal deletion of the segment. If the deletion is subclonal (less than 100% of the tumor cells contain the deletion), the observed allele frequencies will be closer to 50% and they will still be selected in the thresholding process. Similarly, for uneven amplification events, the above fraction will decrease. Thus, for these events, the thresholding will capture the useful SNVs appropriately. In addition, for very high amplifications events, allele frequency will be close to 50% (i.e. shift close to 0). In these cases, CaSpER weighs the expression signal much more than the BAF shift signal. Thus, the thresholding affects these events less.

Single Cell RNA-seq SNV Filtering and Related Threshold: SNV selection procedure for single cell samples is motivated mainly by technical factors around detection of the SNVs. Compared to the bulk RNA-seq samples, there is much smaller number of reads from each cell. As described in the text, we tackle this issue by pooling the reads from all the cells and generate a bulk RNA-seq sample from all the cells. Even though we pool the samples, BAFExtract still detects a relatively small number of SNVs that we can use to compute the BAF shift signal.

In addition, for single cell RNA-seq samples, impurity of RNA-seq sample, i.e. fraction of normal cells in the sample, is much smaller (compared to bulk tumor RNA-seq samples) since the normal cell infiltration can be controlled (using for example FACS) and it is generally much smaller. Thus, the single cell RNA-seq data affected much less from the impurity of the samples. Because the sample is almost pure, we cannot use the observed SNV allele frequencies directly to infer a maximum allele frequency threshold (similar to the threshold we derived in the above equation) for selecting SNVs that are most likely heterozygous in matching normal cells. We therefore include all the SNVs in the analysis except the SNVs whose alternate allele frequencies are smaller than 20%. The reasoning for this lower allele frequency threshold is that these SNVs may be manifestations of technical artefacts such as PCR duplications⁵. This threshold value is a conservative value that is selected by trial-and-error to optimize the quality of the observed BAF shift signal and its consistency with the expression signal. We also included below figure to illustrate the spectrum of allele frequencies of SNVs that are used for bulk and single cell RNA-seq data.

Updated supplementary Figure 24. The illustration of the observed allele frequency ranges (or spectrums) that are used to select the SNVs for computing the BAF shift signal. For bulk RNA-seq, the SNVs whose observed allele frequencies are in 20%-80% range (Illustrated with red box) are selected for computing the BAF shift signal. For single cell RNA-seq, the SNVs whose observed allele frequencies are in the range 20%-100% (Illustrated with blue box) are used for computing the BAF shift signal.

We updated in methods “BAF generation, SNV filtering, and related thresholds” section (new text line 565-643).

"3, In terms of biological results, CaSpER has reported mutually exclusive and co-occurring CNV events in the scRNA-seq GBM dataset (Figure 5). But why these CNV events are significant has not been convincingly discussed."

Author's Response: The reviewer raises an important point that should be clarified. To clarify the significance of the co-occurrence/exclusivity of the CNVs, we discuss the statistical and biological significance of these patterns.

CaSpER identifies the co-occurrence and exclusivity patterns purely based on statistical significance of how likely these events could occur in a randomly distribution of the detected CNVs among samples. For this, CaSpER utilizes Fisher's test to assign the statistical significance of the pattern. In certain scenarios, the statistical significances may not fully reflect the biological significance of the events. For example, in a situation where the somatic CNVs do not uniformly occur along the genome, the statistical significance may not reflect existence of an interesting biological phenomenon but simply reflects the mismatch between assumed background distribution of CNVs. Secondly, the statistical significance of co-occurrence and exclusivity patterns may not correspond to a causal driver event. On the contrary, it may reveal a consequence of other drivers such as existence of mutations in certain DNA repair genes in one clone compared to the other. Thus, these patterns must be scrutinized further by functional assays and with biological knowledge.

The mutual exclusivity and co-occurrence of variants has been used by many studies to study bulk sequencing of tumor samples with respect to different genetic backgrounds and processes^{6,7}. For example, CBioPortal website is a popular website dedicated to analyzing large cancer cohorts for analysis of co-occurrence and exclusivity patterns in somatic variants among multiple tumor samples by generating ***OncoPrint plots***⁸. ***The application of mutual exclusivity and co-occurrence in the context of single cell RNA and DNA sequencing*** has not been studied extensively but it has been analyzed by different studies. We believe CaSpER fills an important gap by providing these plots so that single cell experiments can be evaluated in terms of co-occurrence and exclusivity patterns.

For biological significance of the events, we would like to highlight how the co-occurrence and exclusivity of CNVs relate to and impact the clonal architecture of the tumor. In tumors, different clones evolve over time and each of these clones contain different driver events. Interactions between these clones can affect disease progression and treatment outcome. Some of those clones have fitness advantage in the context of treatment and some may individually be capable of giving rise to metastasis. Therefore, correctly identifying subclones and the driver events in those clones is very important. In one tumor sample, there might be clonally exclusive CNV events meaning that these events happen in two different branches of tumor phylogeny. This specific clone configuration with clonally mutually exclusive CNV events confers a selective advantage, most probably through synergies between the clones with these variations.

We updated manuscript including these points (**new text line 442-451**).

"Reviewer #3 (Remarks to the Author): In this revision, the authors have made a number of key clarifications that have substantially improved the manuscript. In particular, the authors have provided new benchmarks of CaSpER's ability to accurately identify CNVs in bulk RNA-seq data at different length-scales, clonalities, and parameter ranges. The authors

also provide new analyses that compare CaSpER with previously published approaches, HoneyBADGER and inferCNV, again using bulk RNA-seq data. The authors have also extensively improved the Methods section with new details of how CaSpER works.

However, a number of critical outstanding issues remain unclear or were insufficiently addressed in the revision that preclude publication. While CaSpER's capacity for identifying CNVs in bulk RNA-seq data have been sufficiently demonstrated, it remains unclear how CaSpER performs, especially as compared to HoneyBADGER and inferCNV, with respect to single-cell RNA-seq data. The authors fail to provide thorough performance characterization of CaSpER using single-cell RNA-seq data and it remains unclear how suitable CaSpER is for detecting CNVs at different length-scales or clonalities in single-cell RNA-seq data. Likewise, the primarily visualization capabilities of CaSpER appear identical to those of inferCNV and HoneyBADGER so CaSpER's utility as a visualization tool remains to be convincingly shown."

Author's Response: We thank the reviewer for constructive comments and criticism. We have now updated the manuscript to address these comments. In particular, reviewer's comments are centered around performance benchmarks on single cell RNA-seq data and utility of visualization tools that CaSpER provides. We would like to make some early remarks about these so that we can clarify them more coherently and in more detail in our later responses.

Performance benchmarks on single cell RNA-seq Data are challenging because of the lack of known copy number variants in single cells (i.e., ground truth) matching to the scRNA-seq data. But we performed them using DR-Seq data: For performance benchmarks in scRNA-seq data, we need simultaneous measurement of RNA and DNA (or copy number) in individual cells. This is the main reason we could not directly compare HoneyBADGER and CaSpER using the single cell RNA-seq data from Patel et al. 2014 study because this study does not contain the known CNV calls for the cells.

While there are methods to perform simultaneous DNA-RNA sequencing from individual cells, they are very new and are not high throughput. Another major obstacle for performing benchmarks is that we also require the ground truth data to be in cells with numerous CNVs (tumors are the best case for this) in their genomes so that we can perform reliable benchmarks. As we highlight below, we found that DR-Seq and G&T-Seq datasets as fairly well-cited and relatively well-established methods. Among these, DR-Seq technology has data that is publicly available and we used this data for performance comparison. **In general, we observed that HoneyBADGER is very specific and exhibits low sensitivity for detection of CNVs in this single cell data. This is similar to the observation we made for bulk RNA-seq dataset. In comparison, CaSpER exhibits much higher sensitivity on DR-Seq dataset.** We have failed to convey the above points in the previous revision and this is a miscommunication that is caused by us. We hope that we will be able to clarify these in the current revision.

In the previous revision, we did perform single cell RNA-seq simulations for measuring CaSpER's performance with changing clonality and length scales (Supplementary Figure 15).

We also discuss below the utility of visualizations that are provided by CaSpER.

"Major comments:

1. While the application of CaSpER to the bulk RNA-seq of breast cancer samples from TCGA is promising, its performance for single-cell RNA-seq data, particularly as compared to previously published approaches, still requires further clarification and fails to demonstrate improvement over previously published approaches. For example, what is the accuracy of identifying focal segments at different length scales in single-cell RNA-seq data? It cannot be assumed that CaSpER's performance using bulk RNA-seq data will translate to comparable levels of accuracy when applied to single-cell RNA-seq, particularly since different thresholds are used."

Author's Response: We thank the reviewer and we understand the reviewer's concern.

As we discussed above, the performance benchmark on single cell RNA-seq data is not straightforward because we need to have access to known copy number variants in individual cells from which we identify the CNVs using scRNA-seq data. For this, we used the data generated by the DR-Seq technology² which is one of the major simultaneous RNA-DNA sequencing techniques. In the data, the DNA and RNA from 7 breast cancer cells are sequenced. The authors of DR-Seq study provide the copy number variant calls and expression quantifications for each cell. Another major technique for RNA-DNA sequencing is G&T-Seq¹, but we could not gain timely access to the data generated by this technology since the data is under restricted access. It should also be noted that these experiments are not standardized, unlike the genotyping arrays where we can directly identify known CNV calls. Thus, we rely on the absolute copy number variants that are provided by the DR-Seq study to evaluate algorithmic performance.

CaSpER shows good concordance with the DR-Seq Data: The authors used a computational approach and assigned the absolute copy number to the segments. We first evaluated how the copy number states from CaSpER correlate with the absolute copy number assigned to each gene. We show below, for each cell, the distribution of the DR-Seq assigned copy numbers corresponding to the HMM copy number states. We observed a fairly high consistency between the HMM copy number state and the DR-Seq assigned copy number.

Updated Supplementary Figure 16. Figure shows for each cell in DR-Seq dataset the HMM state (x-axis) and the absolute copy number assigned to the genes (y-axis). Box plots indicate the distribution of absolute copy number assigned to each gene, for the corresponding copy number state.

In the above figure, the HMM states 1 and 2 correspond to the deletion states and the states 4 and 5 correspond to the amplification states.

Comparison of CaSpER and HoneyBADGER using scRNA-Seq data. We next performed comparison of CaSpER and HoneyBADGER. To generate the ground truth CNV calls for each cell, we first computed the ploidy in each cell using following:

$$ploidy_i = \sum_j \frac{((Absolute\ CN\ of\ Segment\ j\ in\ cell\ i) \times (Length\ of\ Segment\ j\ in\ cell\ i))}{\sum_k (Length\ of\ Segment\ k\ in\ cell\ i)}$$

where $ploidy_i$ denotes the average DNA content in cell i and *Absolute CN of Segment j in cell i* denotes the absolute copy number (integer as large as 60) assigned by DR-Seq study for segment j for cell i . We next assigned relative amplification and deletion to each cell using $ploidy_i$:

$$CNV_j = \begin{cases} Amplification, & \text{if } (Absolute\ CN\ of\ Segment\ j) > ploidy_i \\ Deletion, & \text{if } (Absolute\ CN\ of\ Segment\ j) < ploidy_i \\ Neutral, & \text{if } (Absolute\ CN\ of\ Segment\ j) = ploidy_i \end{cases}$$

where CNV_j denotes the relative copy number value assigned to segment j in cell i . As we explained before, we had to perform the relative copy number assignments because both

CaSpER and HoneyBADGER assign copy numbers relative to the ploidy of the cell because these tools assign relative copy numbers. We next used the relative copy numbers to assign the average large scale false positive rate (FPR) and true positive rate (TPR) to the CNV calls generated by CaSpER and HoneyBADGER using the single cell RNA-seq data among 7 cells.

On Single Cell RNA-Seq benchmark, HoneyBADGER performs with much lower sensitivity compared to CaSpER: We used the expression quantifications from DR-Seq data as input to CaSpER and HoneyBADGER. We evaluated the TPR and FPR for the detected large-scale deletion and amplification events. CaSpER achieves 32% TPR and 1.6% FPR in deletion events and 49% TPR and 3.8% FPR in amplification events. If we relax the parameters (with $\gamma=1$), CaSpER achieves 45% TPR and 3% FPR in deletion events and 62.6% TPR and 8.6% FPR in amplification events.

HoneyBADGER could not detect any CNV events even though the ground truth data has many amplification and deletion events. We also ran HoneyBADGER with numerous combinations of parameter configurations (clustering depth, inclusion of all genes in analysis, different approaches for data normalization) but these did not change the result from HoneyBADGER (**new text line 318-342**).

In summary, the bulk and single cell RNA-seq comparisons indicate that the CNV calls generated by HoneyBADGER exhibit very low sensitivity compared to CaSpER.

Impact of Varying Clonality and Changing Length of CNVs. In the previous revision, we did perform several single cell RNA-Seq simulations for evaluating the performance of CaSpER with changing clonality and length scales. These are included in the **Supplementary Figure 15**. As expected, the sensitivity increases significantly with the increasing clonality. We did not observe a strict relation between the sensitivity and the deletion length.

"2. In the updated figure 3C as referenced in the response to reviews (5E in the manuscript - "Clustering of large scale events generated by CaSpER in patient MGH31"), CaSpER appears to identify CNVs (amplification on 6q, deletions on 5q, 6p, 10p, 10q) in the putative normal cells in patient MGH31. This seems to be a much higher false positive rate than what would be expected based on bulk-RNA-seq benchmarks (which suggest an FPR of 0.3%). Likewise, there appear to be a number of false negatives, where clonal alterations not being identified by CaSpER in the cancer cells. This suggests that CaSpER's performance in terms of TPR and FPR rates are likely different for single-cell RNA-seq data and should be characterized in order to substantiate the conclusion that CaSpER is suitable for accurate identification of CNVs in single-cell RNA-seq data."

We thank the reviewer for this comment. We have stated in our manuscript that CaSpER achieves 0.3% FPR for detecting large scale deletion events in bulk meningioma RNA-Seq data. Whereas, in TCGA-BRCA bulk RNA-Seq data, CaSpER achieves 3.8% FPR for detecting large-scale amplification events, 7.7% FPR for detecting large scale deletion events. In TCGA-GBM bulk RNA-Seq data, CaSpER achieves 2% FPR for detecting large-scale amplification events, and 3.5% FPR for detecting large-scale deletion events. As it is seen from the above results, we have different FPR values for different datasets. We agree with the reviewer that we do not expect any large scale CNV calls in immune cells. CaSpER achieves 1.7% FPR rate for deletion events in immune cells and 0.56% FPR rate for amplification events. We cannot calculate TPR in immune

cells since ground truth for immune cells is without any CNV events. Based on these observations, we believe bulk RNA-Seq FPR rates are compatible to single cell RNA-Seq datasets.

"3. To assess the CNV-calling capabilities of CaSpER compared to previously published approaches, the authors provide a new performance comparison of CaSpER to HoneyBADGER (Fan et al, Genome Research 2018), using bulk RNA-seq data. However, as the authors note, HoneyBADGER was developed for CNV-calling for single-cell RNA-seq data and thus conclude that HoneyBADGER is not suitable for detection of CNVs for bulk RNA-seq data as expected. So it is unclear why the performance comparison was done using bulk RNA-seq data when single-cell RNA-seq data was also readily available, particularly since both papers have analyzed the same GBM data from Patel et al (Science 2014)."

Author's Response: We thank the reviewer for this comment and we understand the confusion of the reviewer. We have presented the single cell RNA-seq benchmarks in our response to the Comment 1. We respond to the additional comments below.

Patel et al. does not have ground truth CNV information, i.e., there is no known matching CNV information for the cells in Patel et al. study. To be able to perform single cell RNA-seq performance benchmarks, **we require independently generated CNV information for each cell.** For bulk RNA-seq data, the genotyping arrays represent independent measurement of genomic copy numbers from which we infer CNV's. Thus, we used these data as known CNV states for the bulk RNA-seq data. On the other hand, Patel et al. study does not have known CNV information for individual cells. This is why we could not perform benchmark on Patel et al. study. We understand the confusion that reviewer rightly pointed out and we hope that this clarifies why we could not use Patel et al study for an objective benchmark.

Single-cell DNA sequencing is another new powerful approach for understanding the genomic diversity of tumor clonal architecture^{9,10}. Moreover, a number of approaches for simultaneously assaying DNA and RNA information from the same single cell has been developed. However, it is still technically challenging to assay both the genome and transcriptome from the same cell^{1,2}, particularly in terms of cost and labor. As we presented in response to comment 1 above, we used the DR-Seq dataset for performing single cell RNA-seq performance assessment. Even though DR-Seq (and other techniques such as G&T-Seq) represents simultaneous extraction of DNA and RNA, these techniques are still in their infancy and can probe only a small number of cells and small number of genes.

"4. The suitability of CaSpER for usage with 3'-only RNA-sequencing techniques remain unclear. While the authors provide new speculation as to how CaSpER may perform worse with 3'-only sequencing, they provide no new analysis demonstrating such decreased performance even though 3'-only sequencing data of cancer samples are available. For example, the authors speculate that BAF will be sparser from 3'-only sequencing, and fail to support such speculation with data, even though such data are available."

Author's Response: We thank the reviewer for raising an important point. In the previous revision, we have only commented on the fact that 3'-only sequencing (Such as 10X single cell RNA sequencing technologies) and we did not provide any analysis with our response. This was because we assumed the reviewer asked only for comments on this point.

We first searched for a study where we could obtain same single-cell cancer RNA-Seq dataset produced across two different technologies. We explicitly searched for cancer datasets to be able to detect large scale deletion and amplification CNV events. Unfortunately, we could not find any study with same single-cell RNA-Seq produced across two technologies. Therefore, we have analyzed two different datasets to make a comparison for read coverage and B-allele shift over the transcripts. The two datasets are distinct in terms of the technology such that one is generated by Smart-Seq2 full transcript sequencing (MGH31 sample from Patel et al.)¹¹ and the other is generated by 10X Chromium technology (SA1N sample from Kumar et al.)¹². Kumar et al. study generated single cell RNA-sequencing data from mouse tumor models to study cell-cell interactions. As the representative 3' sequencing data, we used the data for the sample with id SA1N from this study (GEO Accession number GSE121861) and mapped the reads using CellRanger software. As the representative full transcript sequencing, we used MGH31 sample's mapped reads as we have them available. For both samples, we first computed the read coverage along the genome. Next, we extracted the coverage signal over protein coding gene exons. For MGH31, we used the latest GENCODE annotations and merged the exons for each gene, then concatenated the exons (from 5' to 3') to build the signal over each gene's exonic regions. This way, we generated the RNA-seq signal over each gene's concatenated-exons that represents the concatenation of RNA-seq signal over only the exons of each gene. We finally aggregated these exonic RNA-seq signals over all the genes. Since different genes have different total exonic lengths, we normalized the length of each gene's exonic signal to the length of the longest exonic gene in the annotation. For SA1N, we performed the same aggregation using latest Mouse GENCODE annotations. We show the aggregation of signal for MGH31 (Smart-Seq2 full transcript sequencing) and for SA1N (10X 3' sequencing) below:

Updated Supplementary Figure 22. Comparison of read depth on exon-concatenated RNA-seq profiles. Left plot shows the aggregation of 3' sequencing read depth. Right plot shows the aggregation of read depth on full transcript sequencing over the exon-concatenated genes. X-

axis shows the normalized position on the concatenated exons. Y-axis shows the read depth. X-axis is oriented from 5' to 3'.

It can be seen that there is very high 3' read depth bias in 3' sequencing. For full transcript sequencing, we do observe slight 3' bias. However, in comparison to 3' sequencing, full transcript sequencing exhibits a much less bias when 5' and 3' ends are compared in terms of read depth coverage.

We next used BAFExtract to generate candidate SNVs from which we compute BAF shift. We computed the candidate SNV density over all the genes and aggregated the SNV density over every position on the genes (from 5' to 3'). As with read depth aggregation, we normalized the length of genes to the longest gene in the annotations. These aggregations are shown below:

Updated Supplementary Figure 23. Comparison of SNV density on exon-concatenated genes as extracted by BAFExtract. Left plot shows the SNV density from the 3' sequenced SA1N sample. Right plot shows the SNV density from the full transcript sequenced MGH31 sample over the exon-concatenated genes. X-axis shows the normalized position on the concatenated exons. Y-axis shows the read depth. X-axis is oriented from 5' to 3'.

It can be seen clearly that the candidate SNVs are enriched around the 3' end of genes for 3' sequencing data. Full transcript sequencing data, on the other hand shows a much more uniform coverage of the transcripts. This result indicates that the BAF shift can be reliably quantified around the ends of genes for 3' sequencing. On the other hand, full transcript sequencing can give a more uniform and comprehensive measurement of the BAF shift signal by covering the SNVs more uniformly along the genes with less bias on the location of SNVs.

Importance of Read Depth. It should also be noted that above arguments are contingent on the fact that the suitability of CaSpER for 3' technologies will be affected by the read depth of sequencing. Although 3' sequencing covers the 5' ends of genes to much lesser extent than 3' ends, there is still some coverage on the 5' ends of sequences as seen in above plots. If both technologies are used with decent coverage, the read coverage over the SNVs (even around 5' ends of genes) can provide similar power to measure BAF shift. Consequently, both technologies can provide useful BAF information with decent sequencing depths.

In the example above, we have observed that we generated similar number of candidate SNVs for measuring BAF shift. Thus, we have now updated our original comments that CaSpER can be potentially effectively used with both technologies when there is decent coverage of genes.

We updated the Supplementary Information to reflect the above points. We also added new Supplementary Figures 22 and 23).

"5. Although the authors suggest that visualization capabilities are a distinguishing feature of CaSpER compared to previously published approaches, the main heat map visualizations are essentially identical to visualizations provided by HoneyBADGER and inferCNV. For example, the new updated Supplementary Figure 16 and 17 actually highlight how similar visualizations in CaSpER and HoneyBADGER are (and to a lesser extent inferCNV) with the only appreciable difference being that CaSpER scales chromosomes by size and uses a different color scheme. Similarly, CaSpER's plotHeatmap function visualizes genome wide gene expression signals at different smoothing scales. However, parameters in both inferCNV and HoneyBADGER can be passed in order to visualize different smoothing scales. Inclusion of these parameters doesn't justify presenting CaSpER as a visualization tool. Likewise, HoneyBADGER provides allele-level visualizations that offers comparable information and insights to CaSpER's BAF shift plots.

The only visualization unique to CaSpER appear to be the graph of mutually exclusive and co-occurring events. However, it remains unclear how this visualization is useful and will allow users to gain additional insights that are not apparent from the heat map visualizations common to all 3 approaches. On a more fundamental level, it is unclear why users would want to visualize the relationship among CNV events. More importantly, it is unclear how these relationships among CNV events fit onto the proposed phylogenetic relationship among cells proposed by CaSpER's plotSCellCNVTree function."

Author's Response: We thank the reviewer for the comment. We believe that we did not clearly state our view on the visualization of CaSpER. We clarify these points below.

We have developed and presented the visualizations of CaSpER to make sure it analyzes and visualizes the data as a comprehensive and complete analysis tool. This requires us to evaluate the full set of visualization options that can be generated from the data. We do not want to create the impression that we claim that CaSpER is the only tool to generate these visualizations. But we would like to point out that we believe that CaSpER generates a more complete set of visualizations compared to other tools that can be used to build further biological hypothesis to be tested. This enables users to analyze and characterize their samples better.

We discuss these points in more detail below.

BAF Shift and Expression Visualizations: We agree with the reviewer that heatmap visualizations generated by CaSpER are similar in flavor to the visualizations generated by HoneyBADGER and inferCNV. For these visuals, CaSpER is different from other tools as it generates a multiscale view of the data by smoothing it at different length scales. As we presented

in the Methods Section, the multiscale smoothing is not a simply linear smoothing of the data, but median filtering, which deals with noise and preserves edges in the signal better than linear and Gaussian filtering techniques¹³. From this perspective, while reviewer's statement that "*parameters in both inferCNV and HoneyBADGER can be passed in order to visualize different smoothing scales*" is correct, neither inferCNV nor HoneyBADGER currently has a way to present the data in a multiscale smoothed manner. In fact, HoneyBADGER does not have an explicit parameter that we know of that a user can change to smooth the data at different length scales. In addition, CaSpER generates the visualizations for multiscale smoothed BAF shift signals using iterative median filtering based smoothing strategy.

Large Scale Event Summarization Plots: Another visualization that is unique to CaSpER is the summary plot of large-scale CNV events among all the samples/cells. Large scale event summarization is useful for summarizing the detected large-scale CNV events (deletions and amplifications) over multiple samples. This plot summarizes the large scale CNVs and may reveal the patterns that may otherwise be missed when data is visualized at smaller scales.

Mutual Exclusivity and Co-occurrence Plots and Phylogenetic Tree Plots: The reviewer indicates these plots are not obviously useful and that there is no clear reason why users would want to visualize these patterns. We do not necessarily agree with this statement because the mutual exclusivity and co-occurrence of variants has been used by many studies to study tumor samples with different genetic backgrounds and processes^{6,7}. For example, CBioPortal website is a popular website dedicated to analyzing large cancer cohorts for analysis of co-occurrence and exclusivity patterns in somatic variants among multiple tumor samples by generating **OncoPrint plots**⁸. **The application of mutual exclusivity and co-occurrence in the context of single cell RNA and DNA sequencing** has not been studied extensively but it has been analyzed by different studies.

In single cell sequencing of a tumor, the exclusivity patterns can potentially reveal different clones within a tumor that have gained evolutionary advantage in different trajectories. Similarly, the co-occurrence of CNVs can reveal clones that require existence of certain variants to exist in the same cell to gain advantage for survival. Thus, these patterns can be used to understand the genetic processes that underpin the survival of tumor cells. It must be noted that since CaSpER can analyze bulk RNA-seq datasets, the visualizations can be generated using bulk RNA-seq data directly to increase their utility.

The mutual exclusivity and co-occurrence patterns are computed by pairwise comparison of detected CNVs. CaSpER also uses the whole set of inferred CNVs and build the phylogenetic tree-based clustering and visualization of the cells based on the CNV events. This plot can potentially provide the user a way to visually inspect the clonal architecture of the cells inferred using only the CNV events.

To clarify the reviewer's question about how co-occurrence and mutual exclusivity relates to phylogenetic tree:

1. The mutual exclusivity events are represented among different branches of the generated phylogenetic tree
2. The co-occurrence events are represented within the same branches of the phylogenetic tree.
3. The phylogenetic tree evaluates all of the CNV calls and uses them in a hierarchical clustering and these information can provide important insight to build new hypothesis

The mutual exclusivity, co-occurrence, and phylogenetic tree visualizations can provide clues about how different clones evolve over time and each of these clones contain different driver

events. Interactions between these clones can affect disease progression and treatment outcome. Some of those clones have fitness advantage in the context of treatment and some may individually be capable of giving rise to metastasis. Therefore, correctly identifying subclones and the driver events in those clones is very important. **In one tumor sample, there might be clonally exclusive CNV events meaning that these events happen in two different branches of tumor phylogeny. This specific clone configuration with clonally mutually exclusive CNV events confers a selective advantage, most probably through synergies between the clones with these variations.**

In total, visualization options of CaSpER are designed complementary to its main function of CNV identification in bulk and single cell RNA-sequencing data. For this, CaSpER provides the expression and BAF shift visualizations as other methods and also generates a multiscale view of the data using a novel smoothing procedure. CaSpER also evaluates all the samples (or cells) with respect to pairwise comparison of CNV events to identify mutually exclusive and co-occurring CNV events. CaSpER also analyzes all events jointly to build phylogenetic tree of the cells.

"6. The authors suggest that although they do not have access to experimental validation for confirming the newly discovered CNVs in CaSpER's analysis of glioblastoma cells, supportive evidence can be seen using the visualizations from inferCNV. However, inferCNV's visualization is also based on smoothed gene expression, and thus does not offer orthogonal information for independent validation. A more orthogonal piece of information that could be used to validate these novel CNVs would be allele data. What are the allele information supporting these novel CNVs?"

Furthermore, the authors note that CaSpER achieves a lower PPV and TPR rate for identifying focal alterations compared to large-scale CNVs (based on benchmarks from bulk RNA-seq). How can we be sure that these novel focal CNVs identified in the single-cell RNA-seq data are not false positives, particularly since they affect only a few cells?"

Author's Response: We thank the reviewer for this comment. As suggested by the reviewer we used BAF shift information to find support for these novel CNVs. First, we pooled the cells harboring 14q deletion and calculated BAF shift signal. As it is seen from the below plot, the BAF shift is around 0.5 which indicates the event is clonal. That is expected since we only pooled the cells harboring 14q deletion events. On the contrary, in Figure 5B it can be seen that the BAF shift wiggles around 0.2 and 0.3 (away from 0.5) in MGH31, which indicates that only subclonal cell populations harbor chromosome 14q deletion.

Figure 1. BAF shift signal after pooling all the cells with 14q deletion in MGH31 sample.

Similarly we also pooled the cells harboring 13q deletion and we observed a shift at the end of chromosome 13 around 0.5.

Figure 2. BAF shift signal after pooling all the cells with 13q deletion in MGH31 sample.

We also pooled the cells harboring 8q amplification, 20p deletion, 5q amplification and 1p amplification separately. We could not detect BAF shift in these event. One of the reasons for this could be that only a few cells harbor 5q, 8q or 1p amplification and hence the pooled BAF shift signal was very sparse (only 11 variants in 8q, 10 variants in 5q, 19 variants in 20p and 20 variants in 1p). Another reason could be that a balanced amplification could be happening where a shift is not expected. In fact, we did observe high level amplification (HMM state 5) in chromosomes 1 and 5 which could be suggestive a balanced amplification event.

"The authors emphasize that visualize inspection can confirm these novel CNVs. However, looking at the updated Supplementary Figure 14, it is

unclear to me why 14q was identified as a deletion when equally blue cells can be seen on chromosome 9, 16, and 22, which are not identified as a deletion. Similarly, it appears that the cells in MGH31 have been reordered to cluster the CNVs on 5q:14q versus 1p:13q. Ordering cells by CNVs patterns on 5q:14q appear to separate cells for the 1q:13q visualization. The reliance on visual inspection is very subjective and can be biased based on the ordering of cells."

Author's Response: Supplementary Figure 14 highlights only the several co-occurrence and mutually exclusive patterns in the expression levels and does not show the full set of CNV events detected in all cells. As per reviewer's question, we assessed the calls on chromosomes 9, 16, and 22 and saw that only chromosome 9 harbors CNV calls in MGH31 sample. While we do agree that there are some blue colors on these chromosomes, they do not pass the thresholds to be called as deletions by CaSpER.

Secondly, the reviewer finds the ordering subjective. While the visualization could be deemed subjective, ***it must be made clear that the mutual exclusivity of CNV events are detected using one-sided Fisher's exact test using a contingency table built by counting the number of cells that harbor and that do not harbor CNVs.*** We do not rely on visual inspection for detecting the mutual exclusion and co-occurrence events. The reviewer is correct that 5q:14q ordering of cells generates a visualization that exhibits the 1p:13q mutual exclusivity and this is also reported in the mutually exclusive CNVs that CaSpER identifies. But these analyses are performed using the Fisher's exact test and not by visual inspection.

Thus, the re-ordering of the cells is performed only for visualization of the co-occurrence and exclusivity patterns. It is worth noting that Patel et al. study also represents these patterns using similar ordering of data in Figure 1B, where cells of MGH31 sample are ordered by 5q:14q CNV patterns. In Patel et al. study, the authors conclude "Chromosomal aberrations were relatively consistent within tumors, with the exception that MGH31 appears to contain two genetic clones with discordant copy-number changes on chromosomes 5, 13, and 14.", where discordancy refers to mutual exclusivity of the CNVs.

"Minor comments:

1. In the software's website, tutorials are provided as .R files that a user must execute. Please provide these as .Rmd files like how the main documentation is done or as compiled PDFs."

Author's Response: We agree with the reviewer that the main documentation might not have been clearly documented and annotated. In the previous revision, we provided the Rmd files at our github page (<https://github.com/akdess/CaSpER>) but we failed to clearly explain where they are located. The Rmd files can be accessed from <https://github.com/akdess/CaSpER>. Now, we also added documentation in Rmd format for the R file tutorials.

"2. In MGH31, there appears to be a (few?) cells that harbor both 5q amplification and 14q deletion. Are these the same cells that also harbor the 13q deletion and 1p amplification? Or are they different? In order for this CNV pattern to occur, a 5q amplification or a 14q deletion will

have had to occur in two independent sub clones, which seems unlikely. Or is this a false positive?"

Author's Response: We greatly appreciate the reviewer's close inspection of the results. However, we are somewhat confused by reviewer's comment. In our inspection of the MGH31 sample, we could identify that there are only two cells that harbor both 5q amplification and 14q deletion. A small number of overlap is expected since the mutual exclusivity pattern between 5q-amp and 14q-del variants is detected and quantified statistically. The two cells could represent a clone but we need a larger cohort of cells to achieve the statistical power to detect such a small clone.

Also, most of the cells that harbor 5q amplification harbor also 1q amplification and most of the cells that harbor 13q deletion also harbor 14q deletion. We apologize if we are misunderstanding reviewer's question and would certainly try and clarify any issues related to this analysis.

"3. In figure 5E, a number of cells appear to harbor 7q and 6q amplifications, but both of these CNVs are not included in 5F. Yet a fewer number of cells, the putative sub clone, appear to harbor 1p and 5q amplifications, which are included in 5F. Again, it is unclear how these relationships among CNV events fit onto the proposed phylogenetic relationship among cells."

Author's Response: We thank the reviewer's for close inspection of the co-occurrence results. As we stated above, the mutual exclusivity or co-occurrence of CNV events are detected using one-sided Fisher's exact test using a contingency table built by counting the number of cells that harbor and that do not harbor pairs of detected CNVs. Figure 5F shows significant mutually exclusive and co-occurring event pairs with a Fisher's exact test p-value less than 0.01. Co-occurrence Fisher's exact test p-value for 7q and 6q amplification event pair is 0.3. Mutual exclusivity test p-value for 1p and 5q amplification event pair is 0.02. Phylogenetic tree inference is based on the clustering of large scale CNV events in all the cells in a specific patient.

- 1 Macaulay IC, Haerty W, Kumar P, Li YI, Hu TX, Teng MJ, *et al.* G&T-seq: Parallel sequencing of single-cell genomes and transcriptomes. *Nat Methods* 2015;**12**:519–22. <https://doi.org/10.1038/nmeth.3370>.
- 2 Dey SS, Kester L, Spanjaard B, Bienko M, Van Oudenaarden A. Integrated genome and transcriptome sequencing of the same cell. *Nat Biotechnol* 2015;**33**:285–9. <https://doi.org/10.1038/nbt.3129>.
- 3 Clark VE, Erson-Omay EZ, Serin A, Yin J, Cotney J, Özduman K, *et al.* Genomic analysis of non-NF2 meningiomas reveals mutations in TRAF7, KLF4, AKT1, and SMO. *Science (80-)* 2013;**339**:1077–80. <https://doi.org/10.1126/science.1233009>.
- 4 Aran D, Sirota M, Butte AJ. Systematic pan-cancer analysis of tumour purity. *Nat Commun* 2015;**6**:. <https://doi.org/10.1038/ncomms9971>.
- 5 Piskol R, Ramaswami G, Li JB. Reliable identification of genomic variants from RNA-seq data. *Am J Hum Genet* 2013;**93**:641–51.

- <https://doi.org/10.1016/j.ajhg.2013.08.008>.
- 6 Leiserson MDM, Blokh D, Sharan R, Raphael BJ. Simultaneous Identification of Multiple Driver Pathways in Cancer. *PLoS Comput Biol* 2013;**9**:. <https://doi.org/10.1371/journal.pcbi.1003054>.
 - 7 Ciriello G, Cerami E, Sander C, Schultz N. Mutual exclusivity analysis identifies oncogenic network modules. *Genome Res* 2012;**22**:398–406. <https://doi.org/10.1101/gr.125567.111>.
 - 8 Gao J, Aksoy BA, Dogrusoz U, Dresdner G, Gross B, Sumer SO, *et al*. Integrative analysis of complex cancer genomics and clinical profiles using the cBioPortal. *Sci Signal* 2013;**6**:. <https://doi.org/10.1126/scisignal.2004088>.
 - 9 Navin N, Kendall J, Troge J, Andrews P, Rodgers L, McIndoo J, *et al*. Tumour evolution inferred by single-cell sequencing. *Nature* 2011;**472**:90–5. <https://doi.org/10.1038/nature09807>.
 - 10 Pan X. Single Cell Analysis: From Technology to Biology and Medicine. *Single Cell Biol* 2014;**3**:1–10. <https://doi.org/10.4172/2168-9431.1000106.Single>.
 - 11 Patel AP, Tirosh I, Trombetta JJ, Shalek AK, Gillespie SM, Wakimoto H, *et al*. Single-cell RNA-seq highlights intratumoral heterogeneity in primary glioblastoma. *Science (80-)* 2014;**344**:1396–401. <https://doi.org/10.1126/science.1254257>.
 - 12 Kumar MP, Du J, Lagoudas G, Jiao Y, Sawyer A, Drummond DC, *et al*. Analysis of Single-Cell RNA-Seq Identifies Cell-Cell Communication Associated with Tumor Characteristics. *Cell Rep* 2018;**25**:1458–1468.e4. <https://doi.org/10.1016/j.celrep.2018.10.047>.
 - 13 Chan RH, Ho CW, Nikolova M. Salt-and-pepper noise removal by median-type noise detectors and detail-preserving regularization. *IEEE Trans Image Process* 2005;**14**:1479–85. <https://doi.org/10.1109/TIP.2005.852196>.

Reviewers' comments:

Reviewer #2 (Remarks to the Author):

All my comments have been thoroughly addressed. Although I still think the methodological innovation of CaSpER is not strong, the authors successfully show its improvement over existing methods (such as HoneyBADGER) on both bulk RNA-seq and scRNA-seq data (which is also one of the main concern of Reviewer 3). These experimental studies suggest that CaSpER could advance the analysis and visualization of RNA-seq data from tumor samples in studying copy number variation.

Reviewer #3 (Remarks to the Author):

Overview:

In this revision, the authors provide an improved performance benchmark using single-cell RNA-seq data from DR-Seq, which provides both RNA and CNV information, but is limited to 7 cells.

Major comments:

1. The authors benchmark the performance of CaSpER to find that "CaSpER achieves 32% TPR and 1.6% FPR in deletion events and 49% TPR and 3.8% FPR in amplification events." This performance appears substantially poorer to the 95% TPR and 0.3% FPR for bulk RNA-seq. Do the authors have a sense for why this is the case?
2. Do the number of cells affect CaSpER's performance? The benchmark on DR-Seq was done with only 7 cells. Most single-cell experiments these days are on the order of 100s if not 1000s of cells. Do we anticipate the performance of CaSpER to improve with more cells?
3. While I appreciate the new more thorough discussion on the suitability of CaSpER for 3' technologies and the importance of the read depth of sequencing, a more quantitative interpretation of the impact on this difference in read depth and 3' bias on the performance of CaSpER is still lacking. For example, do the authors recommend using CaSpER on 10X data? Do the authors anticipate an X% decrease in performance? Are amplifications affected more than deletions or vice versa?

Minor comments:

1. The authors note in the "Comparing the performance of CaSpER with existing tools" that "We also compared the performance of CaSpER with HoneyBADGER and inferCNV". However, only HoneyBADGER appears to be compared. Please rephrase to reflect the actual comparisons performed.
2. For the bulk RNA-seq data, the authors note that CaSpER "achieves 95% TPR and 0.3% FPR for detecting large scale deletion events" for a meningioma dataset. However, for a more complex cohort of breast cancer datasets, CaSpER achieves a "60.3% true positive rate (TPR) and 3.8% FPR for detecting large scale amplification events, and 79% TPR and 7.7% FPR for detecting large scale deletion events." Could this decrease in performance be attributed to worse tumor purity in the breast cancer data? Are purity estimates available for these bulk RNA-seq samples and are they consistent with the assumption the authors note on line 607 that "we assume that there may be around 30% impurity in the tumor samples."

Reviewer #3 (Remarks to the Author):

Overview:

In this revision, the authors provide an improved performance benchmark using single-cell RNA-seq data from DR-Seq, which provides both RNA and CNV information, but is limited to 7 cells.

Author's Response: We sincerely thank the reviewer for careful consideration of our manuscript. The data generated by the DR-Seq technology is limited to 7 cells. As we discussed in our previous revision, the performance benchmark on single cell RNA-seq data is not straightforward because we need to have access to both DNA and RNA information. Unfortunately, currently, this is the only data we could have access with both DNA and RNA information that allows us to evaluate how the CNV calling methods perform.

Major comments:

1. The authors benchmark the performance of CaSpER to find that "CaSpER achieves 32% TPR and 1.6% FPR in deletion events and 49% TPR and 3.8% FPR in amplification events." This performance appears substantially poorer to the 95% TPR and 0.3% FPR for bulk RNA-seq. Do the authors have a sense for why this is the case?

Author's Response: Reviewer is requesting us to interrogate the reasons behind the performance difference in the bulk and single cell RNA-seq data. Several technical and biological factors are underlying the higher sensitivity of CNV detection for bulk RNA-Seq data. We discuss these factors below.

The first factor is that DR-Seq technology or other simultaneous RNA-DNA sequencing techniques are not standardized, unlike the genotyping arrays where we can directly identify known CNV calls. This would mean that the technical artifacts in the DR-Seq technology are potentially affecting the accuracy of the results. We have discussed the details of DR-Seq technology in our previous revision.

The second factor is that, in DR-Seq study, absolute copy numbers are assigned to the segments using a computational approach. However, both CaSpER and HoneyBADGER assigns relative copy numbers to the detected segments where copy numbers are assigned with respect to the average DNA content in each cell. Therefore, we processed the absolute copy numbers from DR-Seq study to build the relative copy numbers (i.e., amplification, deletion, neutral) in each cell. The difference in calculating relative copy number calls among different datasets might lead to poor performance.

The third factor is that, as we have stated earlier, the CNV calling methods work at the level of genes, which means that the smallest resolution for segment coordinates will be at the gene boundaries. In other words, the regions in the genome that do not contain any genes will not be accessible for calling CNVs on them. Single cell RNA-Seq datasets cover on average 6000 genes genomewide, whereas bulk RNA-Seq dataset measure all 20,000 protein-coding genes. As we have stated in previous revision, most of the segments that are shorter than 1

megabases will be very hard to detect (unless in gene dense regions) because they cover very few genes. We believe that these should be kept in mind while interpreting the accuracy results. This has several implications: We cannot identify the CNV segments that are solely in the intergenic space (no gene overlap) and solely in the introns of genes because there is very little or no RNA-seq signal in them. This impacts the way that the accuracy is estimated because inclusion of inaccessible regions in sensitivity estimation will adversely (and unfairly) decrease the sensitivity of the methods.

The fourth factor is that DR-Seq data is applied to metastatic breast cancer cell line (SK-BR-3) harboring very complicated diverse chromosomal alterations. Following our previous point, the complex CNV profiles of SK-BR-3 cell line is affecting sensitivity. It should, however, be noted that even for the complex ovarian cancer the accuracy of the CNV detection of bulk RNA-seq is around 60% TPR. Therefore, other factors are affecting probably more.

The fifth factor is the limited number of cells. CaSpER will have better estimates with more cells.

Each of the above mentioned factors impacts the discordancy between single cell and bulk RNA-seq CNV detection accuracies. These factors are generally fairly hard to control since matching both the technical and biological covariates require extremely well-characterized samples and very strict sample preparation and sequencing checks.

2. Do the number of cells affect CaSpER's performance? The benchmark on DR-Seq was done with only 7 cells. Most single-cell experiments these days are on the order of 100s if not 1000s of cells. Do we anticipate the performance of CaSpER to improve with more cells?

Author's Response: We thank the reviewer for raising an important point. We anticipate that CaSpER will have better performance with more cells. The number of cells will affect two points. One point is with more cells we will have **better initial parameter estimates for HMM model**. We calculated the mean values of the normal distributions corresponding to 5 copy number states (q1:homozygous deletion, q2:heterozygous deletion, q3:neutral, q4:amplification, q5:high-level amplification) that are derived from randomly selected n=10, n=40 cells and all cells in patient MGH31 of single cell GBM dataset. Mean emission probability estimates are different for n=10, n=40 and all cells and the estimations saturate with an increasing number of cells (Figure 1).

Figure 1: Mean values of the normal distributions corresponding to 5 copy number states (q1:homozygous deletion, q2:heterozygous deletion, q3:neutral, q4:amplification, q5:high-level amplification) that are calculated from randomly selected $n=10$, $n=40$ and all cells in patient MGH31 of single cell GBM dataset.

Another point is that the BAF shift signal is computed by pooling the reads from all the cells. BAF signal extracted from one single cell is very sparse and is not informative. However if we pooled the single cell reads from the same patient together then the patient-specific BAF signal shows shifts in chromosome 7 and 10 (Main Figure 5).

3. While I appreciate the new more thorough discussion on the suitability of CaSpER for 3' technologies and the importance of the read depth of sequencing, a more quantitative interpretation of the impact on this difference in read depth and 3' bias on the performance of CaSpER is still lacking. For example, do the authors recommend using CaSpER on 10X data? Do the authors anticipate an X% decrease in performance? Are amplifications affected more than deletions or vice versa?

Author's Response: We thank the reviewer for this comment and we understand the reviewer's question. We describe below the challenges of estimating the X% decrease in CNV detection accuracy using 10x data compared to full transcript sequencing.

An important thing to consider about the reviewer's request is the technical possibility to generate full transcript and 3' sequencing data from the same set of cells. For this to happen, we have to have the same RNA molecule's to be sequenced by 10x-type-of-technology, and without being damaged, it needs to be sequenced again by the full length transcript sequencing. To our knowledge, this technology currently does not exist.

Regardless, in order to evaluate a reliable X% change between full transcript and 3' sequencing technologies, we have to have numerous samples where the real copy numbers (i.e., gold standard) available for assessing the accuracy comparisons. Moreover, it is also necessary a tumor dataset where 3' and full transcript sequencing are performed at the same time. Thus, we have to have all of the following datasets from one tumor sample from the same species:

1. Ground truth CNV data
2. Smart-seq data
3. 3' sequencing data (such as 10x)

In our previous revision, we have discussed extensively the scarcity of datasets where (1) and (2) exist. In fact, we found out that there are only a handful of tumor datasets where at least (3) exists.

The only option for estimating X% decrease is to simulate the data. This is, however, requires an extensive model building, testing, and validation. We believe that this is out of the scope of CaSpER. There are many papers on the sole topic of single cell RNA-seq data simulation^{1,2}. Even these papers do not take the genetic architecture of the samples into consideration, i.e., they cannot simulate CNV events in the samples.

Nevertheless, we assessed the suitability of CaSpER 3' transcript sequencing data using the multiple myeloma 10X RNA-Seq data (MM135) presented in HONEYBADGER study. Even though the exact ground truth CNV calls are unknown for this study, we show the concordance of CaSpER CNV calls with HONEYBADGER.

This MM135 data contains normal and MM cells, therefore, we first performed TSNE to separate MM and normal cells (Figure 2). We also checked the expression levels of MM markers (SDC1, CD38) in all cells to better identify MM cells (Figure 3). We next normalized the expression values of MM cells using normal cells. We plotted heatmap of median filtered expression data (Supplementary Figure 24). We observed large scale deletion events in chromosome 13, 18 and 22 from gene expression heatmap. BAF shift plot showed concordant shifts in chromosome 13 and 18 (Supplementary Figure 25). We also calculated large scale event summary for MM135 (Supplementary Figure 26). As it is seen from the below large scale CNV event summary plot, CaSpER identified various large scale CNV events including chromosome 13 and 22 as expected (Supplementary Figure 26). Moreover these events are also reported in HONEYBADGER paper. In summary, we recommend using CaSpER on 10x data. We added documentation specifically for 10x RNA-Seq data at our website.

We have updated the manuscript to include these analyses (line 488 and Supplementary Information).

Figure 2. TSNE plot for 10x MM135 data. Nearest neighbor clustering labels are shown in the plot. Clusters with labels 0, 1, 3 and 4 corresponds to MM cells.

Figure 3. Expression levels of MM markers (SDC1 and CD38) are plotted. Blue colors indicate higher expression.

Supplementary Figure 24. Heatmap of smoothed median filtered 10X RNA-Seq data. Whole chromosome chromosome 13, 16 and 22 deletion events can be seen from the heatmap

Supplementary Figure 25. BAF shift plot for sample MM135 is shown. Whole chromosome BAF shifts for chr 13 and 18 can be seen, as expected.

Supplementary Figure 26. Large scale CNV event summary for MM135 data.

Minor comments:

1. The authors note in the "Comparing the performance of CaSpER with existing tools" that "We also compared the performance of CaSpER with HoneyBADGER and inferCNV". However, only HoneyBADGER appears to be compared. Please rephrase to reflect the actual comparisons performed.

Author's Response: We thank the reviewer for this comment. We have rephrased the line 344 as "We also compared the performance of CaSpER with HoneyBADGER on bulk and single cell RNA-Seq datasets."

2. For the bulk RNA-seq data, the authors note that CaSpER "achieves 95% TPR and 0.3% FPR for detecting large scale deletion events" for a meningioma dataset. However, for a more complex cohort of breast cancer datasets, CaSpER achieves a "60.3% true positive rate (TPR) and 3.8% FPR for detecting large scale amplification events, and 79% TPR and 7.7% FPR for detecting large scale deletion events." Could this decrease in performance be attributed to worse tumor purity in the breast cancer data? Are purity estimates available for these bulk RNA-seq samples and are they consistent with the assumption the authors note on line 607 that "we assume that there may be around 30% impurity in the tumor samples."

Author's Response: We thank the reviewer for this comment. We believe lower clonality and higher tumor purity rates in meningiomas lead to better CNV event identification. Similarly, high clonality rates and lower tumor purity in GBM and breast cancer tumors lower the detection accuracy of low-level CNV events. We investigated the tumor purity rates from TCGA cohort. The average purity for breast cancer dataset (BRCA) is 72%, ovarian cancer (OV) dataset is 86% and for low grade glioma (LGG), which should be similar to meningioma, is 89% (Supplementary Figure 27). These results show that for BRCA the admixture rate (impurity) is around 30% which is consistent with our assumption.

We have updated the manuscript to include these analyses (line 607).

Supplementary Figure 27. Boxplot of tumor purity percentages across different cancer types in TCGA cohort. Breast cancer (BRCA), Ovarian cancer: OV, Low grade glioma: LGG

- 1 Zappia L, Phipson B, Oshlack A. Splatter: Simulation of single-cell RNA sequencing data. *Genome Biol* 2017;**18**:. <https://doi.org/10.1186/s13059-017-1305-0>.
- 2 Zhang X, Xu C, Yosef N. Simulating multiple faceted variability in single cell RNA sequencing. *Nat Commun* 2019;**10**:. <https://doi.org/10.1038/s41467-019-10500-w>

REVIEWERS' COMMENTS:

Reviewer #3 (Remarks to the Author):

In this revision, to assess CaSpER's ability to analyze 3' only scRNA-seq data, the revised manuscript includes new analysis of an MM135 sample assayed by 10X 3' sequencing with many single cells. The authors note that while ground truth is not available, CaSpER CNV calls are concordant with previous CNV calls by HoneyBADGER.

Major comments:

1. As both the MGH31 and the MM135 sample have more than 7 single cells, does the concordance of CaSpER with previous analyses change when analyzing 7 cells versus all cells? Does the parameter estimates for the HMM model change as the authors expect when analyzing only 7 versus all cells?
2. The MM135 sample also notably has many non MM cells (SDC1- and CD38-). Does CaSpER properly identify these normal cells as not having any CNVs? Or are there false positives (CNVs called in normal cells)? Is this rate within what has been previously estimated?

Minor comments:

1. Please clarify how many single cells were analyzed from 10X, as was done for DR-Seq.
2. It is very difficult to identify the BAF shifts for chr13 and 18 in Supp Figure 25, particularly given the many small-sized BAF shifts on every chromosome. Please highlight the region that readers should focus on as was done in Fig 3B, Supplementary Figure 21, and other BAF shift plots. The BAF shift plot for MM135 is also notably noisier than other samples analyzed. Is this due to the 3' only sequencing of 10X or other quality aspects specific to this sample?
3. I appreciated the authors' discussion on the reasons behind CaSpER's performance differences in bulk and scRNA-seq data. While these responses are available to me in the reviewer response, these discussions have not been incorporated into the text for the reader. Please include a discussion in the conclusion or supplementary methods so that readers may benefit from this information as well.

REVIEWERS' COMMENTS:

Reviewer #3 (Remarks to the Author):

In this revision, to assess CaSpER's ability to analyze 3' only scRNA-seq data, the revised manuscript includes new analysis of an MM135 sample assayed by 10X 3' sequencing with many single cells. The authors note that while ground truth is not available, CaSpER CNV calls are concordant with previous CNV calls by HoneyBADGER.

Author's Response: We thank the reviewer for the careful consideration of our manuscript and the insightful comments.

Major comments:

1. As both the MGH31 and the MM135 sample have more than 7 single cells, does the concordance of CaSpER with previous analyses change when analyzing 7 cells versus all cells? Does the parameter estimates for the HMM model change as the authors expect when analyzing only 7 versus all cells?

Author's Response: We thank the reviewer for this comment. As we stated earlier, the complete ground truth CNV calls are unknown for neither MM135 and nor MGH31 study since there is no matching large scale genome sequencing or similar orthogonal data for validating CNVs. However, MM135 has validated CNVs for targeted regions using low throughput FISH-based cytogenetic experiments. Among these, most of the MM cells in MM135 harbor large scale chromosome 13 and 18 deletions, as are also shown in the HONEYBADGER study. Therefore we focus on these large scale events and randomly selected n=7, n=50 and n=100 cells among the MM135 cells that harbor 18p and 13q deletion. When we sampled n=7 cells, the accuracy of calling chromosome 13 and 18 deletions were 71%, whereas with n=50 accuracy was 92% and with n=100 cells accuracy was 94%. Supplementary Figure 33 shows the large scale CNV event calls for chromosome 18 and 13 for sampled n=7, n=50 and n=100 cells. As it is seen from Supplementary Figure 33, concordance of CNV calls increases with the number of cells. We calculated the mean values of the normal distributions corresponding to 5 copy number states (q1:homozygous deletion, q2:heterozygous deletion, q3:neutral, q4:amplification, q5:high-level amplification) that are derived from randomly selected n=7, n=50, n=100 cells and all cells in MM135 dataset. We observed that the initial HMM model parameters change when different numbers of cells are analyzed (Supplementary Figure 34).

Similarly in MGH31, based on CaSpER and HONEYBADGER calls we know that most of the MGH31 cells harbor chromosome 10 deletion. Therefore we first randomly selected n=7 and n=50 cells among the MGH31 cells that harbor 10p and 10q deletion. When we sampled n=7 cells accuracy of calling chromosome 10 deletions were 35%, whereas with n=50 accuracy was 99%. Supplementary Figure 35 shows the large scale CNV event calls for chromosome 18 and 13 for randomly selected n=7 and n=50 cells. As it is seen from Supplementary Figure 35, the concordance of CNV calls increases with the number of cells. We calculated the mean values of the normal distributions corresponding to 5 copy number states (q1:homozygous deletion, q2:heterozygous deletion, q3:neutral, q4:amplification, q5:high-level amplification) that are derived from randomly selected n=7, n=50, n=100 cells and all cells in MM135 dataset. We

observed that the initial HMM model parameters change when different numbers of cells are analyzed (Supplementary Figure 36).

Updated Supplementary Figure 33. *Large scale CNV calls for chromosome 18 and 13 for randomly selected $n=7$, $n=50$ and $n=100$ cells that are known to harbor these deletion events. Blue cells correspond to the deletion event whereas light blue cells correspond to neutral events. Rows correspond to single cells.*

Updated Supplementary Figure 34. *Mean values of the normal distributions corresponding to 5 copy number states (q1:homozygous deletion, q2:heterozygous deletion, q3:neutral,*

q4:amplification, q5:high-level amplification) that are calculated from randomly selected n=10, n=50, n=100 and all cells in MM135 dataset.

Updated Supplementary Figure 35. Large scale CNV calls for chromosome 10p and 10q for randomly selected n=7 and n=50 cells that are known to harbor these deletion events. Blue cells correspond to deletion event whereas light blue cells correspond to neutral events. Rows correspond to single cells.

Updated Supplementary Figure 36. *Mean values of the normal distributions corresponding to 5 copy number states (q1:homozygous deletion, q2:heterozygous deletion, q3:neutral, q4:amplification, q5:high-level amplification) that are calculated from randomly selected n=7 and n=50 in MGH31 dataset.*

2. The MM135 sample also notably has many non MM cells (SDC1- and CD38-). Does CaSpER properly identify these normal cells as not having any CNVs? Or are there false positives (CNVs called in normal cells)? Is this rate within what has been previously estimated?

Author's Response: We thank the reviewer for this comment. As the reviewer pointed out, MM135 data contains normal and MM cells, However, cell sorting was not performed before sequencing. Therefore, we do not know exactly which cells are normal and which cells are MM and we had to identify these cells computationally. We estimated normal cells computationally by performing tSNE-based dimensionality reduction followed by clustering and we manually annotated the normal cells. This approach for estimating normal cells might include some MM cells. We next normalized the expression values of MM cells using estimated normal cells and calculated CNV events in MM cells. Our analysis workflow is similar to HONEYBADGER where they also computationally estimated normal cells and used those cells as control for calculating CNV events in MM cells. Since normal cells are used as control for normalizing MM expression values, it is not possible to calculate false positive CNV events in these normal cells. In MGH31 study, we have used GTEX brain expression data as control. Therefore it was possible to estimate FPR of CNV events in normal cells.

We also would like to point out that we have addressed a similar comment about the CNV's called from normal cells in Patel et al dataset and shown that there are considerably smaller number of CNV calls in normal cells than tumor cells (new text line 293).

Minor comments:

1. Please clarify how many single cells were analyzed from 10X, as was done for DR-Seq.

Author's Response: We thank the reviewer for this comment. There are 947 single cells in MM135 10x study. We have included this in the manuscript (new text line 510).

2. It is very difficult to identify the BAF shifts for chr13 and 18 in Supp Figure 25, particularly given the many small-sized BAF shifts on every chromosome. Please highlight the region that readers should focus on as was done in Fig 3B, Supplementary Figure 21, and other BAF shift plots. The BAF shift plot for MM135 is also notably noisier than other samples analyzed. Is this due to the 3' only sequencing of 10X or other quality aspects specific to this sample?

Author's Response: We thank the reviewer for this comment. We have highlighted the BAF shift regions in Supplementary Figure 24 (previously Supplementary Figure 21). As the reviewer

has pointed out the noisy BAF shift plot might be due to 3' only sequencing. As it is shown in Supplementary Fig 22 there is a very high 3' read depth bias in 3' sequencing. For full transcript sequencing, we do observe a slight 3' bias. However, in comparison to 3' sequencing, full transcript sequencing exhibits a much less bias when 5' and 3' ends are compared in terms of read depth coverage.

Updated Supplementary Figure 24. BAF shift plot for sample MM135 is shown. Whole chromosome BAF shifts for chr 13 and 18 is highlighted. Both chromosomes exhibit BAF shift that is higher than 0.4 BAF shift for most of the SNVs.

3. I appreciated the authors' discussion on the reasons behind CaSpER's performance differences in bulk and scRNA-seq data. While these responses are available to me in the reviewer response, these discussions have not been incorporated into the text for the reader. Please include a discussion in the conclusion or supplementary methods so that readers may benefit from this information as well.

Author's Response: We thank the reviewer for this comment. We have included the discussion on the reasons behind CaSpER's performance differences in bulk and scRNA-seq data in the Supplementary methods (New text line 157-196).